# Baffin Bay sea ice extent and synoptic moisture transport drive water vapor isotope ($\delta^{18}$O, $\delta^2$H, deuterium excess) variability in coastal northwest Greenland

Pete D. Akers[1], Ben G. Kopec[2], Kyle S. Mattingly[3], Eric S. Klein[4], Douglas Causey[2], Jeffrey M. Welker[2,5]

1. Institut des Géosciences et l'Environnement, CNRS, Saint Martin d'Hères, 38400, France.
2. Department of Biological Sciences, University of Alaska Anchorage, Anchorage, 99508, AK, USA.
3. Institute of Earth, Ocean, and Atmospheric Sciences, Rutgers University, Piscataway, 08854, NJ, USA
4. Department of Geological Sciences, University of Alaska Anchorage, Anchorage, 99508, AK, USA.
5. Ecology and Genetics Research Unit, University of Oulu, Oulu, 90014, Finland, and University of the Arctic-UArctic

Correspondence to: Pete D. Akers (pete.d.akers@gmail.com)

## Abstract

At Thule Air Base on the coast of Baffin Bay (76.51° N 68.74° W), we continuously measured water vapor isotopes ($\delta^{18}$O, $\delta^2$H) at high frequency (1 s$^{-1}$) from August 2017 through August 2019. Our resulting record, including derived deuterium excess (*dxs*) values, allows analysis of isotopic–meteorological relationships at an unprecedented level of detail and duration for High Arctic Greenland. We examine isotopic variability across multiple temporal scales from daily to interannual, revealing that isotopic values at Thule are predominantly controlled by sea ice extent in northern Baffin Bay and synoptic flow pattern. This relationship can be identified through its expression in five interacting factors: a) local air temperature, b) local marine moisture availability, c) the North Atlantic Oscillation (NAO), d) surface wind regime, and e) land-based evaporation/sublimation. Each factor's relative importance changes based on temporal scale and in response to seasonal shifts in Thule's environment. Winter sea ice coverage forces distant sourcing of vapor that is isotopically light from fractionation during transport while preventing isotopic exchange with local waters. Sea ice breakup in late spring triggers a rapid isotopic change at Thule as the newly open ocean supplies warmth and moisture that has ~10‰ and ~70‰ higher $\delta^{18}$O and $\delta^2$H values, respectively, and ~10‰ lower *dxs* values. Sea ice retreat also leads to other environmental changes, such as sea breeze development, that radically alter the nature of relationships between isotopes and many meteorological variables in summer. On synoptic time scales, enhanced southerly flow promoted by negative NAO conditions produce higher $\delta^{18}$O and $\delta^2$H values and lower *dxs* values. Diel isotopic cycles are generally very small as a result of a moderated coastal climate and counteracting isotopic effects of the sea breeze, local evaporation, and convection. Future losses in Baffin Bay sea ice extent will likely shift mean annual isotopic compositions toward more summer-like values, and local glacial ice could potentially preserve isotopic

evidence of past reductions. These findings highlight the influence that the local environment can have on isotope dynamics and the need for dedicated, multi-season monitoring to fully understand the controls on water vapor isotope variability.

## 1. Introduction

The Arctic environment is rapidly entering a new state dominated by warmer air temperatures in all seasons, accompanied by dramatic sea ice loss, ecological changes, and ice sheet mass loss. Improving our knowledge of how the Arctic water cycle
responds to regional warming and develops feedbacks within the changing climate is key to planning for a more resilient Arctic environment and economy (Meier et al., 2006; Vihma et al., 2016). While many modeling and satellite-based studies are examining this issue (NOAA, 2020), ground-based observations are still critical to understanding the present-day hydrological cycle and tracking ongoing changes in local environments (e.g., Steen-Larsen et al., 2014; Bonne et al., 2015; Klein et al., 2015). Despite this need, the Arctic is data-sparse in terms of spatial coverage and temporal span of quality surface
observations. More environmental monitoring efforts are needed across the region and particularly at the highest latitudes to better capture the hydroclimate processes and moisture transport across the Arctic domain.

The stable isotopes of water are well-established environmental tracers of the water cycle, reflecting both local weather conditions as well as moisture history and synoptic patterns (Craig, 1961; Dansgaard, 1964; Rozanski et al., 1993; Gat, 1996). Although less commonly studied than lower latitude regions, analyses across the Arctic have also shown that local observations
of water isotopes can reveal important connections to wider atmospheric parameters such as teleconnections and storm patterns (e.g., Moorman et al., 1996; Welker et al., 2005; Theakstone, 2011; Bailey et al., 2015b; Puntsag et al., 2016; Putman et al., 2017). While our knowledge of isotope dynamics in the hydrosphere is largely based on studies of precipitation and surface waters (e.g., Dansgaard, 1964; Rozanski et al., 1993; Welker, 2000; Gurney and Lawrence, 2004), the development of field-deployable infrared laser spectrometers has fostered a number of recent studies focused on water vapor. With their ability to
analyze water isotopes with high frequency ($>1$ min$^{-1}$) in a continuous vapor flow, these spectrometers are well-suited to long-term monitoring studies (e.g., Sturm and Knohl, 2010; Aemisegger et al., 2012; Bailey et al., 2015a; Wei et al., 2019). Several studies at land-based sites in the high latitudes have reported continuous water vapor isotopic observations for periods ranging from a single season to multiple years (Table S1), and their data sets are proving highly useful in understanding polar hydroclimate dynamics.

These observational studies are critical to tracking and understanding the ongoing climate changes in high latitude regions, especially as the impacts of amplified polar warming on global weather patterns are hotly debated (Francis and Vavrus, 2012; Francis et al., 2018; Pithan et al., 2018; Nusbaumer et al., 2019; Cohen et al., 2020). Already, the published results from these polar water vapor isotope sites have highlighted the potential in tracking shifting moisture sources during extreme weather events (Bonne et al., 2015; Klein et al., 2015) as well as the importance of local geography in short-term isotopic variability
(e.g., Kopec et al., 2014; Bréant et al., 2019). However, these sites reflect only a small portion of the vast and diverse polar

environment, and fewer than half report data covering multiple years. More spatial coverage and longer periods of record are needed to fully harness water vapor isotope monitoring to resolve unanswered questions on polar hydroclimate processes and to accurately predict and detect future changes.

Greenland is particularly important to our efforts to better understand past, current, and future climate, as its immense ice sheet has archived millennia of past climate changes (e.g., Steffensen et al., 2008) and environmental feedbacks from the increasing Greenland Ice Sheet loss reverberate globally (e.g., Box et al., 2012; Nghiem et al., 2012; Castro de la Guardia et al., 2015). The declining extent and duration of sea ice in its surrounding oceans are altering atmospheric moisture fluxes and transport across the Arctic (Gimeno et al., 2019; Nusbaumer et al., 2019), restructuring marine and terrestrial ecology (Bhatt et al., 2017; Laidre et al., 2020), and harming the health and welfare of native communities (Meier et al., 2006). Northwest Greenland in particular is one of the fastest warming regions on Earth with massive ice loss observed from glacial retreat and surface ablation (van As, 2011; Carr et al., 2013; Noël et al., 2019). Despite Greenland's importance, water vapor isotope monitoring has only been reported from four locations on the island (Steen-Larsen et al., 2013; Bonne et al., 2014; Kopec et al., 2014; Bailey et al., 2015a), and one record (Summit) has not had its data published.

We present here a new multi-year dataset from Thule Air Base in northwest Greenland that has recorded stable oxygen and hydrogen ratios ($\delta^{18}O$ and $\delta^2H$, respectively) of ambient water vapor nearly continuously from 04 August 2017 through 31 August 2019. As a result, this record is the first reported for Greenland that continuously spans over two years, permitting comparative multi-year analysis of isotopic patterns and anomalies. Thule Air Base has been a focus of High Arctic research for many years (e.g., Schytt, 1955; Mastenbrook, 1968; Sullivan et al., 2008; Rogers et al., 2011; Leffler and Welker, 2013; Schaeffer et al., 2013), and our observing station on northern Baffin Bay allows a focus on how changing seasonal and interannual sea ice coverage affect the local climate and water vapor isotopes.

Our research addresses how interactions between the hydroclimate, cryosphere, and ocean are manifested in water vapor isotopes, and we focus here on identifying how these broad interactions are expressed and detected in local weather observations. The long period of record and very high frequency of isotopic (one observation per second, aggregated to 10 minutes) and meteorological observations (one observation per 10 minutes) enable us to study these water vapor isotope dynamics across varying temporal scales from daily to interannual. This Thule dataset and our continuing observations are part of the MOSAiC (Multidisciplinary drifting Observatory for the Study of Arctic Climate) project's Arctic Water Isotope Network (Welker et al., 2019; MOSAiC, 2020), a series of nine pan-Arctic sites simultaneously observing water vapor isotopes. To that end, our research at Thule provides a focused examination of the many local and regional environmental controls on water vapor isotopes at a coastal High Arctic site in northwest Greenland.

## 2. Field description

### 2.1 Local landscape

Our water vapor isotope observation station is located at Thule Air Base on the Pituffik Peninsula in far northwest Greenland (Fig. 1). Landmarks are referred to here by their locally-known English names with Greenlandic names also listed where known. The main base and airfield, established in 1951 by the United States and Denmark, occupy a low-lying (<100 m a.s.l.) region along the North River (Pituffiup Kuussua) facing North Star Bay and Bylot Sound, small arms of Baffin Bay that are semi-protected by Saunders and Wolstenholme Islands (Appat and Qeqertarsuaq). The main base is bracketed to the south and north by two broad ridges (summits: 300 and 240 m a.s.l., respectively) locally called South Mountain (Akinnarsuaq) and North Mountain. The land-terminating margin of the Greenland Ice Sheet is approximately 14 km southeast of the main base and extends as low as 300 m a.s.l. This local section of the ice sheet, known as the Tuto Ice Dome, has a summit greater than 800 m a.s.l. and a mass balance that is semi-independent from the main Greenland Ice Sheet (Schytt, 1955; Hooke, 1970; Reeh et al., 1990).

Other military installations, both abandoned and presently occupied, are scattered throughout the wider Thule Defense Zone that covers the northern half of the Pituffik Peninsula. Near the southern edge of the Thule Defense Zone lies higher terrain including Pingorsuit, the highest point of the peninsula at 815 m. In addition, rugged terrain and mountains over 600 m a.s.l. run along the southern coastline of the peninsula toward Cape York (Serfarmiut Nuuat). Wolstenholme Fjord (Uummannap Kangerlua) forms the northern coast of the Pituffik Peninsula, separating it from Steensby Land and the North Ice Cap (summit: >1200 m). The three marine terminating glaciers that feed into the fjord have severely retreated and thinned in the past century, including a retreat of over 8 km for the largest glacier (Harald Moltke/Ullip Sermia) since the 1940s (Mock, 1966, Hill et al., 2018).

### 2.2 Local climate

Thule has a polar desert/semi-desert climate (Gold and Bliss, 1995; Sullivan et al., 2008), with a mean annual temperature of -10.0°C and mean monthly temperatures ranging from -23.7°C in February to +6.5°C in July (2000–2018 observations, USAF, 2019). The high latitude (76° N) produces long periods of polar night (November–February) and midnight sun (May–August), and daylength changes by 15–30 minutes each day during transitional months. Extended periods of extremely cold temperatures below -25°C are frequent in winter, and frosts and snowfall are possible in all months. Mean monthly temperatures are above freezing for only the three summer months, but summer can be surprisingly mild under continuous insolation with inland temperatures sometimes rising above 10°C. Coastal sea ice and the seasonal snow pack develop by October and last through May to early June (Barber et al., 2001; Fetterer et al., 2017; Stroeve and Meier, 2018; USAF, 2019). At early spring maximum, sea ice covers Baffin Bay to an extent over 1000 km south of Thule, except for the biologically important North Water Polynya located to the northwest (Barber et al., 2001; Tang et al., 2004; Heide-Jørgensen et al., 2016).

Annual precipitation is 130 mm water equivalent with half of this precipitation falling mostly as rain in June–August (USAF, 2019). Moisture in western Greenland primarily is sourced from the North Atlantic, but local sources of moisture, such as Baffin Bay, increase in importance during sea ice retreat in summer and early fall (Sodemann et al., 2008; Gimeno et al., 2019; Nusbaumer et al., 2019). Although the wind at Thule makes accurate snow measurements difficult and prone to overestimation (Chen et al., 1997), existing records report an average annual snowfall of 900 mm and October–December as the snowiest months (USAF, 2019). Synoptic storm systems dominate short-term weather variability with the prevailing storm track consisting of extratropical cyclones that form to the south in Labrador Bay before tracking north and strengthening over Baffin Bay (Chen et al., 1997). These cyclones can be very intense, including a 333 km h$^{-1}$ observation at Thule in 1972 that is among the highest winds ever recorded on Earth (Stansfield, 1972; Moore, 2016). Atmospheric rivers (narrow corridors of strong horizontal moisture advection) have outsized effects on polar weather (Woods et al., 2013; Liu and Barnes, 2015; Wille et al., 2019), and periods of intense ice sheet melt and mass loss in Greenland, such as July 2012, often coincide with atmospheric rivers impacting the ice sheet (Neff et al., 2014; Bonne et al., 2015; Mattingly et al., 2018; Ballinger et al., 2019; Oltmanns et al., 2019).

Differential radiative heat loss between the ice sheet and coastal region drives katabatic winds (van As et al., 2014) from the east and southeast that dominate the wind regime in Thule outside of the summer months. From April through September, a west to northwest sea breeze develops along the coast due to differential warming between the ocean and the local area of snow-free tundra (Atkinson, 1981), and it brings cooler marine surface air, often associated with fog, several kilometers inland. This sea breeze generally strengthens in the afternoon and weakens at night, although continuous summer insolation often prevents the full shift to nightly katabatic flow that has been observed at other coastal polar sites (e.g., Kopec et al., 2014). These two wind patterns at Thule produce an overall bimodal wind distribution with azimuth peaks at 100° (katabatic) and 270° (sea breeze).

## 3. Methods

### 3.1 Equipment setting

A Picarro L2130-i with Standards Delivery Module (SDM) setup was installed in a temperature-controlled building on the crest of South Mountain (76.514° N 68.744° W, 229 m a.s.l.) in October 2016, with quality data observations beginning in August 2017 after a system reset. Local air flow on top of South Mountain is unimpeded by topographic barriers from any direction within 20 km. Potential anthropogenic impacts to observations at the monitoring site are limited as the main base lies 2 km to the north and 200 m lower in elevation. Typically, less than 20 vehicles a day travel the access road that is 200 m from the building housing the L2130-i, and visitors are uncommon.

The L2130-i uses cavity ring-down spectroscopy to measure $\delta^{18}O$ and $\delta^2H$ in ambient water vapor and is particularly suited to long-term monitoring with limited human interaction. The vapor collection point is five meters above the ground surface on

the building's roof-edge, away from parking zones and building exhaust, and outfitted with a plastic cap to prevent precipitation entering the inlet. Five meters of Bev-A-Line IV EVA tubing (3/4 in diameter) connects the inlet to the L2130-i with an air residence time of less than three minutes. The tubing is not independently heated, but it is routed immediately into the heated

building interior which is always warmer than ambient outside air, even during extreme warm events in summer. This limits potential condensation within the tubing, and data quality control included a cull on observations with abnormal isotopic values that suggested possible condensation or precipitation contamination.

Certain tubing types, such as Synflex tubing, are known to adversely affect water vapor isotopic values by a slower vapor response time, increased fractionation of $\delta^2H$, and general smoothing of $\delta^2H$ values at timescales less than 30 minutes (Sturm

and Knohl 2010; Tremoy et al., 2011). We see no evidence of substantial tubing effects on our data, as the $\delta^2H$ has the same response time and degree of variability as $\delta^{18}O$ in our 10 minute resolution data, and the strongest $\delta^{18}O$–$\delta^2H$ correlations are with zero time lag. Additionally, we do not observe any significant differences in the $\delta^{18}O$–$\delta^2H$ relationship between 10 minute, hourly, and daily aggregations which supports that both isotopic values are varying at a similar short-term frequency.

### 3.2 Isotopic observations and calibrations

Our L2130-i at Thule took an observation of the oxygen and hydrogen isotopic ratios in ambient water vapor approximately once per second, and these ratios are expressed in $\delta$-notation relative to the Vienna Standard Mean Ocean Water (VSMOW) standard as:

$$\delta = \frac{R}{R_{VSMOW}} - 1 \tag{1}$$

where $R$ is the measured ratio of rare to abundant isotopologue ($^{18}O$ / $^{16}O$ or $^2H$ / $^1H$) and $R_{VSMOW}$ is the matching isotopic ratio

of the reference standard water VSMOW. Deuterium excess values (*dxs*) were calculated from each isotope observation by $dxs = \delta^2H - 8 * \delta^{18}O$ (Dansgaard, 1964). Observations were continuous from August 2017 through August 2019 except for one period when the analyzer suffered power failure (05–13 Sep 2018). Some additional gaps in the data were introduced as a result of quality checking (S1) and standard calibrations, but these gaps were typically less than 24 hours in duration. We acknowledge that technical difficulties and some initial system design choices have prevented our data from being calibrated

to the maximum quality level recommended by peer publications, and we attempt to be fully forthright with these issues here.

Every 25 hours, two water standards, USGS 45 ($\delta^{18}O$ = -2.24‰, $\delta^2H$ = -10.3‰) and USGS 46 ($\delta^{18}O$ = -29.80‰, $\delta^2H$ = -235.8‰), were injected sequentially by the SDM into a stream of air supplied by a dry air canister in order to normalize to the VSMOW/SLAP (Standard Light Antarctic Precipitation) scale. Each standard was injected for ten minutes, and the mean of the last 200 seconds of each standard injection was kept as the final isotopic value for each standard calibration. Preliminary

tests at Thule suggested that acceptably stable isotopic value readings were reached in less than ten minutes, and in 51 calibrations that were run in August and September 2019 (Table S2), the final 200 observations (i.e., the data used to calculate

calibration values) have flat trends suggesting acceptable isotopic stability (Table S3; mean slope ± 1 standard deviation (σ) for USGS 45: $\delta^{18}O$ = -0.0003 ± 0.0016‰ $s^{-1}$, $\delta^2H$ = 0.0026 ± 0.0062‰ $s^{-1}$; USGS 46: $\delta^{18}O$ = 0.0014 ± 0.0016‰ $s^{-1}$, $\delta^2H$ = 0.0026 ± 0.0064‰ $s^{-1}$). Our calibration interval and duration is longer and shorter, respectively, than in other similar studies

(e.g., Bastrikov et al., 2014; Bailey et al., 2015a; Bréant et al., 2019), but deemed a necessary compromise to extend the life of the dry air canister and standard waters at the remote location while still allowing an estimate of the machine's accuracy drift.

The water standards used in calibration were originally chosen to cover the summer water vapor isotopic values at Thule before a project extension, but unfortunately most non-summer isotopic values are below the standard waters' value coverage. As a

result, our reported values for non-summer observations may suffer from undetermined accuracy biases relative to the VSMOW/SLAP scale, particularly at the lowest isotopic values. The potential impact on our analyses is likely limited as long as any bias is internally consistent and/or small in magnitude, and comparison of our reported isotopic values to other polar sites did not reveal any clear sign of major inaccuracies. However, anyone performing a future comparison between our Thule data and the data from another site should certainly take into account the potential for unquantified accuracy bias below the

isotopic values of USGS 46. A wider range of standard waters is planned for continued future operation of the Thule observation station.

At lower water vapor mixing ratios (<1500 ppmv), the L2130-i loses accuracy and precision, and a humidity response curve must be developed in order to correct for any analytical bias in isotopic values (Steen-Larsen et al., 2013; Bastrikov et al., 2014; Bailey et al., 2015a). In July 2019, the two standard waters were each injected for 10 minutes into dry air at ten different

flow rates to produce a sequence of standard observations between 500 and 7000 ppmv. Humidity response curves were created using nonlinear regression to relate isotopic value offsets due to sensor biases at low humidity with water vapor mixing ratios, and ambient isotopic data were then corrected for sensor bias with these curves based on the ambient mixing ratio at the time of observation (S2, Fig. S1, Table S4). Ambient observations with a mixing ratio less than 500 ppmv (fewer than 500 individual 1 $s^{-1}$ observations) fell outside our humidity response observations and were thus excluded from further analysis.

The analyzer has a consistent accuracy offset of +1.9‰ and +1.1‰ in all $\delta^{18}O$ and $\delta^2H$ observations, respectively, at mixing ratios higher than 1500 ppmv. The accuracy bias changes below 1500 ppmv, and the offset for $\delta^{18}O$ and $\delta^2H$ is -4.5‰ and -20‰, respectively, at 500 ppmv. Ambient data were corrected for these offsets, and 95% confidence intervals for humidity response corrections for $\delta^{18}O$ ranged from ±0.24‰ (500 ppmv) to ±0.09‰ (all observations >1500 ppmv) and for $\delta^2H$ ranged from ±1.6‰ (500 ppmv) to ±0.6‰ (all observations >1500 ppmv). Precision linearly decreases with the reciprocal of the water

vapor mixing ratio (S2, Fig. S1, Table S5). Standard errors of the mean $\delta^{18}O$ values for each flow rate in the humidity response curve ranged from 0.57‰ (500 ppmv) to <0.05‰ (>5000 ppmv) while the standard errors for $\delta^2H$ ranged from 3.7‰ (500 ppmv) to <0.4‰ (>5000 ppmv). In comparison, each ambient water vapor data point from the highest resolution used in our analysis is mean value of 600 observations (i.e., 10 minutes) compared to the 200 observations in the humidity response curve,

and thus the precision of our ambient data should as precise or greater than observed in the humidity response curves. After the humidity response corrections, data were quality checked again and offending data (e.g., extreme outliers, residual calibration vapor impacts, extreme moisture spikes unrelated to weather) removed. The final quality checked isotope data were aggregated into 10 minute, hourly, and daily means.

The calibration system described as above was installed in August 2019. Before this installation, standards were injected into an air stream dried by a Drierite column rather than supplied by a dry air canister. However, the Drierite column did not fully remove all ambient moisture and increasingly lost effectiveness over time at Thule. The residual ambient moisture had a significant effect on observed isotopic values during calibrations, and very few of the calibration runs prior to the dry air canister installation accurately recorded the standard values needed to correct for machine analytical drift. As a result, we do not have accurate daily calibrations for most of our observation period and cannot apply a correction to account for possible analytical drift.

We believe that the database can still provide robust analytical results despite the lack of drift correction. Although we cannot be certain that an estimate of recent stability is representative of the stability over the entire observation record at Thule, the calibrations completed after the dry air installation show consistently high precision and limited day to day sensor drift (Table S2). The standard deviation of the 51 calibration means for $\delta^{18}O$ and $\delta^2H$ were 0.2‰ and 0.7‰, respectively, with deviations the same for both standard waters, and the maximum difference between any two calibration run means were 1.2‰ and 3‰ for $\delta^{18}O$ and $\delta^2H$, respectively. This magnitude of recent drift at Thule and the magnitude reported at other similar stations (e.g., Steen-Larsen et al., 2013; Bréant et al., 2019) is very small relative to the natural short-term isotopic variability and large seasonal patterns that we observe in the ambient water vapor. As our study focuses on these large isotopic changes, the risk of substantial impact on our analyses and conclusions from undetected short-term sensor drift is minor. Comparison of isotope-climate relationships across the period of record and before/after unexpected shutdowns show no major changes that suggest long-term drift or post-restart shifts in instrument sensitivity that would negatively affect our analyses.

### 3.3 Meteorological observations

An automated weather station (SMT, for South Mountain), located on the roof of the building that houses the L2130-i, took a reading of air temperature, relative humidity (with respect to the saturation vapor pressure over ice), and station barometric pressure every 10 minutes throughout the duration of isotopic observations (Muscari, 2018). For analysis at different temporal resolutions, these data were also aggregated into hourly and daily means. The L2130-i takes a reading of the water vapor mixing ratio with every isotopic observation, and observations of this variable were added to the SMT datasets as an independent meteorological observation. The mixing ratios recorded by the L2130-i have a very strong linear relationship ($r^2$ = 0.99) with mixing ratios calculated from the SMT observations, but the L2130-i mixing ratios are greater by a consistent factor of 1.23. For analytical simplicity, all mixing ratio values used in analyses and reported here are the unaltered observations from the L2130-i.

Hourly mean near-surface wind speeds and azimuths (USAF, 2019) were taken at the Thule Airport (THU), aggregated to daily values, and joined to the hourly and daily SMT databases. Wind azimuths were converted into u and v components for accurate aggregation and then reconverted back into azimuths. While the THU recording station is 170 m lower than SMT, the two stations are less than 1 km apart and their meteorological data agree well ($\rho_{temperature} = 0.98$, $\rho_{dew.point} = 0.98$, $\rho_{station.pressure} = 0.96$). Daily climate teleconnection indices for the North Atlantic Oscillation (NAO) and Arctic Oscillation (AO) were downloaded from the US National Weather Service Climate Prediction Center (NOAA, 2019) and joined with the daily SMT database. Because the AO did not produce significant correlations or results in our analyses, its discussion is limited here and largely subsumed under discussion of the NAO. Daily sea ice extent for Baffin Bay was obtained through the National Snow and Ice Data Center (NSIDC) Multisensor Analyzed Sea Ice Extent (MASIE) product (Fetterer et al., 2010).

To reduce issues with the radial nature of azimuth data during correlation analysis, wind azimuths were converted to 'katabatic deviations' where an azimuth of 100° (i.e., the mean katabatic wind azimuth) was defined as zero, and other observations were redefined as their minimum absolute degree distance from 100°. The katabatic deviations therefore range in value between 0° and 180°. While this does not fully solve the radial issue (e.g., a katabatic deviation of 80° can represent both a northly 20° wind or a southerly 180° wind), it fares well in distinguishing between the bimodal "easterly katabatic" and "westerly sea breeze" regimes.

### 3.4 Statistical analyses

Spearman correlations between isotopic and meteorological variables were calculated across the full datasets at all three temporal resolutions. Strong correlations that arise between variables in this analysis, though, may be spurious due to common responses to seasonal change. Likewise, variables that do not have a seasonal cycle, such as relative humidity, may have stronger correlations once the interference from seasonality is removed. To examine this, we removed the strong seasonal cycles of the isotopic values, air temperature, and mixing ratio by sinusoidal curves fitted to the annual cycle of each full data time series. With these seasonally-adjusted (SA) data, we calculated a second set of correlations.

Data at each temporal resolution were also binned by month, and correlations then calculated for each month to examine how isotope-climate relationships change over the course of a year. To assess temporal covariance and possible lead/lag relationships between teleconnection indices (NAO, AO) and isotopic variables, we performed cross-correlation analysis (Addinsoft, 2020) with a first differencing methodology (Peterson et al., 1998) to remove autocorrelations and avoid cross-correlation bias (Olden and Neff, 2001; Runge et al., 2014).

### 3.5 Back trajectory analysis

We compiled a Thule-specific moisture sourcing and transportation "quasi-climatology" with air parcel back trajectory analysis using the HYSPLIT tool and based on three-dimensional MERRA-2 wind field data that has a horizontal grid

resolution of 0.5° latitude / 0.625° longitude and 72 hybrid-eta vertical levels from the surface to 0.01 hPa (Stein et al., 2015; Gelaro et al., 2017). To build this "quasi-climatology", we randomly selected 10 days from each month during 1980–2018 as we judged 10 days to provide an acceptable representation of the true climatology while remaining within reasonable computation and data storage limits. We then initiated 10 day back trajectories for each randomly selected day from the

280 MERRA-2 grid point nearest Thule at the four synoptic hours (0000, 0600, 1200, and 1800 UTC) on 6 vertical levels (10, 100, 200, 500, 1000, and 1500 m above ground level), resulting in 2880 total trajectories. Following previous studies (e.g., Sodemann et al., 2008; Molina and Allen, 2019), the moisture uptake algorithm implemented in the PySPLIT python library (Warner, 2018) infers moisture uptake into the parcel when specific humidity change ($\Delta q$) is greater than +0.2 g kg$^{-1}$ and infers precipitation when $\Delta q$ is less than -0.2 g kg$^{-1}$. Later moisture uptakes are weighted more heavily than uptakes occurring prior

to precipitation (Sodemann et al., 2008). A domain covering Baffin Bay from the Davis Strait to Nares Strait was defined, and the percentage of moisture uptake occurring within and outside this domain was calculated with raw values at each grid cell multiplied by the cosine of latitude to compensate for smaller grid cell area with northward extent.

## 4. Environmental controls on water vapor isotopes at Thule

The mean annual water vapor $\delta^{18}$O, $\delta^2$H, and *dxs* values at Thule are -33.2 ± 6.1‰, -249 ± 43‰, and +16.2 ± 8.1‰, respectively

(10 min data, mean ± 1 σ). The isotopic records reveal strong annual cycles, with the highest values for $\delta^{18}$O and $\delta^2$H in summer and lowest values in winter and early spring, while *dxs* values show the opposite pattern with minimum values in summer and maximum values in winter (Fig. 2, Table 1, S3). The isotopic values also have very wide overall ranges, with $\delta^{18}$O, $\delta^2$H, and *dxs* spanning over 30‰ (-49.5‰ to -17.5‰), 230‰ (-377‰ to -142‰), and 55‰ (-7.6‰ to 47.5‰) respectively, in our 10 minute database. Temporal aggregation reduces the overall range and variability of isotopic data by

reducing extreme values, but this does not greatly affect the distribution or means. Conclusions drawn from one level of aggregation are consistent across all levels of aggregation.

The water vapor isotopes also have substantial sub-seasonal variability, and the magnitude of irregular hourly to weekly variations caused by synoptic weather events can approach 30–50% of the entire annual isotopic ranges. These large isotopic changes are typically tied to the different sourcing and transport of warm and cold sector moisture during cyclone passage, as

well as changes in air temperature, surface winds and cloud cover (Gedzelman and Lawrence, 1990; Coplen et al., 2008; Dütsch et al., 2016). Similarly, strong anticyclones and blocking events can also dominate local weather and alter atmospheric flow throughout the polar region (Davini et al., 2012; McLeod and Mote, 2016; Wernli and Papritz, 2018). Diel cycles in isotopes are very small (generally <1‰ for $\delta^{18}$O and *dxs*), but they help reveal which environmental factors are driving isotopic changes at the local scale.

Based on our statistical analyses (Table 2) and field observations, the variability of water vapor isotope ratios at Thule can be explained largely by five interacting factors: a) local air temperature, b) local marine moisture availability, c) the NAO, d)

surface wind regimes, and e) the evaporation/sublimation of local surface waters and snow. These factors interact and compete with each other, with the dominance of particular factors varying over the course of a year with the changing seasons. Together, these factors can be considered as specific expressions of the broad environmental responses to changes in Baffin Bay sea ice extent and polar synoptic flow. To clarify this complex situation, this discussion will first detail the effects of these factors specifically in the Thule environment. The impact of these factors on isotopic systematics will then be examined across seasonal, synoptic, and daily scales to reveal how observed isotopic values are produced.

## 4.1 Local air temperature

Our back trajectory analysis shows that most water vapor is supplied to Thule from evaporation in more southerly locations and then advected north along the western coast, with a substantial fraction of this water vapor having previously passed over the southern Greenland Ice Sheet (Fig. 3a). As the air mass cools in this northerly transport, water vapor will condense into clouds and precipitation that selectively removes isotopically heavier water molecules (Dansgaard, 1964; Rozanski et al., 1993). This Rayleigh fractionation results in an air mass with water vapor that is increasingly isotopically lighter (i.e., depleted in $^{18}$O and $^{2}$H) as it travels to Thule, and colder conditions at Thule create a steeper temperature gradient versus the more southerly moisture source that enhances the degree of fractionation (e.g., Rozanski et al., 1993; Bonne et al., 2014; Kopec et al., 2019). This connection is supported by the strong correlations of $\delta^{18}$O and $\delta^{2}$H with air temperature ($\rho_{\delta 18O.10min}$ = +0.76, $\rho_{\delta 2H.10min}$ = +0.73), which are surpassed in strength only by correlations with the closely linked water vapor mixing ratio (Table 2).

However, strong annual correlations between water isotopes and temperature often result from common responses between air temperature and other variables that also follow seasonal patterns (e.g., Akers et al., 2017). This also appears true for Thule, as the correlations are much weaker when seasonal cycles are removed ($\rho_{SA-\delta 18O.10min}$ = +0.24, $\rho_{SA-\delta 2H.10min}$ = +0.20). Most likely, much of the strength of the non-seasonally-adjusted correlations is due to the common response with broader environmental changes linked to seasonally-covarying sea ice extent. Despite this, the weaker seasonally-adjusted correlations show that a temperature effect is still expressed to a weaker degree in shorter-term, intra-seasonal isotopic variations (Table 2).

Local air temperature is not typically seen as a primary driver of *dxs* variability; rather, variables related to environmental conditions at the initial moisture source, such as sea surface temperature and relative humidity, are the principal controls (e.g., Merlivat and Jouzel, 1979; Vimeux et al., 1999; Pfahl and Sodemann, 2014). However, *dxs* values at Thule have a strong correlation with air temperature ($\rho_{dxs.10min}$ = -0.74), and the correlation using seasonally-adjusted data is actually stronger ($\rho_{SA-dxs.10min}$ = -0.31) than the correlations observed in $\delta^{18}$O and $\delta^{2}$H. Temperature-driven changes on *dxs* have been reported (Jouzel and Merlivat, 1984; Dütsch et al., 2017; Kopec et al., 2019), particularly for very cold conditions, and it is possible that this is contributing to the short-term correlation between *dxs* and air temperature as well as the strong negative correlations between

*dxs* and both $\delta^{18}O$ and $\delta^2H$. However, it is also possible that the correlation of *dxs* with temperature is, like with $\delta^{18}O$ and $\delta^2H$, actually responding to another environmental factor that covaries with temperature.

## 4.2 Local marine moisture availability

The isotopic composition of water vapor in an air mass is initially determined by the isotopic composition of the source water and by the environmental conditions during vapor uptake. At synoptic scales, the relative humidity near the sea surface strongly determines the *dxs* values for the resulting moisture parcel, with higher humidity producing lower *dxs* values and vice versa (Craig and Gordon, 1965; Pfahl and Sodemann, 2014) while, at longer timescales, sea surface temperature has traditionally been cited as a primary control on *dxs* values (e.g., Merlivat and Jouzel, 1979; Vimeux et al., 1999). If these isotopic signatures are well-preserved through later moisture transport, isotopic analysis of water vapor or precipitation can be used to remotely infer climate changes at the moisture source and/or shifts to different moisture sources (e.g., Merlivat and Jouzel, 1979; Feng et al., 2009; Bonne et al., 2015; Dütsch et al., 2017; Bonne et al., 2019). This has been applied most notably to ice cores in order to reconstruct moisture source changes extending back deep through time (e.g., Vimeux et al., 1999; Steffensen et al., 2008; Steen-Larsen et al., 2013; Osterberg et al., 2015).

Local polar waters have high surface relative humidity when open in summer (Fig. 3c) and supply Thule with water vapor that has high $\delta^{18}O$ and $\delta^2H$ values and low *dxs* values. At Thule, changes in the seasonal sea ice extent permit or restrict the delivery of local moisture from the nearby ocean, and this availability is a primary control on mean isotopic values. Our back trajectory analysis shows that the regions of predominant water vapor uptake for air masses arriving at Thule vary substantially over the year (Fig. 3b), supporting the role of sea ice in determining moisture sourcing. Baffin Bay is the dominant evaporative moisture source from the late spring through early winter, contributing ~50% of water vapor transported to Thule (Table S6). Substantial contributions from the Labrador Sea, Denmark Strait, Hudson Bay, and Canadian Arctic Archipelago regions are also observed during this time.

From January until May, however, local seas are extensively ice covered and the majority of water vapor present at Thule originates from more distant sources in the Labrador Sea and North Atlantic (Fig. 3b). Large evaporation events in the North Atlantic tend to occur in the cold sector of extratropical cycles (Aemisegger, 2018; Aemisegger and Papritz, 2018) where the dry air (Fig. 3c) produces water vapor with high *dxs* values. As this moisture travels north to Thule, fractionation from rain out during transport results in isotopically light water vapor, and sea ice coverage in Baffin Bay and other local Thule waters prevent isotopic exchange that would mitigate some of this fractionation. Interestingly, our analysis reveals that a large amount (>30%) of Thule's winter water vapor is supplied by sections of southern Baffin Bay and the Labrador Sea with 5–95% climatological mean sea ice cover. Moisture originating from such an environment often has very high *dxs* values from the frequent invasions of very dry Arctic air originating over expanses of continuous sea ice (Kurita, 2011) and/or vapor sourcing from snow accumulated on top of the sea ice (Bonne et al., 2019).

## 4.3 The North Atlantic Oscillation

Phase shifts in the NAO can serve to enhance or limit moisture transport from the south to Thule (Fig. 3a) as atmospheric mass is redistributed between the Arctic and North Atlantic, and changes in water isotopes resulting from these atmospheric shifts have been detected in Arctic snow and ice (e.g., Vinther et al., 2003; Vinther et al., 2010; Zheng et al., 2018) and plants (Welker et al., 2005). In western Greenland, the negative phase of the NAO (NAO-) is associated with enhanced southerly flow that brings warmer temperatures and greater regional snow, while the positive phase (NAO+) has stronger westerlies that limit the

northward penetration of southerly air masses (Sodemann et al., 2008; Bjørk et al., 2018). These effects are manifested in our observations where more negative NAO indices are correlated with a stronger Greenland anticyclone (i.e., higher Thule station pressure, $\rho_{NAO.pres} = -0.60$).

When southerly advection along the western edge of the anticyclone is promoted during NAO- phases, Thule tends to be warmer ($\rho_{NAO.SA-temp} = -0.31$) and have a higher water vapor mixing ratio ($\rho_{NAO.SA-MR} = -0.27$). Cross-correlation analysis (Fig.

4, Fig. S2) reveals that shifts to more negative NAO indices produce isotopically heavier water vapor observations with lower *dxs* values two to three days later at Thule, presumably as air advected north over Baffin Bay picks up local moisture and/or the warmer conditions reduce the degree of fractionation during transport. We did not find noteworthy isotopic correlations with the AO at Thule, likely because changes in AO phase relate to pan-Arctic conditions while the NAO changes are more focused on Greenland and the North Atlantic.

The relationship between water vapor isotopes and NAO phase across interannual timeframes at Thule is uncertain from our data's limited period of record, but climatology and ice core studies have argued for clear impacts on Greenland's precipitation, glacial mass balance, and sea ice at this scale (Stern and Heide-Jørgensen, 2003; Vinther et al., 2003; Sodemann et al., 2008; Bjørk et al., 2018). NAO indices in summer 2018 were almost continuously positive, coinciding with relatively cool conditions at Thule and below average melt of the northwestern Greenland Ice Sheet. In contrast, summer 2019 was dominated by NAO-

conditions and was nearly 4°C warmer than summer 2018 at Thule (USAF, 2019). Greenland as a whole recorded near-record ice sheet melt over the 2019 melt season, linked in part to extensive warm and sunny conditions over the ice sheet (Maslanik and Stroeve, 1999). Despite these markedly different conditions, mean $\delta^{18}O$ and $\delta^{2}H$ values are practically identical between the two summers (<0.01‰ and <1‰ different, respectively) and mean *dxs* values differ only by 1‰. This suggests that changes in synoptic flow patterns have little impact when local seas are free of sea ice, as isotopic exchange with local marine moisture

can buffer any potential isotopic changes from longer-distance synoptic transport shifts.

However, spring 2018 was also characterized by NAO+ indices and colder conditions while spring 2019 had lower NAO indices and warmer weather. In contrast to the summer observations, $\delta^{18}O$ and $\delta^{2}H$ values are 4.4‰ and 34‰ higher, respectively, in spring 2019 than spring 2018, and *dxs* values are 1.7‰ lower. As sea ice is still largely intact through spring near Thule, the limited potential for local water input and exchange at this time appears to allow an NAO signal to be expressed

in the mean isotopic values. Additional seasons and years of observation with different NAO conditions will help clarify the magnitude and seasonal extent of interannual effects from the NAO on Thule water vapor isotopes.

## 4.4 Surface wind regimes

Between April and September, the surface winds at Thule alternate between east-southeasterly katabatic winds off the nearby ice sheet and a west-northwesterly sea breeze from Bylot Sound, although not necessarily on a regular daily cycle. Such wind
shifts at other coastal sites are isotopically identifiable because katabatic winds supply isotopically light vapor from ice sheets while sea breezes supply isotopically heavy vapor from the nearby ocean (e.g., Kopec et al., 2014; Bréant et al., 2019). After adjusting for the seasonal biases at Thule where katabatic winds dominate in colder months and the sea breeze dominates in warmer months, a weak correlation between katabatic deviation (i.e., sea breeze occurrence) and the water vapor isotopes ($\rho_{SA.\delta18O.hr}$ = +0.19, $\rho_{SA.dxs.hr}$ = -0.18) indicates that on a given day the sea breeze will bring isotopically heavier water vapor
with lower $dxs$ values than katabatic flow.

Although the difference in mean isotopic values are statistically significant between the two wind regimes for all isotopic species (p <0.001, Welch's t-test), there is substantial overlap in the overall range (May–September data, mean ± 1 σ: $\delta^{18}O_{sea.breeze}$ = -26.8 ± 3.5‰ vs. $\delta^{18}O_{katabatic}$ = -28.5 ± 3.1‰; $\delta^{2}H_{sea.breeze}$ = -207 ± 25‰ vs. $\delta^{2}H_{katabatic}$ = -219 ± 22‰; $dxs_{sea.breeze}$ = 7.6 ± 4.8‰ vs. $dxs_{katabatic}$ = 9.3 ± 4.8‰; Fig. S3). The isotopic difference between the two wind regimes may be smaller at
Thule than other studied sites because of Thule's location on the west-pointing Pituffik Peninsula. Much of the southeasterly katabatic flow here is not sourcing air directly off the main Greenland Ice Sheet, but rather air that is traveling over Baffin Bay along the western coast and potentially over nearby De Dødes Fjord (Fig. 1). Thus, both the katabatic winds and the sea breeze are bringing water vapor from locations where it has been able to isotopically exchange with local ocean waters.

The slightly lower water vapor isotope values during katabatic flow are then likely due to the vapor's brief passage over
topographic highs of Cape York and the Tuto Ice Dome while the sea breeze carries moisture onshore unimpeded from Bylot Sound. Additionally, we note that as our wind data comes from a lower elevation site than our isotopic sampling, it is possible that particularly shallow sea breezes could produce wind observations identified as 'sea breeze' while the air at the isotopic sampling site on the South Mountain ridgetop is still katabatic sourced. However, no consistently clear signs of different air masses affecting the two stations in summer were found when examining periods with anomalous observations (e.g., sea breeze
azimuth with high air temperature).

## 4.5 Evaposublimation of local snow and surface waters

Evaporation and sublimation of local surface waters and snowpack (referred here as "evaposublimation") can also alter local water vapor isotopic values by supplying vapor that is isotopically lighter with higher $dxs$ values relative to the source snow/ice/water's isotopic composition (Casado et al., 2018; Kopec et al., 2019). Most of the snow at Thule falls in autumn and
early winter (USAF, 2019) and surface waters in Thule generally match the isotopic composition of the winter snowpack

(Csank et al., 2019). Thus, evaposublimation during warmer months will generally serve to lower $\delta^{18}O$ and $\delta^2H$ values and raise *dxs* values of water vapor in the local environment, although these two processes may fractionate the water isotopes different, and the exact isotopic composition of supplied vapor will depend on the relative balance of evaporation to sublimation (Christner et al., 2017). Additionally, very warm events in summer induce surface melt and sublimation across the Tuto Ice Dome and Greenland Ice Sheet (Box and Steffen, 2001; Nghiem et al., 2012; van As et al., 2012; Neff et al., 2014), and any of this moisture reaching Thule would be depleted in $^{18}O$ and $^2H$ and have a higher *dxs* relative to typical summer ocean-sourced moisture as it originates in a higher and colder location on the ice sheet. Isotopic effects from plant transpiration (Gat and Matsui, 1991; Farquhar et al., 2007; Aemisegger et al., 2014) are assumed to be very weak and inconsequential due to the sparse plant cover at Thule (Gold and Bliss, 1995).

## 4.6 Summary of environmental drivers of water vapor isotopes at Thule

To summarize, observations at Thule when the water vapor $\delta^{18}O$ and $\delta^2H$ values are higher and *dxs* values are lower are linked to: a) warmer local air temperature, b) more local marine moisture source, c) synoptic pattern that favors more southerly flow (i.e., NAO-), d) sea breeze surface winds, and/or e) lower evaposublimation. Likewise, lower $\delta^{18}O$ and $\delta^2H$ values and higher *dxs* values are linked to: a) colder local air temperature, b) more distant marine moisture source, c) synoptic pattern that restricts southerly flow (i.e., NAO+), d) katabatic surface winds, and/or e) higher evaposublimation. These different factors operate on different temporal scales (Fig. 5a). For example, shifts in local versus distant moisture sourcing affect isotopic composition at all timescales, but evaposublimation is mostly important only on shorter timescales like diel cycles.

These five factors should not be seen as independent drivers of isotopic change. Rather, these factors are detectable facets expressed in our observational data of the diverse environmental changes controlled by variations in Baffin Bay sea ice extent and synoptic moisture transport to the Arctic. As such, these factors greatly interact with each other, and isotopic variability cannot and should not be reduced down to a single predominant driver. However, these factors largely interact constructively to enhance relationships between environmental conditions and water vapor isotopic values (Fig. 5b, c). For example, the NAO- enhances southerly advection to Thule, which results in warmer local air temperature and potentially reduced sea ice extent that exposes local waters. Similarly, the sea breeze is enhanced during warmer periods of the year when sea ice extent is low, and it directly supplies local moisture to Thule from Bylot Sound. All these factors support isotopically heavier water vapor isotopes with lower *dxs* values when sea ice extent is low and synoptic conditions favor southerly flow.

The main exception to these constructive interactions is evaposublimation, which supplies isotopically light vapor with high *dxs* during warmer periods which would otherwise favor higher $\delta^{18}O$ and $\delta^2H$ values and lower *dxs* values as previously described. Due to the ample supply of marine moisture in the coastal Thule setting, evaposublimation is not expected to be a predominant moisture source or primary driver of isotopic variability. However, periods where water vapor isotopes are lighter

and *dxs* values higher than expected for a given temperature may be a good identifier of an evaposublimation effect, and such analysis could be used to better quantify ice sheet vapor flux during large surface melt events.

## 5. Seasonal changes in the drivers of isotopic variability

The strong annual cycle in water vapor isotopes at Thule (Table 1) is directly tied to seasonal changes in weather and the regional environment driven by the presence or absence of local sea ice. While the annual sea ice breakup directly influences vapor isotopic composition by allowing local marine moisture supply, it also produces a cascade of other environmental changes that isotopically alter water vapor at Thule (Fig. 5). Beyond simply affecting the mean values of water vapor isotopes, the dramatic environmental transformations that occur at Thule after sea ice breakup also modify correlative relationships between isotopic and meteorological variables (Fig. 6). As a result of all these factors, the seasonal growth and breakup of the Baffin Bay sea ice is readily identifiable in our water vapor isotope record, particularly in the *dxs* data (Fig. 2).

### 5.1 Spring and sea ice breakup

In both spring seasons covered by our record, the isotopic and meteorological variables had a very abrupt shift from winter-typical values to summer-typical values (Fig. 2). During these shifts that began on 14 May 2018 and 29 Apr 2019, temperatures rose more than 15°C in a few hours, followed by a ~10‰ increase in $\delta^{18}O$ and ~10–15‰ drop in *dxs* over the next few days. Sea ice concentrations and satellite imagery from NSIDC (Fetterer et al., 2017) and MODIS (Hall and Riggs, 2015) show that these abrupt spring shifts were associated with the breakup of sea ice near and to the northwest of Thule as well as a general reduction in sea ice concentration throughout Baffin Bay (Fig. S4). Similar isotopic responses to sea ice breakup have been previously reported in the Arctic Ocean as well (Klein and Welker, 2016). Before the breakups in 2018 and 2019, the NAO index dropped 2 and 4 points, respectively, and the resulting extreme isotopic shifts create very strong correlations in May between isotopic variables and both Baffin Bay sea ice extent (Fig. 6e) and the NAO index (Fig. 6f).

Sea ice breakup and environmental warming is followed in short order by the first sustained sea breeze developments of the year that also aid the delivery of newly available local vapor to Thule. At this time in late spring, radiative heat loss at night drives a semi-consistent diel wind cycle between afternoon sea breezes and nightly katabatic flow, and synoptic storm systems may bring multiple days when winds predominantly come from either the ocean or over the ice sheet, depending the relative position of the storm system to Thule. The environmental contrast between the relatively warm open ocean and still frigid Greenland Ice Sheet and tundra maximize isotopic differences in the two wind regimes, and, as a result, the strongest correlations between isotopic variables and katabatic deviation are observed in April through June (Fig. 6d).

## 5.2 Summer

The height of summer brings striking changes in isotope-climate relationships. Most notably, the $\delta^{18}O$ and $\delta^2H$ correlations
with air temperature switch from positive to negative in summer months (mean $\rho_{non\text{-}summer}$ = +0.38 vs. mean $\rho_{summer}$ = -0.32 for
$\delta^{18}O$) with *dxs* showing a similar, though opposite, pattern (mean $\rho_{non\text{-}summer}$ = -0.53 vs. mean $\rho_{summer}$ = +0.40) (Fig. 6a). As a
result, the highest $\delta^{18}O$ and $\delta^2H$ values and lowest *dxs* values are not observed when air temperatures are greatest, but rather
near 0°C (Fig. S3a, S3d) During the same summer period, isotopic correlations with mixing ratio greatly weaken (Fig. 6b)
while correlations with relative humidity strengthen (Fig. 6c). The relationship between air temperature and mixing ratio,
strongly correlated with a positive slope through most of the year, decouples in summer, and temperatures warmer than 5°C
largely result in no higher water vapor content (Fig. S5)

Summer in Thule is typically cool and humid under shallow marine air from local open seas, helped onto land by the sea
breeze. As the local ocean never warms much above freezing, most of the warmest periods at Thule (i.e., >5–8°C) require a
high pressure ridge over Greenland, often associated with southerly air flow up along the western Greenland coast and aided
by NAO- conditions. In many cases, the position of this high favors large-scale downsloping and subsidence off the
northwestern ice sheet (Fig. 7). This pattern was exceptionally persistent in summer 2019 (Tedesco and Fettweis, 2019), and
this period supplied most of the warmest observations in our database. Moisture arriving at Thule through southerly advection
and/or downsloping in these warm events is isotopically lighter and less humid than local marine air as it is more distantly
sourced and also must cross topographic highs. When contrasted against cool local marine air that has isotopically heavier
vapor with low *dxs* values, this produces the observed flip in summer correlations between isotopic species and temperature
as well as the strengthened correlations with relative humidity.

Correlations between isotopes and water vapor mixing ratio greatly weaken in summer because mixing ratios do not clearly
differentiate between local marine and southerly advected air. Although the shallow marine air has a higher relative humidity,
it is also colder than the southerly advected air and thus has a lower maximum mixing ratio. In contrast, the relative humidity
of southerly advected air is lower, but its warmer temperature means that the actual water content of the air can be very similar
to that of local marine air. As a result, the relationship between temperature and mixing ratio decouples (Fig. S5), and isotopic
correlations with mixing ratio approach zero (Fig. 6b).

Two additional processes that emerge in summer may also lead to these relationship changes. First, the very warm conditions
that occur during strong ridging over Greenland and enhanced southerly advection to Thule are also associated with intense
surface melt and sublimation on the Tuto Ice Dome and greater Greenland Ice Sheet (Nghiem et al., 2012; van As et al., 2012;
Neff et al., 2014; McLeod and Mote, 2016; Ballinger et al., 2019). As the ice sheet surfaces are higher in elevation and colder
than Thule, water vapor coming from their snow and glacial ice would be particularly isotopically light with high *dxs* values
(Steen-Larsen et al., 2013; Kopec et al., 2014; Bréant et al., 2019), and this would enhance the isotopic signature of southerly
advected air as it passes over the ice sheet on its way to Thule.

Second, the high pressure of these warm events often brings sunny weather to Thule that promotes intense heating of the local tundra and exposed rocky surfaces. Outside of polar night, water vapor mixing ratio typically follows the daily temperature cycle and peaks in the early afternoon. However, on days when the mean temperature is greater than 10°C, the mixing ratio drops as temperature rises and hits its minimum daily value around local noon. This loss of water vapor suggests that daytime heating on the warmest days induces convection (Duynkerke and van den Broeke, 1994) that vertically mixes drier air aloft

down to the shallow marine surface layer. This results in a net upward water vapor transport (Sherwood et al., 2010; Kiemle et al., 2013; Homeyer et al., 2014), and the drier air that mixes downward from the free troposphere brings vapor with lower $\delta^{18}O$ and $\delta^2H$ and higher $dxs$ values (Bailey et al., 2013). The surface heating and drying during midday could also increase local evaporation from the land and surface waters that, while not enough to overcome the net upward vapor transport from convection, would be an additional source of isotopically light water vapor to near surface moisture.

**5.4 Autumn and sea ice growth**

The transition to winter-type isotopic and meteorological conditions is much more gradual than the spring transition at sea ice breakup (Fig. 2), in accordance with the comparatively steady growth of sea ice in autumn. The $\delta^{18}O$ and $\delta^2H$ values slowly decrease while $dxs$ values slowly increase from October through November, and values through winter do not generally overlap summer values. While isotopic correlations with sea ice extent are near zero throughout summer (likely because most changes

to sea ice extent at this time are occurring too far north or west to directly impact Thule), the autumnal growth of sea ice results in moderately strong correlations in October (Fig. 6e) as the region of sea ice formation again affects Thule. Although small in magnitude, correlations with the NAO index also strengthen in autumn (Fig. 6f), and this is likely due to the close relationship between NAO phase, sea ice extent, and local marine moisture availability. The return of true day–night cycles in September creates a surface wind regime similar to that of spring, and correlations between isotopes and katabatic deviation

strengthen in the autumn months until the sea breeze ceases due to polar night in November (Fig. 6d).

**5.5 Winter**

During the polar night of winter, the water vapor isotopic system at Thule is relatively simple because northern Baffin Bay and all surface waters are frozen over. Mean values for $\delta^{18}O$ and $\delta^2H$ in winter are lower and $dxs$ values higher than other seasons in the absence of a local marine moisture supply (Fig. 3b), and isotopic variability largely reflects extratropical cyclone

impacts and broad synoptic flow changes. Strong winter correlations between isotopic species and both air temperature and mixing ratio (Fig. 6) reflect this as warmer and moister air transported to Thule produces higher water vapor $\delta^{18}O$ and $\delta^2H$ values and lower $dxs$ values.

The $dxs$ values in winter are also notably more variable than during summer (Fig. 2). Since nearly all moisture arriving to Thule in summer has to pass over the open water of Baffin Bay, vapor exchange and uptake probably helps stabilize the

summer $dxs$ variability. In contrast, when Baffin Bay is frozen, the $dxs$ values from different moisture sources may be preserved

better and contribute to the higher winter variability in *dxs* values. More focused back trajectory analysis on winter *dxs* variability in the future may help quantify the variability and effects of different moisture sources.

Interestingly, $\delta^{18}O$ and $\delta^2H$ have very strong positive correlations with the late winter sea ice extent in February and March, while *dxs* has very strong negative correlations with sea ice extent in February (Fig. 6e). These relationships are opposite to
those observed in spring and autumn, when the presence or absence of sea ice plays a clear isotopic role through local water availability. It is not entirely clear what is driving the strong correlations in late winter, as sea ice during these months is near its maximum extent and largely complete in coverage near Thule. These unusual correlations may be due to coincidental extreme events. Shortly after the maximal extent of sea ice was reached in 2018, a period of enhanced southerly moisture advection supplied abnormally heavy water vapor isotopes with very low *dxs* values. In 2019, the isotopes also become
unusually heavy with low *dxs* for two weeks after the peak in sea ice extent, but no associated increase in southerly moisture was identified and the root cause is unclear. With only two years of record, it is difficult to conclude whether the observed correlative strengths in late winter accurately represent a true change in isotopic character after a certain sea ice threshold is exceeded, or if it is simply due to coincidental occurrence with two unusual late winter weather events. Additional years of observation may help clarify this uncertainty.

**5.6 Cold season moisture pulse events**

During winter months, rapid shifts to extremely low NAO indices often coincide with intense poleward transport of southerly heat and moisture to Thule that lasts 1–5 days. We refer to these distinct episodes as "moisture pulse events" as they appear very clearly in the Thule mixing ratio time series (Fig. 2). During these events, air temperature rises 10–15°C and peaks near or above freezing while water vapor concentrations can reach four times greater than mean winter levels. These pulses appear
to notably reduce or slow the growth of sea ice across Baffin Bay (Fig. 2j), although the sea ice stays largely intact in the northern reaches near Thule. Although not all these events meet the defined criteria of an atmospheric river (Mattingly et al., 2018), all have similar impacts to an atmospheric river event due to their anomalously high moisture and heat transport.

Unsurprisingly, such extreme weather changes are reflected by impressive water vapor isotopic responses. During these moisture pulse events, the $\delta^{18}O$ and $\delta^2H$ rise 6–10‰ and 50–100‰, respectively, to reach values more typical of late spring
and early summer. Concurrently, the *dxs* drops 15–25‰, and the minimum *dxs* values observed in these moisture pulse events match or exceed the minimum values observed at the height of summer. As sea ice coverage prevents the uptake of isotopically heavy moisture with low *dxs* values from local waters during these moisture pulse events, the anomalous isotopic values must signify the presence of deep southerly moisture transport (Bonne et al., 2015). A focused analysis of these events is currently underway.

The rapid isotopic and environmental shifts coinciding with sea ice break up in 2018 and 2019 also fit the general pattern of a moisture pulse event, except the isotopic and meteorological variables do not revert back to their preceding values after 1–5

days. Indeed, the shift in 2019 is associated with an atmospheric river event impacting western Greenland and an extreme drop in the NAO index from +2 to -2 while the 2018 shift coincided with an NAO index drop from +2.63 (the highest observed in our record) to +0.40 and an intense pulse of southerly moisture advection. In typical cold season moisture pulses, sea ice remains intact around Thule and the isotopic and meteorological effects from the southerly moisture advection are short lived as the moisture pulse air mass moves on or mixes out. In these two spring events, however, local sea ice coverage is broken, and the new ample supply of local water vapor allows isotopic and meteorological values to remain elevated.

## 6. Diel cycles

### 6.1 Diel cycles overview

Most polar sites report diel cycling of water vapor isotopes in summer (Steen-Larsen et al., 2013; Bastrikov et al., 2014; Bonne et al., 2014; Kopec et al., 2014; Casado et al., 2016; Ritter et al., 2016; Bréant et al., 2019), but this cycling is attributed to different causes including katabatic wind cycling (Kopec et al., 2014; Bréant et al., 2019), vapor exchange between snow and air (Steen-Larsen et al., 2013; Casado et al., 2016), and dew formation (Bastrikov et al., 2014). In contrast to these sites, Thule exhibits relatively limited isotopic diel cycling in all months except March, with total cycle magnitudes less than 1‰ for both $\delta^{18}O$ and *dxs*, even when observations with possible cyclonic system impacts (i.e., lower than average station pressure) are excluded. Additionally, the existence of these small diel cycles is only clearly evident when the larger synoptic variability is removed by averaging multiple days together. This subdued diel cycling is probably due to Thule's moderated coastal climate (Bréant et al., 2019; Bonne et al., 2020) while lengthy periods of midnight sun/polar night also reduce the day-night contrasts that power diel cycles at lower latitude sites.

Looking broadly at the entire Thule isotopic record, the magnitude of isotopic diel cycles is not a significant driver of isotopic variability. However, we believe that analyzing these cycles gives better insight into the broader environmental controls on water vapor isotopes, including those at synoptic and annual scales. Based on observed isotopic and meteorological patterns, we divide diel cycles at Thule into four regimes: polar night, March, summer, and transition (Fig. 8). These regimes are superimposed on the broader seasonal changes in isotopic variability and control previously discussed. Out of the major drivers of isotopic change at Thule, air temperature, surface wind regime, and evaposublimation vary enough on hourly timescales to contribute to diel isotopic cycles, while the NAO and local marine moisture availability are expressed at longer timeframes and/or not on a consistent daily cycle. We note that the magnitude of daily change in the isotopic variables is typically within the confidence intervals of the mean with the exception of the March regime. We believe the daily patterns of variable change are still informative, but discussion of these trends should be viewed as more speculative than other conclusions made in this study.

## 6.2 Polar night regime

The polar night regime includes the months from October through February, when daylight is very short (October, February) or nonexistent (November–January). With the lack of a strong insolation cycle, meteorological variables change little over a day, and winds are strong and consistently katabatic. Without any diel changes in the factors that drive isotopic variability (i.e., air temperature, surface wind regime, evaposublimation), the water vapor isotopes show no diel cycling in the polar night regime (Fig. 8). A very slight hint of a rise or fall in values can be seen around local noon, but this is primarily attributed to data from the end of February that resemble the upcoming March regime.

## 6.3 March regime

The observed isotopic cycles in March (maximum diel range: $\delta^{18}O$: 2.4‰, $\delta^2H$: 16‰, $dxs$: 5.1‰) are by far the largest magnitude of any month of the year at Thule (Fig. 8), although similar cycles with smaller amplitudes are also observed at the end of February. This heightened isotopic response appears due to several coinciding factors: a maximized daily insolation cycle near the equinox, an extensive snowpack to supply water vapor, and the enhanced impact of snow-supplied vapor in the very cold and dry environment (aided by the maximal seasonal extent of sea ice that limits external moisture input). This environment approximates the summer ice sheet setting of two other water vapor isotope observation sites at NEEM, Greenland (Steen-Larsen et al., 2013), and Dome C, Antarctica (Casado et al., 2016), and Thule's March isotopic diel regime appears similar to the cycles reported from those sites.

Moisture at Thule in March predominantly arrives from distant transport through katabatic flow with resulting low $\delta^{18}O$ and $\delta^2H$ values and high $dxs$ values. However, daytime heating of the snow surface across the landscape promotes release of water vapor held between snow grains that have equilibrated with the local snow isotopes (Steen-Larsen et al., 2013; Casado et al., 2016). Because most snow in Thule falls in late autumn/early winter and is sourced from Baffin Bay, the equilibrated vapor is isotopically heavier and has a lower $dxs$ than the katabatic-supplied moisture, producing the observed $\delta^{18}O$ and $\delta^2H$ peak and $dxs$ minimum around local noon. As insolation and air temperature drop in the evening, this vapor release ceases and potentially reverses, allowing the vapor isotopic composition to revert back to one of katabatic origin with lower $\delta^{18}O$ and $\delta^2H$ and higher $dxs$ values. The lack of a large drop in relative humidity coinciding with the midday temperature rise suggests that additional vapor is being supplied, and the local snowpack is the only likely source with all local waters frozen. A large isotopic diel cycle predominating in spring has also been reported in the Lena River delta and similarly attributed to the release of water vapor from the preceding winter snowpack (Bonne et al., 2020).

A parallel isotopic regime does not reappear around the autumnal equinox at Thule or at the Lena Delta site (Bonne et al., 2020). This is likely due to little to no extensive snowpack and a much higher water vapor content from warmer temperatures and open nearby seas that buffer any minor potential input from a snowpack vapor exchange. One exception to this lack of autumnal diel cycles, unique in the Thule record, occurred from 20–25 September 2017 when an isotopic diel cycle very

similar to the March regime appeared (Fig. S6). This cycling occurred shortly after a snowfall and coincided with a diel surface wind cycle where the katabatic winds calmed each afternoon but did not switch to a sea breeze. This appears to have allowed vapor released from the recent snow to raise isotopic values in the afternoon before the returning katabatic winds mixed the snow-derived moisture out and dropped isotopic values in the evening. In other September periods with a similar wind regime but without a recent snow cover, no isotopic cycling was recorded.

## 6.4 Summer regime

In the summer months (June–August), the midnight sun reduces the diel insolation cycle. Combined with the moderating influence of fully ice free local seas, most meteorological variables are more stationary over the course of a day. With this reduced environmental variability, diel cycles in isotopes are subdued. The subtle daily patterns in $\delta^{18}$O and *dxs* that do exist appear largely attributable to evaposublimation cycles: daily warming supplies low $\delta^{18}$O/high *dxs* water vapor through increased evaposublimation, with the resulting $\delta^{18}$O minimum and *dxs* maximum around midday (Fig. 8). Increased boundary layer mixing at midday may also supply moisture from the free troposphere with low $\delta^{18}$O and high *dxs* values to produce a similar isotopic effect to evaposublimation.

At Kangerlussuaq (Kopec et al., 2014) and Dumont d'Urville (Bréant et al., 2019), the local surface winds have a clear diel cycle between daily sea breezes and nightly katabatic winds in summer. At Thule, the sea breeze has a diel cycle where it strengthens in the afternoon and weakens at night (Fig. 8g), but there is not a full switch back to katabatic flow at night in summer. This dominance of the sea breeze at Thule is likely due to 24 hour summer insolation that can fuel a sea breeze even at "night", while Kangerlussuaq and Dumont d'Urville still have true night at their lower latitudes. As a result, sea breeze-supplied local water is likely present throughout a typical day and night at the height of summer in Thule, and the nightly weakening of the sea breeze has limited isotopic effect without a full shift to katabatic flow.

## 6.5 Transition regime

The seasonal transition months of April, May, and September have a true day–night cycle that gives meteorological variables a greater diel amplitude than the summer or polar night regimes (Fig. 8). Yet unlike March, these transitional months are much warmer and more humid with moisture supplied from an open or opening Baffin Bay. Winds exhibit a strong diel cycle as surface heating leads to afternoon sea breeze development (Fig. 8g), and radiative heat loss over the ice sheet at night strengthens katabatic flow. We might expect a clear diel response in water vapor isotopes due to these heightened diel meteorological cycles, but the isotopes have little to no daily cycle (Fig. 8a–b).

This limited isotopic cycling is likely due to competing effects of diel surface wind regime and evaposublimation cycles. While the sea breeze-sourced vapor coming directly off Bylot Sound in afternoon is isotopically heavy, midday heating also supplies isotopically light moisture from evaposublimation and boundary level mixing. Cooling at night brings a wind shift to

isotopically lighter katabatic flow but also suppresses evaposublimation and boundary level mixing while promoting dew condensation. As a result, no clear diel isotopic cycle emerges, although the subtle late evening peak in $\delta^{18}O$ may arise because evaposublimation rates are reduced while the sea breeze is still ongoing.

## 7. Implications

The improved understanding of water vapor isotopic variability granted by our Thule observations can aid the interpretation of regional ice cores. The primary drivers of isotopic changes in water vapor are likely to also affect local precipitation, although there are many additional processes involved in the transfer of an isotopic signature from vapor to precipitation and eventually to ice core that must be considered (Steen-Larsen et al., 2014; Casado et al., 2018; Madsen et al., 2019). Changes in $\delta^{18}O$ and $\delta^2H$ have long been used to reconstruct climate change from deep ice cores, typically interpreted as local temperature variability in Greenland (e.g., Dansgaard et al., 1969; Grootes and Stuiver, 1997; Johnsen et al., 2001). While air temperature is strongly correlated with isotopic change at Thule, much of this strength appears to arise as a common response to seasonal change between air temperature and sea ice extent. Air temperature might still be robustly reconstructed from isotopic archives based on our observed strong correlations, but the lack of a causative relationship risks misinterpretation if applied back far through time when the local environment may have significantly changed. Of particular warning is our observation that the basic relationship between air temperature and water vapor isotopes inverts in summer, leading to very different isotopic interpretations depending on season. While this may be a local and/or coastal effect that is not expressed on ice sheets (Ballinger et al., 2019), we still advise caution.

More recently, additional consideration has been given in ice core analysis to secondary isotopic variables like *dxs* and other environmental drivers such as moisture source, sea ice extent, and atmospheric circulation (e.g., Grumet et al., 2001; Vinther et al., 2003; Steffensen et al., 2008; Landais et al., 2018; Kopec et al., 2019). While our two year Thule record is too short to statistically determine the strongest drivers of interannual isotopic variability, changes in the duration of sea ice coverage and mean NAO phase appear most likely to control year-to-year differences in mean isotopic composition. In line with recent interpretations in regional ice cores (Osterberg et al., 2015), our results suggest that past periods with decreased sea ice extent in Baffin Bay will have increased local marine moisture sourcing that produces higher $\delta^{18}O$ and $\delta^2H$ and lower *dxs* values.

In recent years, Baffin Bay has had later sea ice freezes, earlier breakups, and a decrease in overall sea ice extent that accounts for 22% of all recently observed loss in March Arctic sea ice (Onarheim et al., 2018). We expect that the resulting changes in the regional hydroclimate will be isotopically preserved in future glacial ice in Greenland. However, predicting the isotopic responses to these changes is challenging. The future of the Nares ice bridge between Greenland and Ellesmere Island is one potential complication: should it consistently fail to form in a warmer world, the North Water Polynya may see increased sea ice cover that slows the overall sea ice loss impact on Thule (Barber et al., 2001; Puntsag et al., 2016; Vincent, 2019). Reductions in Arctic sea ice coverage and duration may also shift precipitation seasonality (Kopec et al., 2016) and atmospheric

circulation (McLeod and Mote, 2016; Ballinger et al., 2018; Francis et al., 2018), changing the annual isotopic balance of precipitation. More open winter waters in Baffin Bay could decrease annual $\delta^{18}O$ and $\delta^2H$ values as the relative fraction of isotopically lighter winter precipitation events increases, but this may be counter-balanced in part by the greater sourcing from isotopically heavy local waters.

Enhanced southerly moisture advection aided by NAO- conditions would also favor higher $\delta^{18}O$ and $\delta^2H$ and lower *dxs* values in northwestern Greenland, though some caution might be warranted for isotopic complications from greater ice sheet surface melt and vertical atmospheric mixing similar to those we have observed during summer NAO- phases. Additionally, the positive relationship between NAO phase and Baffin Bay sea ice extent (Mysak et al., 1996; Grumet et al., 2001; Stern and Heide-Jørgensen, 2003) makes it difficult to quantifiably split their influences on regional water isotopes. The extreme isotopic values we observe during moisture pulse events suggest that any changes in their frequency and magnitude (and more generally of related atmospheric river events) will have an outsized effect due to the sheer volume of moisture and precipitation they can bring to Greenland. For highly-resolved ice cores, it may be possible to identify particularly strong moisture pulse events or years with high frequency of these events as extreme minima in *dxs* values. Taken as a whole, it is clear that in the absence of additional clarifying evidence from other ice core proxies or nearby records, isotopic interpretations should be cautious in assigning cause solely to one individual environmental factor without further analysis to tease the many highly-integrated potential factors apart.

## 8. Conclusions

Our two years of water vapor isotope monitoring in northwest Greenland have produced a record of unprecedented extent and resolution for High Arctic Greenland and one of the longest records of any polar site reported to date. The variability in water vapor isotopes at Thule is explained by five interrelated environmental mechanisms: local air temperature, local marine moisture availability, the NAO, surface wind regime, and evaposublimation. The relative importance of each mechanism changes over the course of the year and overall produces a clear annual isotopic cycle that is closely linked to sea ice extent in Baffin Bay and synoptic polar moisture transport. On top of these seasonal trends, local environmental drivers such as sea breeze development, vapor supply from surface water and snow, and convection can subtly modify near surface water vapor isotopes through diel cycles.

This analysis highlights the importance of local geography and climate in isotope systematics. Compared to other high latitude water vapor isotope studies, Thule is more substantially impacted by sea ice fluctuations, both seasonally and interannually. This results in a clearer identification of the isotopic effects of sea ice extent which is particularly valuable for the interpretation of deep ice cores. Additionally, some aspects of isotopic variability, such as the changes in correlative relationship strength and direction during spring and summer, are so far uniquely reported for Thule, and it is unclear at this time if these seasonal

relationship patterns are present at other polar sites. As a result, conclusions based on data at one high latitude site should not be applied indiscriminately to other sites without extensive validation that such a comparison is warranted.

Caution should also be made when making strong conclusions about isotopic controls based on observations of short duration (i.e., a single year or less). If data is only taken during a single season, any observed relationships may be specific only to that season and should not be assumed to be applicable to the entire year. For variables with relatively long-term variability, such as teleconnection indices, data from a single year is likely not long enough to reveal fully robust correlations. In our Thule data, we were fortunate that the synoptic patterns were different enough between 2018 and 2019 to allow useful analytical

comparisons of the NAO at monthly and seasonal resolutions. However, even two years of data are not enough to fully clarify the relationships between teleconnections and water vapor isotopes for all months, and additional years of isotopic data from continued observation are needed for an improved understanding.

Moving forward, this isotope and meteorological database offers many opportunities for more in-depth and focused analyses of specific weather events and atmospheric patterns. As previously stated, our identified moisture pulse events are a focus of

additional research, and we encourage the use of our data in other focused case studies. An expansion of our "quasi-climatology" approach could be used to model moisture source environments for direct comparative analysis with the water vapor isotope data, which can help resolve existing questions relating to the nature of the *dxs*–sea surface temperature relationship in polar regions (Vimeux et al., 1999; Aemisegger and Sjolte, 2018).

Water vapor isotope observations at Thule continue, and additional months and years of data will help refine and verify our

conclusions made here. The long temporal coverage of our database makes it an excellent option for validating high resolution isotope-enabled simulations of air-ice and air-sea interactions in the Arctic. These results from Thule will help greatly in interpreting isotopic variability at other sites in the Arctic Water Isotope Network (Welker et al., 2019), and future collaborative analysis across this network will allow tracking the effects of synoptic weather patterns on water vapor isotopes in real time across the circumpolar region.

**Data availability**

Isotopic data is available at https://doi.org/10.18739/A21J9779S. Meteorological data for SMT is available at https://doi.org/10.1594/PANGAEA.895059 and in associated followup datasets by G. Muscari also on PANGAEA. Meteorological data for THU is available upon request of the 821st Weather Squadron of the US Air Force.

**Author contribution**

JMW and ESK installed the L-2130i; PDA and BGK installed the dry air calibration system and performed maintenance on the L-2130i; PDA wrote the code in R for organizing and calibrating data and for statistical analyses; KSM performed back

trajectory and water vapor transport analyses; DC performed cross-correlation analyses; PDA prepared the manuscript with contributions from all co-authors.

## Competing interests

The authors declare that they have no conflict of interest.

## Acknowledgments

This project was funded by NSF Arctic Observing Network-ITEX 1504141 and Arctic Observing Network- EAGER MOSAiC 1852614 and supported in part by the inaugural UArctic Research Chairship to JMW. We greatly thank the assistance of the United States Air Force, the 821st Air Base Group at Thule Air Base, Vectrus, and Polar Field Services for logistical and
hosting support throughout this research. We thank Giovanni Muscari for graciously providing the SMT meteorological data. Special thanks go to the 821st weather squadron for meteorological data collection and to Shawn Arnett, Rich Biggins, Devin Brewer, Joe Burns, Matthew Burns, David Craig, Jarrod Dodgen, David Drainer, John Gaston, Missa Goldun, Pablo Londono, and Josh Neighbours for onsite assistance and maintenance. We also thank Matheiu Casado, Kazimierz Rozanski, and Hans Christian Steen-Larsen for their advice on isotopic calibration and interpretations, Tom Mote for discussions about Greenland
climatology, and our three manuscript reviewers for their many helpful comments and edits.

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

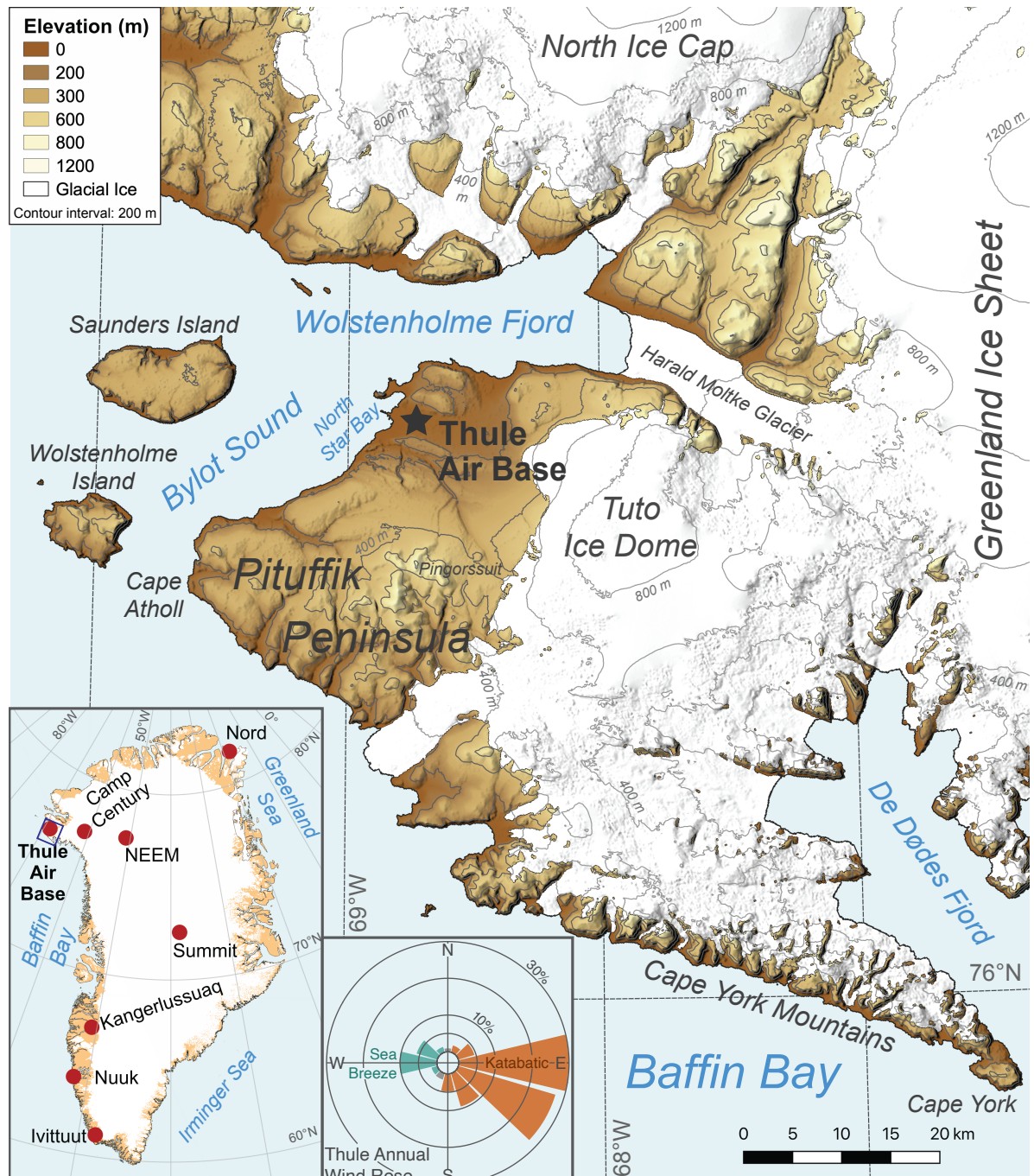

**Figure 1. Map of the local geography around Thule Air Base, the site of water vapor and meteorological measurements for this study. The inset map at lower left shows the location of Thule Air Base in Greenland and other important Greenland sites. Elevation data was taken from ArcticDEM, Polar Geospatial Center (Porter et al., 2018), and ice sheet, land, and ocean extent data were taken from Greenland Ice Sheet Mapping Project, NSIDC (Howat et al., 2014; Howat, 2017). Wind rose created from Thule airport data covering Aug 2017-Aug 2019 (USAF, 2019).**

120

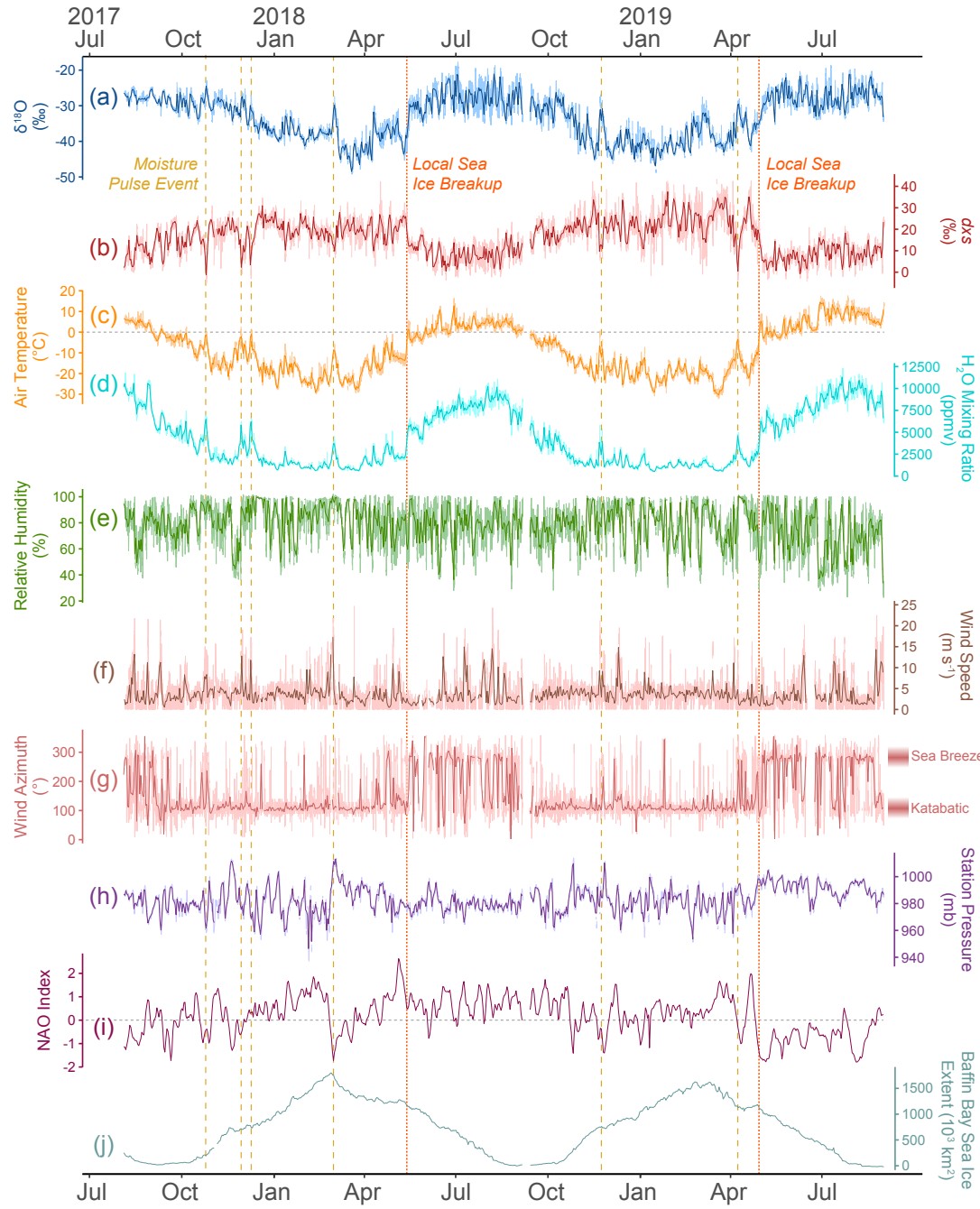

**Figure 2.** Observed data from Thule, Greenland, of (a) water vapor δ¹⁸O, (b) water vapor *dxs*, (c) mean air temperature, (d) water vapor mixing ratio, (e) relative humidity, (f) wind speed, (g) wind azimuth, (h) station pressure, (i) NAO index (NOAA, 2019), and (j) Baffin Bay sea ice extent (Fetterer et al., 2010). The time series of δ²H is very similar to δ¹⁸O and not shown. Data shown are daily mean values with hourly values as lighter backdrop line for higher resolution variables (a–h). All observations were taken at the

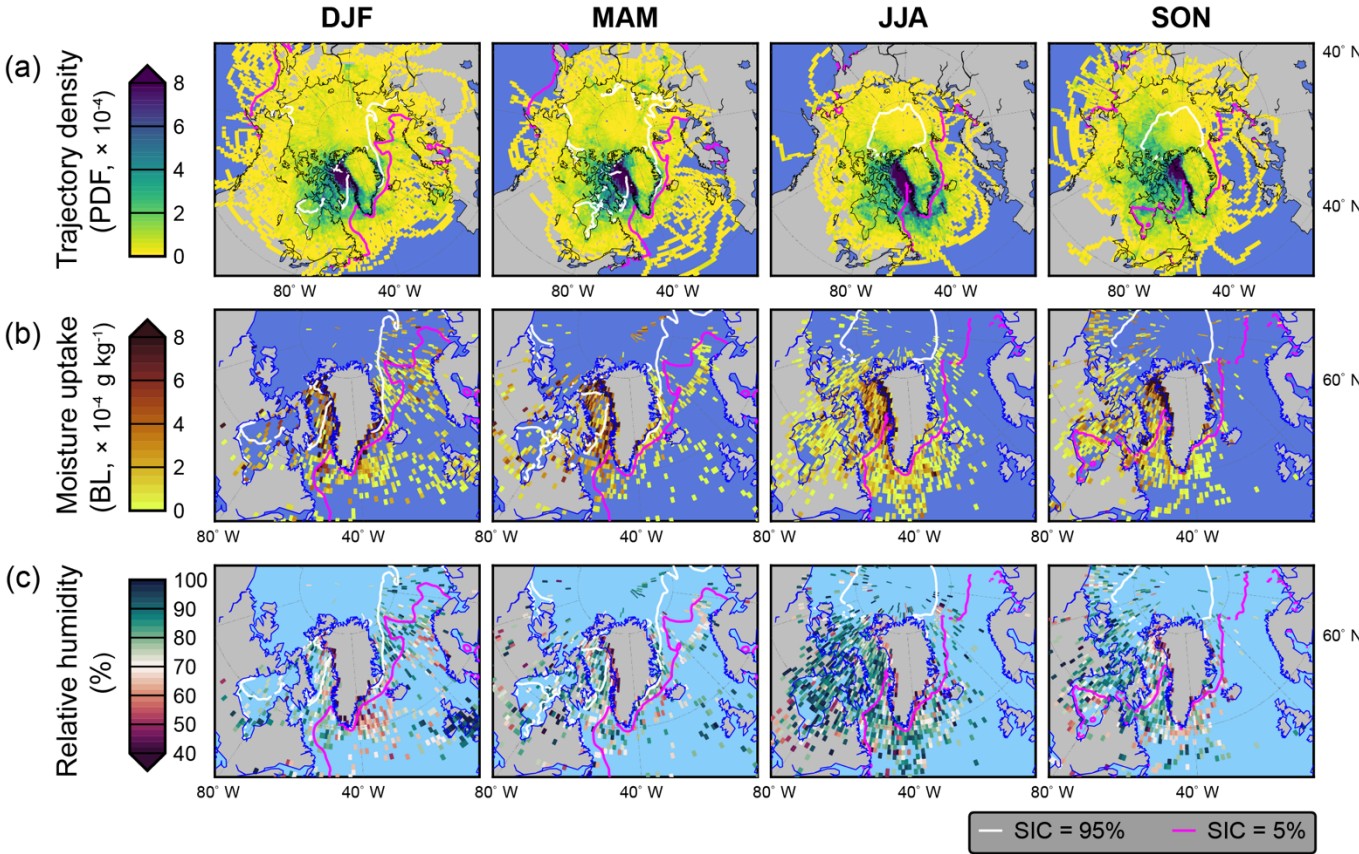

SMT site except for wind speed and azimuth which were observed at the THU airport. Yellow dashed vertical lines indicate moisture pulse events and orange dotted vertical lines indicate the timing of sea ice breakup near Thule.

130 **Figure 3. Back trajectory analysis results for air parcels arriving at Thule for each meteorological season, including (a) trajectory density, (b) regions of water vapor uptake from the boundary layer (BL), and (c) relative humidity for these regions of vapor uptake. Analysis based on a "quasi-climatology" that sampled 10 random days from each month during 1980–2018. Trajectory points are binned onto a 1° latitude/longitude grid as (a) probability density functions or (b, c) mean values within each grid cell. White and magenta lines show the extent of 95% and 5% sea ice concentration (SIC), respectively, for the given season.**

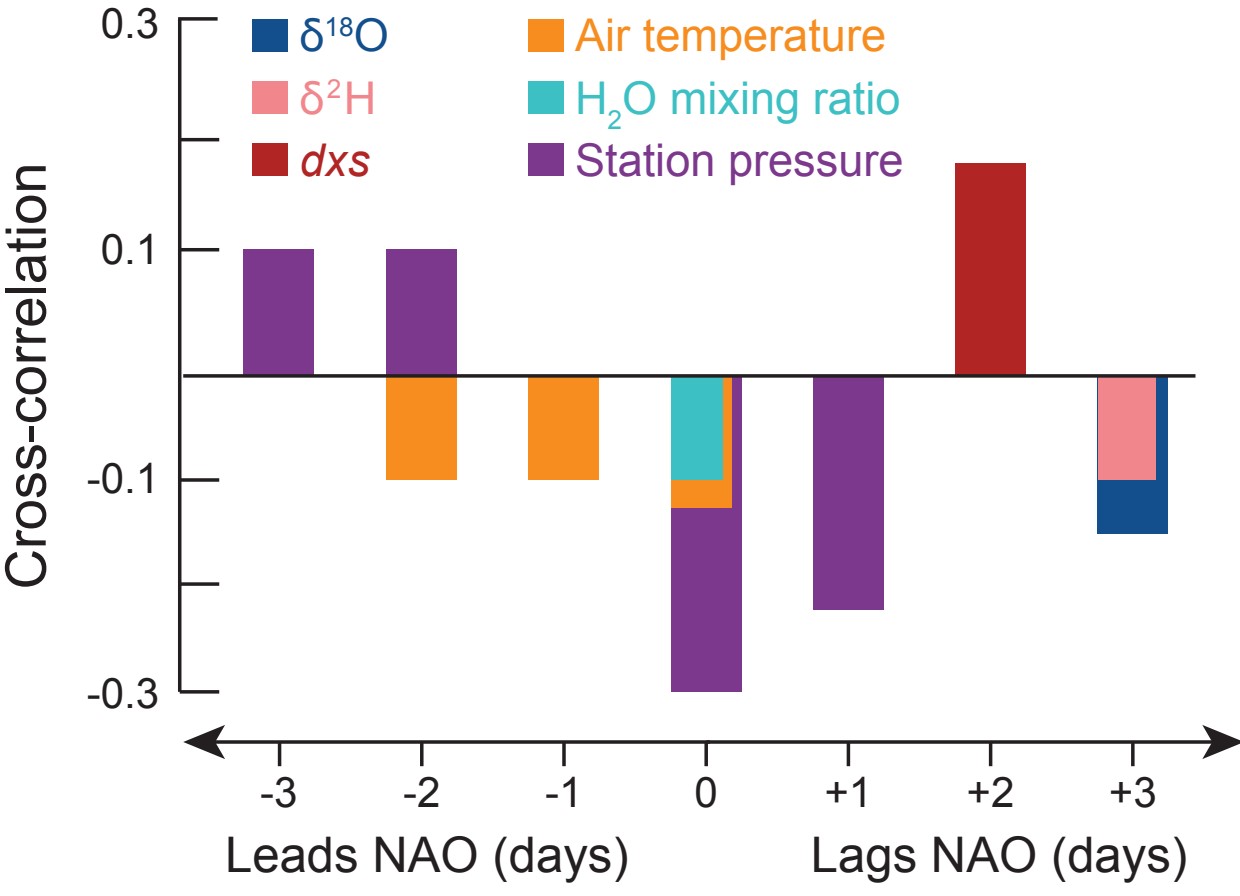

135

**Figure 4. Cross-correlation analysis results between NAO index and selected isotopic and meteorological variables. Results shown are all statistically significant at p < 0.05 (n = 737), and non-significant results are not illustrated.**

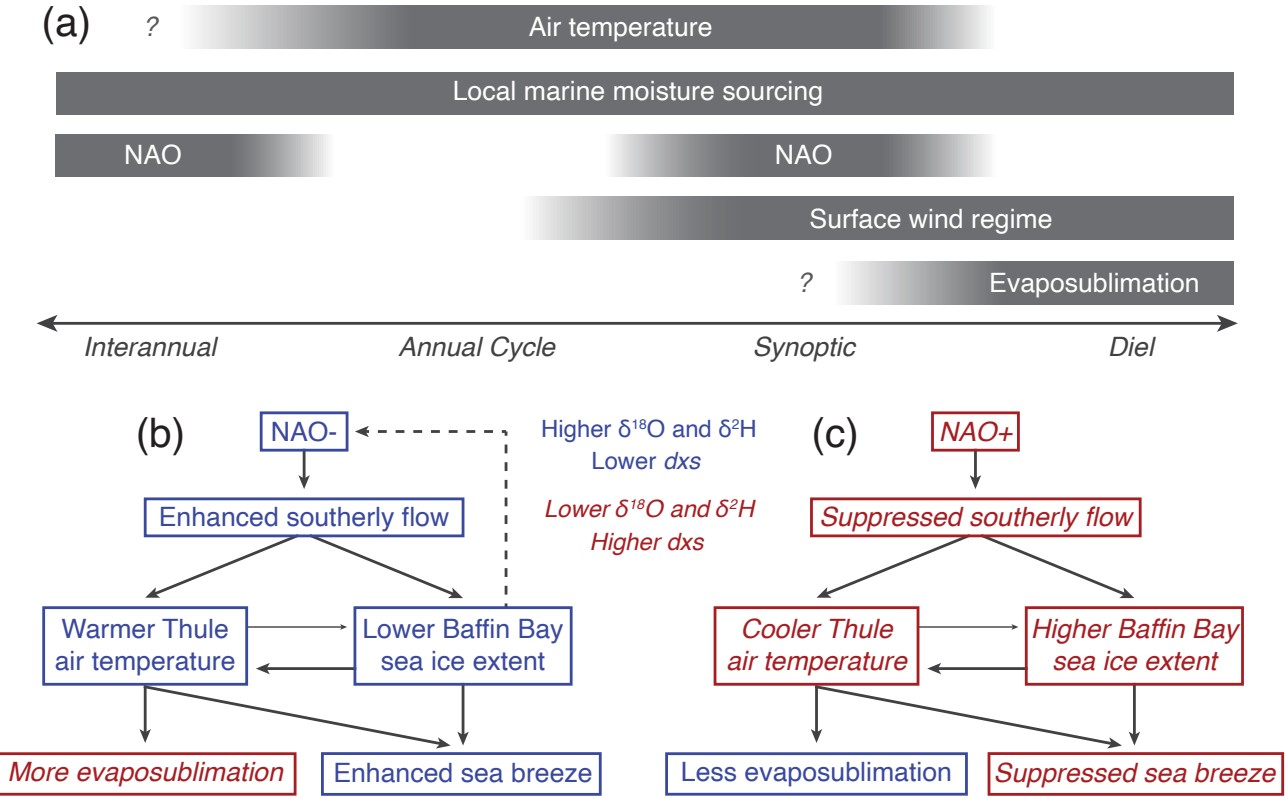

**Figure 5. Summary of effects of environmental factors affecting water vapor isotope composition at Thule. (a) Illustration of temporal scales when each factor is most impactful. Question marks indicate periods of uncertain degree of impact. (b) Interactions between environmental factors during NAO- conditions, with factors that produce higher $\delta^{18}O$ and $\delta^2H$ values and lower *dxs* values in blue, and factors that produce lower $\delta^{18}O$ and $\delta^2H$ values and higher *dxs* values in italicized red. The dashed line refers to the possibility that low sea ice extent reinforces the NAO- phase (e.g., Petrie et al., 2015). (c) Same as (b), but for NAO+ conditions.**

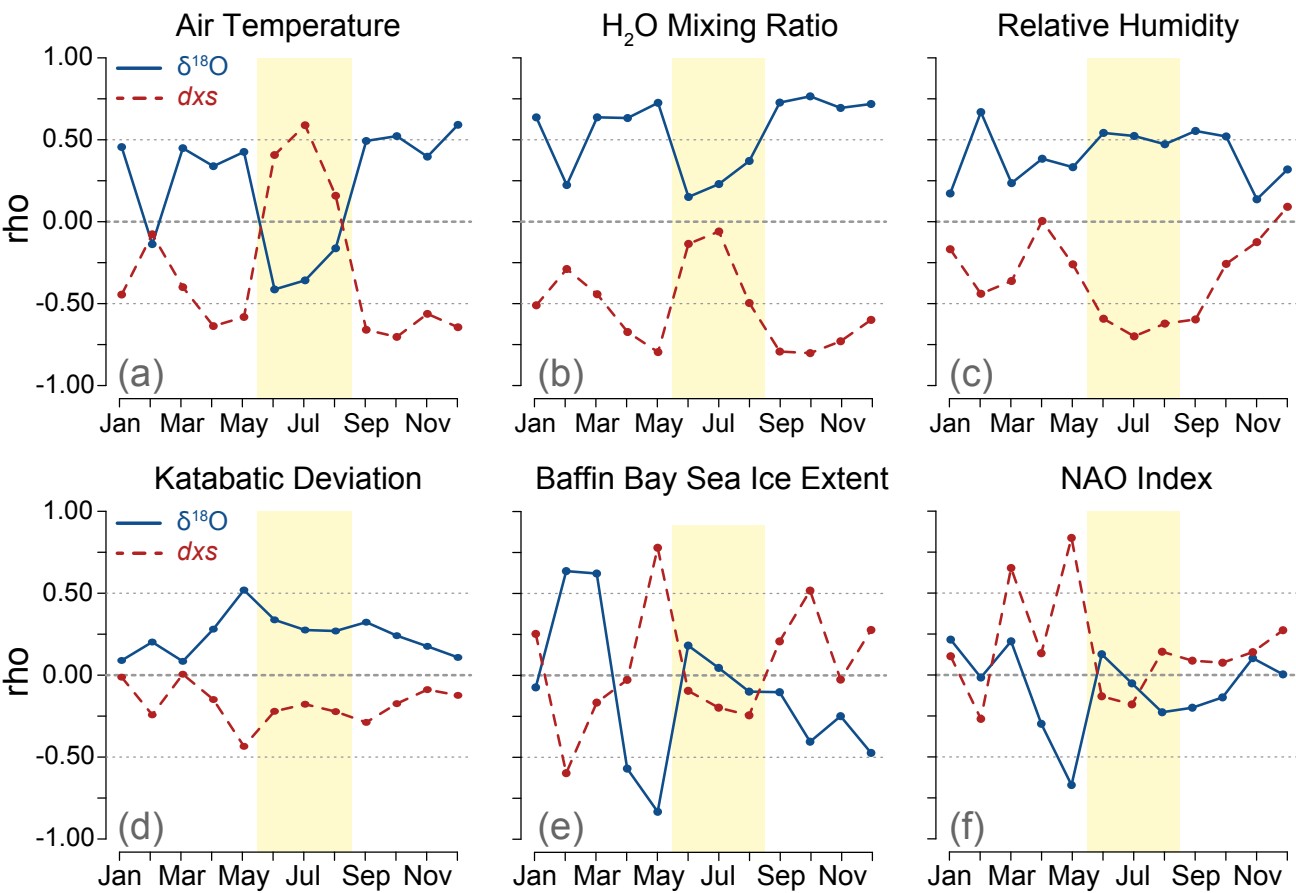

Figure 6. Spearman correlations of meteorological variables (a-f) with $\delta^{18}O$ (blue, solid) and *dxs* (red, dashed) with data binned by month. The yellow box highlights the summer season (JJA) and horizontal dotted lines show rho values of 0.00 and ±0.50 for reference. Correlation patterns for $\delta^2H$ (not shown) are very similar to those of $\delta^{18}O$. Correlation values were calculated using the highest available data resolution: 10 minute for (a–c), hourly for (d), and daily for (e–f).

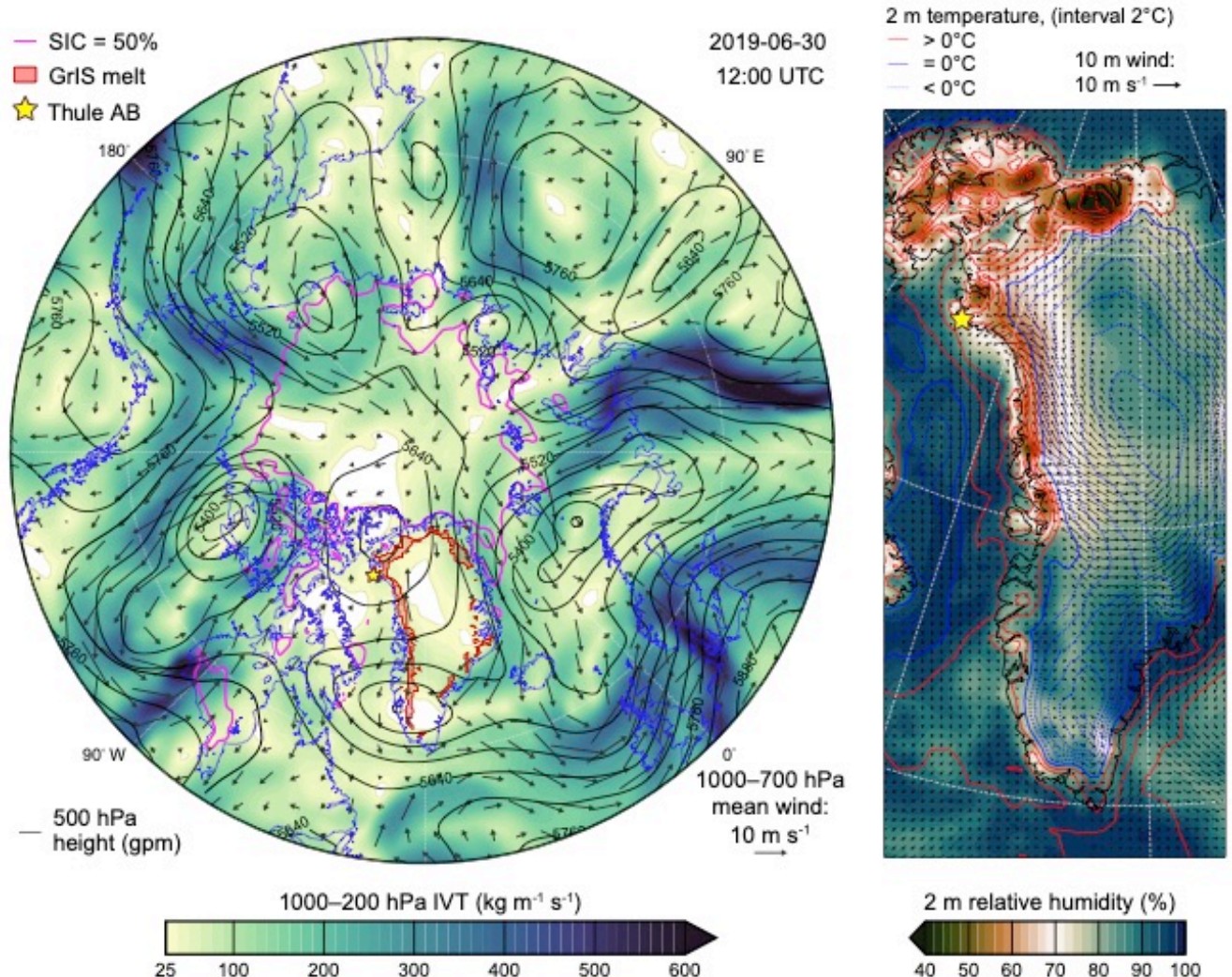

Figure 7. Synoptic pan-Arctic (left) and regional Greenland-focused (right) atmospheric conditions during an anomalously warm period on 30 June 2019, when a ridge over northwest Greenland directed downsloping air toward the observing site at Thule Air Base. Synoptic map shows integrated water vapor transport (IVT), 1000–700 hPa mean wind (arrows), and 500 hPa height (gray isolines), as well as the 50% sea ice concentration (SIC) boundary (pink line) and areas of Greenland Ice Sheet (GrIS) surface melt (red shaded zones). Areas in white have less than 25 kg m⁻¹ s⁻¹ IVT. Regional map on right shows 2-meter relative humidity (primary shading) and temperature (red and blue isolines) along with 10 meter wind (arrows). All data are from MERRA-2 (Stein et al., 2015; Gelaro et al., 2017) except for ice sheet surface melt, which is from the NSIDC MEaSUREs Greenland Surface Melt daily data set derived from passive microwave satellite observations (Mote, 2014).

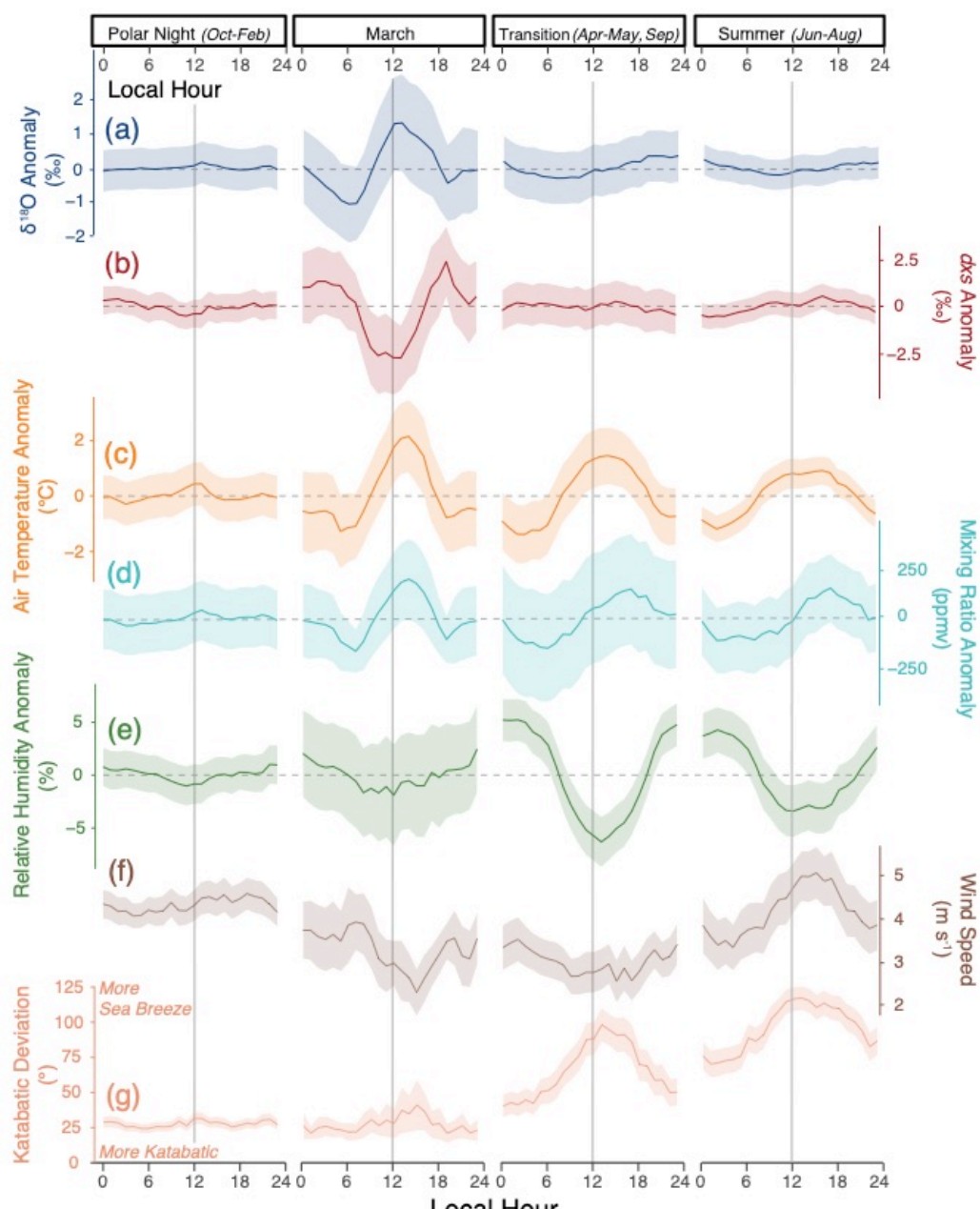

**Figure 8.** Diel patterns in isotopic and meteorological variables in Thule, Greenland, showing the mean hourly value (solid thick line) and 95% confidence intervals of mean estimates (colored shading) for each diel regime. Diel patterns for $\delta^2H$ are very similar to $\delta^{18}O$ (a) and not shown. Variables other than wind speed and katabatic deviation are expressed as deviations from the mean value in each regime, with zero deviation indicated as a dashed horizontal line. Note that because the winds in Thule are binary (katabatic or sea breeze), the diel means of katabatic deviations should be viewed as a probability of being katabatic or sea breeze at a given time and not that the winds are smoothly transitioning from east to west or vice versa. Additionally, because katabatic deviations increase in value both directions away from 100° (e.g., winds from 80° and 120° both have a katabatic deviation of +20°), the natural variance of purely katabatic winds produces an average value of ~25° rather than 0°.

**Table 1. Annual and seasonal mean ± 1 standard deviation values of selected water vapor isotopic and meteorological variables for Thule, Greenland, over the analytical period of this study. Values calculated using 10 minute resolution data.**

|  | $\delta^{18}O$ (‰) | $\delta^2H$ (‰) | *dxs* (‰) | Air temperature (°C) | $H_2O$ mixing ratio (ppmv) | Relative humidity (%) |
|---|---|---|---|---|---|---|
| Annual | -33.2 ± 6.1 | -249 ± 43 | +16.2 ± 8.1 | -8.1 ± 11.1 | 4180 ± 3000 | 78 ± 16 |
| Spring (MAM) | -35.9 ± 5.8 | -270 ± 40 | +17.7 ± 8.5 | -12.1 ± 9.2 | 2850 ± 1940 | 79 ± 16 |
| Summer (JJA) | -27.6 ± 3.5 | -212 ± 24 | +8.6 ± 4.9 | +5.4 ± 4.1 | 8310 ± 1290 | 73 ± 18 |
| Autumn (SON) | -31.3 ± 4.4 | -233 ± 31 | +17.7 ± 6.0 | -8.5 ± 6.4 | 3600 ± 1740 | 78 ± 13 |
| Winter (DJF) | -38.5 ± 3.6 | -287 ± 26 | +21.6 ± 6.4 | -18.9 ± 5.2 | 1550 ± 900 | 83 ± 16 |

**Table 2. Spearman correlation values between isotopic and meteorological variables at Thule. Two sets of correlations are reported here: analyses from the original data and analyses from the seasonally-adjusted data where the seasonal cycle was removed for isotopic, air temperature, mixing ratio, and sea ice data. Seasonally-adjusting the data weakens nearly all correlations by removing the common response to seasonal change, but strengthens isotopic correlations with relative humidity. Correlations are given for all available levels of temporal aggregation, but some variables do not have data at finer resolutions. Temporal aggregation slightly strengthens correlations, but does not generally change the order of variables when ranking by correlation strength. Due to the large sample sizes, all correlations are significant at p < 0.05 except for a few with very weak correlations (*italicized*).**

|  | $\rho$ (10 min) | | | $\rho$ (hourly) | | | $\rho$ (daily) | | |
|---|---|---|---|---|---|---|---|---|---|
| **Original** | $\delta^{18}O$ | $\delta^2H$ | *dxs* | $\delta^{18}O$ | $\delta^2H$ | *dxs* | $\delta^{18}O$ | $\delta^2H$ | *dxs* |
| Air temperature | +0.76 | +0.73 | -0.74 | +0.76 | +0.73 | -0.75 | +0.81 | +0.77 | -0.79 |
| $H_2O$ mixing ratio | +0.83 | +0.80 | -0.79 | +0.83 | +0.80 | -0.79 | +0.86 | +0.83 | -0.82 |
| Relative humidity | +0.06 | +0.06 | -0.07 | +0.06 | +0.06 | -0.07 | *0.00* | *-0.01* | *-0.01* |
| Station pressure | +0.09 | +0.08 | -0.12 | +0.09 | +0.08 | -0.12 | +0.10 | +0.09 | -0.13 |
| Wind speed | - | - | - | -0.08 | -0.07 | +0.11 | -0.14 | -0.13 | +0.16 |
| Katabatic deviation | - | - | - | +0.50 | +0.48 | -0.50 | +0.48 | +0.47 | -0.41 |
| NAO | - | - | - | - | - | - | -0.25 | -0.23 | +0.29 |
| AO | - | - | - | - | - | - | *-0.06* | *-0.04* | +0.17 |
| Baffin Bay ice extent | - | - | - | - | - | - | -0.69 | -0.71 | +0.47 |
|  |  |  |  |  |  |  |  |  |  |
| **Seasonally-adjusted** | $\delta^{18}O$ | $\delta^2H$ | *dxs* | $\delta^{18}O$ | $\delta^2H$ | *dxs* | $\delta^{18}O$ | $\delta^2H$ | *dxs* |
| Air temperature | +0.24 | +0.20 | -0.31 | +0.25 | +0.20 | -0.32 | +0.32 | +0.27 | -0.38 |
| $H_2O$ mixing ratio | +0.55 | +0.49 | -0.58 | +0.55 | +0.49 | -0.58 | +0.57 | +0.51 | -0.60 |

| | | | | | | | | |
|---|---|---|---|---|---|---|---|---|
| Relative humidity | +0.46 | +0.43 | -0.44 | +0.38 | +0.36 | -0.33 | +0.32 | +0.30 | -0.29 |
| Station pressure | +0.04 | +0.03 | -0.07 | +0.04 | +0.03 | -0.07 | *+0.03* | *+0.02* | -0.07 |
| Wind speed | - | - | - | *-0.03* | *-0.02* | +0.06 | *-0.03* | *-0.03* | *+0.06* |
| Katabatic deviation | - | - | - | +0.19 | +0.17 | -0.18 | +0.10 | +0.09 | *-0.03* |
| NAO | - | - | - | - | - | - | -0.14 | -0.12 | +0.20 |
| AO | - | - | - | - | - | - | *-0.04* | *-0.02* | +0.17 |
| Baffin Bay ice extent | - | - | - | - | - | - | -0.02 | -0.03 | +0.05 |