# Peer review of "Baffin Bay sea ice extent and synoptic moisture transport drive water vapor isotope ( $\delta^{18}\text{O}$ , $\delta^2\text{H}$ , deuterium excess) variability in coastal northwest Greenland"

_Atmospheric Chemistry and Physics, 2020_

## Short Comment (SC1) · 5 Jun 2020

Dear Dr. Akers,

Thank you for making available such a long record of atmospheric water vapor isotopes from one of the regions of the world where climate change is most manifested.

When reading your manuscript a question about your calibration came up, which is not clearly explained by your discussions in the text.

You write that from August 2019 a calibration dry air system was installed (Line 164). No information about the dry air system is given in the text such as the humidity level

of the dry air produced.

But you also write that you perform a humidity-isotope-calibration function estimation in July 2019. I therefore must assume that this calibration is carried out using a Drierite system. You carried the humidity-isotope calibration curve between 500 and 7000 ppmv and you show in the supplementary material a strong influence of the humidity on the isotopic composition. However as shown clearly in Bastrikov 2014 there is an effect on the humidity-isotope-calibration function when using dry air produced using Drierite as the material using in the Drierite column creates some amount of isotopic fractionation. The conclusion from Bastrikov 2014 is that you cannot use the humidity isotope correction function below 3-5000 ppm when using drierite. I have attached the figure below.

Would it not be more correct to flag the data below 5000 ppm as potentially biased by the humidity-isotope correction function, or have you in other ways corrected for the use of Drierite?

I had a look at the data which you uploaded to the Arctic Data Center. I assume that this is the calibrated data, which you have uploaded, but would it be possible to also upload the non-calibrated data and information or a script to see the effect of each step of the calibration.

It is not clear if you have calibrated and how you have performed the calibration of the humidity (ppmv) of the picarro. Maybe you could add a figure similar to Bastikov 2014 as also shown below.

Could you in the supplementary data also add a plot showing the measurements of the calibration pulses and the estimate VSMOW-SLAP slope. I do notice that a great deal of your measurements are outside your range of the standards used, so it would be very useful to get an estimate of how stable and accurate the VSMOW-SLAP slope is.

Thank you for your help making such a valuable dataset extra useful for the community.

Hans Christian Steen-Larsen

[Figure]

**Figure 4.** Picarro humidity–isotope response functions. Green error bars: calibration performed using DW standard and DRIERITE column, blue error bars: calibration performed using YEKA standard and DRIERITE column, red error bars: calibration performed using DW standard and dry gas, black error bars: calibration performed using YEKA standard and dry gas. Solid lines represent linear fits to the data. For the measurements of DW and YEKA standards using dry air one conjoint fitting line is shown in red. The $y$ axis shows a bias with respect to the mean value measured at 12 000 ppmv.

**Fig. 1.**

[Figure]

**Figure 3.** Humidity measurements: meteorological sensor (Gill Instruments) vs. Picarro. Black curve: linear fit (Eq. 3).

**Fig. 2.**

[Figure]

**Figure 5.** Picarro calibration data. Top to bottom: measured $\delta^{18}O$ and $\delta D$ values in ‰ for DW standard (green dots) and YEKA standard (blue dots), calculated calibration slope for $\delta^{18}O$ measurements (red dots) and $\delta D$ measurements (purple dots), humidity concentration in ppmv (black dots).

**Fig. 3.**

---

## Referee Comment (RC1) · Anonymous Referee #2 · 15 Jun 2020

General comments:

The paper introduces a new valuable data set of water vapor isotopic observations in the coastal high Arctic Greenland. Such a data set in a region where long term monitoring is still relatively scarce would be very useful to the community in particular to the understanding of the processes influencing water isotopic composition. The overall article is well written and clearly presented. The results are presented in detail, together with a precise analysis of the local meteorology and climate. The article presents a clear statistical analysis of the influences of multiple parameters and their interactions on the water vapor isotopic composition at multiple timescales and is therefore potentially beneficial to the fields of research of paleoclimates using water isotopic proxies

or of present day atmospheric moisture cycle in the Arctic region. However, some very important aspects of the calibration procedure of the water vapor isotopic observations on which all the analyses rely are not totally clear and need to be clarified in order to validate this data set. The calibration method description is sometimes a little enigmatic and some aspects should really be clarified as several of the recommendations for long term measurements of vapor isotopes do no seem to be totally respected (Bailey et al. 2015). Technical issues did not allow applying a normal calibration procedure and therefore choices have been made not to apply all the usually recommended corrections. These choices seem to have been made in favor of a larger temporal data coverage, but the data quality is probably affected. The deviations which could be introduced by a low quality calibration are not necessarily sufficient to invalidate the data set analyses, as the ambient air variability is very strong. But at the end, a clear overall estimate of the precision and accuracy of the data set is needed to justify subsequent analyses. If the precision is too low, some of the analyzed variations might not be significant. I believe that these questions must be answered before publishing these data set and analysis.

Specific comments:

L.29: "and past reductions should be similarly preserved in local glacial ice": this is based on the hypothesis that the local precipitation and vapor isotopic composition are similarly affected by the local sea ice cover and that the isotopic signal of precipitation is preserved in ice cores. This hypothesis is better detailed in the "implications" section, which is sufficient. I believe this affirmation is rather strong for an abstract, as this has not been proven, and should be removed here or expression with more caution.

L.38: Casado et al. 2018 is about the Antarctic environment, not the Arctic and should therefore be referred differently.

L. 140-142: What is the elevation of the inlet compared to the ground level? How long is the inlet tubing? Is the inlet tubing heated and if yes, at which temperature? This last information is important as there might be condensation occurring in a non-heated

tube, in particular in very cold environments. Does the setup include any protection at the inlet to prevent snow flakes or rain from entering the system?

L. 144 to 148: Part of this paragraph could be moved to the previous section describing the local climate.

L. 155 to 157: The two liquid water isotopic standard have isotopic values of -2.24 and -29.80 ‰in $\delta^{18}$O, whereas your ambient air measurements have values ranging from around -20 to -50 ‰. If the calibration scale does not encompass the values which are measured, how can you be sure of the validity of this calibrationÂă?

L. 157 to 162: To justify the standard injection duration, which is lower than in many other studies, the stability of the isotopic values over the averaging period of the injection should be verified. A statistical analysis of the stability of $\delta^{18}$O and $\delta$D values or a figure showing their evolution over the 10 minutes should be added to the supplementary materials at least.

L. 168 to 169: If I understand yell, you finally estimated the stability of the system over two months and did not apply any correction over the two years. However, you can hardly justify that there is no drift of the instrument sensitivity at time scales longer than a few months with these observations. The instrument sensitivity can also be different each time it is restarted or the monitoring program is launched and a new calibration scale should be calculated. Has the instrument never been restarted over two years, or can you justify its stability? I believe this is a very important point regarding the quality and accuracy of your dataset.

L.177 to 186: If the humidity sensitivity has been estimated based on a experiment performed in July 2019, the dry air system had not yet been installed. If the humidity response function has been estimated based on measurements performed with the DRIERITE as a dry air source, it is highly possible that the remaining moisture in the dry air source is strongly influencing the measurements at low humidity. This can lead to strong deviations at humidity values below a few thousand ppms and could

explain a very large part of the deviations depicted on Figure S1. It would be helpful to have an idea of the background humidity levels when injecting dry air produced via the DRIERITE without any liquid standard injection, in order to evaluate the validity of the humidity sensitivity corrections. If you apply a correction of the humidity sensitivity based on a experiment biased by a low quality air source, you will introduce this bias into your calibrated data set. Furthermore, more details should be given on the procedure to apply the humidity response correction has been applied.

L. 196 to 197: How did you add the CRDS mixing ratio values to the values recorded by the weather station? Did you use these measurements to fill gaps in the weather station records, or did you use any kind of average between the two sensors? Did you apply any calibration of the mixing ratio measured by the CRDS analyzer, or did you compare it with measurements performed by the weather station?

L. 580 – 604: The unique diel cycle observed in March in water isotopes is very similar to the observations of Bonne et al. 2020 in Siberia, which is cited in the previous section. The environment is a little different as there can not be any influence of katabatic winds in the Siberian sector, but this spring diel cycle is also attributed to the sublimation of the snow deposited earlier in winter, which has an isotopic composition different than the spring water vapour isotopic composition. Similarly, they don't see any significant diel cycle in other seasons. I think this would be worth including a short comment relating both situations.

Technical comments:
L. 122: Change "kmh$^{-1}$" into "km h$^{-1}$".

L. 139: "The L2130-i + SDM uses cavity ring-down spectroscopy". It does not seem necessary to add "+SDM", as the SDM is the calibration samples injection module, not the spectrometer itself.

L.140: Wouldn't "suited" be more suited than "amenable"?

L. 252: "over 30‰(-49.5‰ to -17.5‰) 230‰(-377‰ to -142‰), and 55‰(-7.6‰ to 47.5‰)": correct positions of the commas "over 30‰(-49.5‰ to -17.5‰), 230‰(-377‰ to -142‰) and 55‰(-7.6‰ to 47.5‰)"

L. 271: "the effects of the these factors"

L. 307: "predominate" > "predominant"

L. 318: our analysis reveals that a large amount

L. 342: "data's" > "data's"

L. 360-361: winds source have isotopically light vapor / winds source isotopically heavy vapor

L. 367: "on the L2130-i" is not necessary

L. 385: "composition, (Casado et al., 2018; Kopec et al., 2019)" : comma is not needed

L. 469: very warm observations at Thule come during this period

L. 590: "snow grains that have equilibrated"

L. 692: "adding additional": formulation could be improved

References:
Bailey, A., Noone, D., Berkelhammer, M., Steen-Larsen, H. C., and Sato, P.: The stability and calibration of water vapor isotope ratio measurements during long-term deployments, Atmos. Meas. Tech., 8, 4521–4538, https://doi.org/10.5194/amt-8-4521-2015, 2015.

---

## Short Comment (SC2) · 16 Jun 2020

Thank you for the comments Hans Christian,

Further discussion and elaboration on the calibration procedures will be included in a longer upcoming response. But for a quick clarification on your main question, please see the following:

In our paper we stated that we installed our dry air system in August 2019, but that we performed the humidity response calibration in July 2019. This would suggest that our calibration was done before the dry air system. Actually, this due to poor phrasing

in our manuscript. We began installing the dry air system in late July. Once it was running well, we first performed a humidity response calibration with the dry air to check the system and calibrate further results. This and other testing of the SDM was completed by the first of August, when the system was fully set to run on the dry air with an automated standard run cycle. So the humidity response calibration was, in fact, performed with the dry air system, which has a residual humidity under 100 ppmv when no standards are being injected.

We also had performed similar humidity response calibrations using the drierite system before we had the dry air system installed. The results from these runs clearly showed the influence of incomplete drying (as highlighted in Bastrikov 2014), which motivated our installation of the dry air system. Namely, as the humidity lowered, the standard isotopic values trended away from the standard value and toward the intermediate ambient humidity isotopic value. This resulted in calibration curves with opposite trends between the two standards. In contrast, with the dry air system, the humidity response curves are very similar in trend direction and shape for both standards, which is what we expect if we are capturing the innate analytical bias of the machine itself.

I will continue to work on answering your remaining questions for our larger future response. Thank you again!

---

## Referee Comment (RC2) · Anonymous Referee #3 · 29 Jul 2020

The article presents a 2-year record of water vapor isotopic measurements at a coastal station in Northwestern Greenland. This is part of a larger effort to build a network of water isotopic observations at different Arctic sites. The continuous record allows to analyze the isotopic variability at the diel, synoptic and seasonal time scales. Isotopic variations are interpreted in terms of meteorological factors. Implications for paleoclimate studies are discussed.

The paper reads very well. The rationales are sound. I only have minor comments.

[Figure]

**Minor comments**

- section 4 "general overview of isotopic results": I found it difficult to read because it's only text and there is no reference to any figure or table. I suggest to add references to figure 2 and/or table 1 and to relate the text to them. Also, this section is very short and does not add much to the paper. Maybe it could be moved into an introductory sub-section of the next section?

- l 255-265: this paragraph does not add much to the paper. This information is re-used in the next sections. This paragraph could be removed and the information could be added when needed in the next sections. The correlation numbers could also be gathered in a table, which would make them easier to read and to refer to in the text.

- l 365: "personal field observations on multiple occasions": can we see them in Figure 2? It would be more convincing to directly show it in the Figure (or in a zoom of this figure) than to refer to personal observations.

- l 391: Is this effect supported by observations? Or is it too small to be seen? If so, clarify that this effect is expected to be weak?

- section 6 "Annual isotopic cycle"; the title does not represent the content of this section. This section looks at correlations within each month, so it is rather synoptic variability. This section could be renamed "Synoptic-scale isotopic variability".

- l 429: can we see it in a Figure? If so, refer to it.

- Same l 436.

- l 442: "isotopic variables": only dxs, isn't?

- l 444-449: is this paragraph about diel variability? If so, maybe this can be moved to a later section on diel variability?

- l 469: "com"?

- section 7: this section is actually a deeper analysis of the previous paragraph. Therefore, I suggest to move this section into a last sub-section of section 6. It could be renamed "6.5. Moisture pulse events". The content can be merged with the last paragraph of section 6.4 to avoid redundancy.

- l 611: Why can't it just be the effect of boundary layer mixing? At midday, boundary layer mixing is more intense, bringing more depleted and high-dxs water vapor from the free troposphere down to the surface, e.g. the mechanism that you describe in lines 490-493.

- l 661: Start a new paragraph?

- Figure 3: what are the white and magenta lines? Explain in caption

- Figure 5: I really enjoyed this figure.

- Figure 6: what is the time scale for the correlations? 10 minute, 1 hour or daily? Explain in caption.

---

## Referee Comment (RC3) · Franziska Aemisegger (Referee) · 11 Aug 2020

**Review of "Baffin Bay sea ice extent and synoptic moisture transport drive water vapor isotope ( $\delta^{18}$ O, $\delta$ D, d-excess) variability in coastal northwest Greenland " by Pete D. Akers et al.**

This paper presents a two-year time series of high time resolution water vapour isotope measurements from Thule in the northern Baffin Bay with the aim to investigate the synoptic drivers of the isotope variability measured in the region of the Greenland high Arctic. Five interacting factors are presented, that are thought to determine the isotope signals' variability at the daily to annual timescales. These factors include mainly local environmental conditions (temperature, marine moisture availability, surface winds, NAO and the contribution of land evaporative sources). The relative contribution of the different factors is thought to change with the seasonal cycle and in the coming years with the interannual variability in the extent of the sea ice. Overall, I found this well-written paper inspiring to read, it presents good quality measurement data, shows carefully compiled figures and several interesting analyses. I found the discussion related to the role of sea ice particularly interesting. I have three major comment on the science as well as a few minor comments listed below.

My major comments are:

- 1) The analysis on the five factors determining the stable water vapour isotope variability measured at Thule at different timescales is very interesting. However, I had difficulties to evaluate the independence of these five factors and also found them to be chosen in a subjective way. The authors give no motivational framework of the basic physical mechanisms that would justify choosing these 5 factors as basic variables that determine isotopic variations. Could the authors provide a more thorough introduction into why they think these five factors are the relevant ones to be studied? Others would be just as relevant such as e.g. the relative humidity with respect to sea surface temperature, cloud condensation temperature or sea surface temperature, which are the traditional variables that are studied as environmental controls of stable water isotope variability. For these traditional variables, physical frameworks exist that explain why they are relevant: e.g. the formation of clouds during moist adiabatic ascent of air parcels (Rayleigh distillation framework, Dansgaard 1964) for cloud condensation temperatures and the Craig and Gordon 1965 ocean evaporation model for SST and the relative humidity with respect to SST.
- 2) To me highlighting the importance of the atmospheric circulation and at the same time underlining the relevance of local environmental conditions is somewhat contradictory. Many previous studies have used trajectory analysis to show the relevance of environmental conditions at the moisture source for the variability of stable water isotope measurements in water vapour (e.g. Pfahl and Wernli 2008; Aemisegger et al. 2014, Aemisegger 2018; Thurnherr et al. 2020). Here the authors say the circulation and the local conditions are key. I would find it useful, if there was a comment on this apparent contradiction in the paper. Or if it is not a contradiction, then to resolve the misunderstanding and explain why the results of this paper are in agreement with these previous studies.

3) My third major comment is a more technical one: the presentation of the calibration and postprocessing framework of the exceptionally long and very valuable Arctic water vapor isotope time series lacks some details in particular on the total uncertainty of the measurements (see also my minor comments 5-11, below).

Minor comments:

- P. 1, L. 1: The sea ice extent seems to come out as the most important factor controlling if moisture is mainly sourced from the local environment or if it is transported from further away. This could be mentioned more clearly in the abstract before the five controlling factors. In my opinion it comes a bit late in the current version.
- 2) P. 2, L. 50: There were many studies investigating the quality of laser spectrometric measurements in the early 2010s, add "e.g." and maybe Sturm and Knohl 2012 and Aemisegger et al. 2012 could be cited as well, since the latter study particularly focused on the capability of laser systems to resolve the synoptic timescale variability of water vapour isotopes.
- 3) P. 2, L.50: The Yale database could be cited here, since it groups most of the already published water vapour isotope data: Wei et al. 2019.
- 4) P. 3, L. 75 "critically giving a second set of observations to derive annual patterns and anomalies" not sure if I understand this correctly. What do the authors mean here?
- 5) P. 5, L. 140: Did the authors test the response times of their system using "Tygon tubing". Several early studies (e.g. Sturm and Knohl 2010; Tremoy et al. 2011; Aemisegger et al. 2012) showed that certain tubing materials induce very large residence times and unwanted strong interactions between the tubing wall and the sample gas.
- 6) P. 5, L. 140: What was the residence time of the sampled air in the tubing, how long was the tubing, was it heated, was the inlet shielded? These are all very important points for performing high quality stable water vapour isotope measurements especially in extreme environments such as in northern Greenland.
- 7) P. 5, L. 151: Introduce the delta notation and the normalisation to the international VSMOW-VSLAP scale.
- 8) P.6, L. 155: The standards' isotopic composition does not bracket the measured isotope signals. The authors should explicitly mention this and comment on the expected impact of this extrapolation on the total uncertainty of their measurements.
- 9) P. 6, L. 175: Even though the drift of the Picarro laser spectrometers is limited regular calibrations should be carried out to 1) survey the good functioning of the system and 2) to provide a long term assessment of the total uncertainty of the measurements (see, Aemisegger et al. 2012; Thurnherr et al. 2020). In particular, Thurnherr et al. 2020 shows that different post processing procedures lead to substantial changes in the isotope data, in particular, with respect to the treatment of the water vapour mixing ratio dependent isotope bias correction.
- 10) P. 6, L. 181: The precision (Allan variance, or standard deviation of a constant water vapour isotope signal) strongly depends on the water vapour mixing ratio (see Aemisegger et al. 2012 and Sodemann et al. 2017). Please indicate the total uncertainty of the measurements as a function of water vapour mixing ratio. This is very important, given the very low levels of humidity observed at Thule in winter.

- 11) P. 7, L. 195: Was the water vapour mixing ratio of the L2130 calibrated using a dew point generator or another humidity sensor installed in parallel? Without calibration the reading of the laser spectrometric volume mixing ratio may be biased.
- 12) P. 8, L. 225: Please indicated the horizontal and vertical grid resolution of the MERRA-2 reanalysis data. Note that HYSPLIT is not a model but a post-processing tool, thus it cannot be "forced". I would suggest to write: "... with air parcel backtrajectories calculated based on three-dimensional MERRA-2 wind fields...". I am not convinced that choosing only 10 days per months produces a robust two-year climatology. But given the limited use that is made of the trajectory climatology in this paper, the approach is ok.
- 13) P. 9, L. 254: "The magnitude of irregular hourly to weekly variations" do you mean synoptic timescale variations?
- 14) P. 10, L. 291: What is meant by "temperature-driven equilibrium fractionation"? I think the cited literature is a bit misleading. Dütsch et al. 2017 shows that the condensation temperature indeed has a certain impact on the deuterium excess. Pfahl and Sodemann et al. 2014 discuss the effect of the SST.
- 15) P. 10, L. 293: Interesting that a negative correlation between the deuterium excess and the temperature is found! Did the authors also look at the correlation with nearby SSTs? In climate reconstructions based on ice cores a positive relation between the deuterium excess and moisture source SST is assumed (e.g. Johnsen et al. 1989; Vimeux et al. 1999; Stenni et al. 2001). However, in a detailed analysis of the correlation behaviour between the deuterium excess and SST a recent study (Aemisegger and Sjolte 2018) found different regions (in particular at high latitudes) that are expected to exhibit a negative correlation based on the Craig Gordon model and the closure assumption. It is thought that this negative correlation arises from a positive feedback mechanism between the SST and the relative humidity with respect to SST. Such a negative correlation regime is expected to be dominant particularly in regions where the variability in air-sea interactions is mainly driven by variability in atmospheric circulation and not primarily by variations in ocean circulation. If the deuterium excess also shows a negative correlation to the nearby SST at Thule this would be evidence for such a behaviour. Of course, the time series at Thule is too short to look at this in detail. But still, I find the negative correlation between air temperature and deuterium excess that is found here very interesting and since it is of opposite sign with respect to the traditional interpretation of the deuterium excess in ice core studies, I would find it worthwhile to shortly discuss this.
- 16) P. 11, L. 320: The relative humidity is very low above forming ice? Can the authors show some evidence or cite some literature?
- 17) P. 11, L. 323: Phase changes in the NAO might be confusing in the context of isotopes and water phase changes. Is there another way to formulate the change in the NAO sign?
- 18) P. 12, L341-L356: Very interesting discussion on the role of the NAO and the sea ice extent!
- 19) P. 14, L. 394: I think the relations to the "traditional driving variables" SST and relative humidity with respect to the SST should at least be shortly mentioned.
- 20) P. 14, L. 396 and 399: "southerly" flow instead of "southern" flow.

- 21) P. 15, Section 6.3 How large is the influence of more distant land sources of e.g. northern Canada? From Fig. 3 one might think that they may play a significant role.
- 22) P. 18, Section 7: Additionally, I suspect that Arctic anticyclones would play a key role in the synoptic timescale variability at Thule (in Greenland blocking situations). A case study of an Arctic Blocking event (over northern Russia) is shown in Schneider et al. 2019, which highlights the very large horizontal gradients resulting from the subsidence induced drying at the core of the anticyclone and the progressive atmospheric moisture uptake along the anticyclone edges.
- 23) P. 18, L. 542: I doubt that in the case of sea ice much water vapour at Thule originates from the deep tropics or subtropics. The analysis shown in Fig. 3 does not support such a statement. Here I think one can safely write "advection from the midlatitudes".
- 24) P. 21, L. 609: I don't understand why an input of low  $\delta^{18}$ O/high dxs from evaposublimation results in a  $\delta^{18}$ O maximum and dxs minimum at midday.
- 25) P. 21, L. 627: It would be great to clearly mention that evaporation of meltwater and sublimation of snow might not carry the same isotopic composition (see e.g. Christner et al. 2017).
- 26) P. 22, L. 664: Maybe not single extreme events but years with high frequency of occurrence of warm advection events, e.g. due to a northward shift of the storm track?
- 27) P. 23: In the conclusion it would be very nice to mention the great use of this dataset for the validation of high-resolution isotope-enabled simulations in the Arctic to study the importance of air-ice and air-sea interaction processes in more detail. Here the great value of the data is their long temporal coverage, thanks to which model-based sensitivity experiments could be performed to test the importance of different driving factors (similar to the climatological sensitivity study over Europe by Christner et al. 2018).

**References:**

Aemisegger, F., Sturm, P., Graf, P., Sodemann, H., Pfahl, S., Knohl, A., and Wernli, H.: Measuring variations of  $\delta^{18}$ O and  $\delta^{2}$ H in atmospheric water vapour using two commercial laser-based spectrometers: an instrument characterisation study, Atmos. Meas. Tech., 5, 1491-1511, doi:10.5194/amt-5-1491-2012, 2012.

Aemisegger F., Pfahl S., Sodemann H., Lehner I., Seneviratne S., and Wernli H.: Deuterium excess as a proxy for continental moisture recycling and plant transpiration. Atmos. Chem. Phys. 14, 4029-4054, doi:10.5194/acp-14-4029-2014, 2014.

Aemisegger, F. and Sjolte, J.: A climatology of strong large-scale ocean evaporation events. Part II: relevance for the deuterium excess signature of the evaporation flux. J. Climate, 31, 7313–7336, doi:10.1175/JCLI-D-17-0592.1, 2018.

Aemisegger, F.: On the link between the North Atlantic storm track and precipitation deuterium excess in Reykjavik, Atmos. Sci. Lett., 19:e865, doi:10.1002/asl.865, 2018.

Christner, E., Kohler, M., and Schneider, M.: The influence of snow sublimation and meltwater evaporation on  $\delta D$  of water vapor in the atmospheric boundary layer of central Europe. Atmospheric Chemistry and Physics, 17(2), 1207–1225, doi:10.5194/acp-17-1207-2017, 2017. Christner, E., Aemisegger, F., Pfahl, S., Werner, M., Cauquoin, A., Schneider, M., Hase F., Barthlott S., and Schädler G.: The climatological impacts of continental surface evaporation, rainout, and subcloud processes on  $\delta D$  of water vapor and precipitation in Europe. Journal of Geophysical Research: Atmospheres, 123, 4390–4409, doi:10.1002/2017JD027260, 2018.

Pfahl, S., and Wernli, H.: Air parcel trajectory analysis of stable isotopes in water vapor in the eastern Mediterranean, J. Geophys. Res., 113, D20104, doi:10.1029/2008JD009839, 2008.

Craig, H. and Gordon, L.: Deuterium and oxygen 18 variations in the ocean and the marine atmosphere, in: Proceedings of the Stable Isotopes in Oceanographic Studies and Paleotemperatures, 1965.

Dansgaard, W.: Stable isotopes in precipitation, Tellus, 16, 436–468, doi:10.1111/j.2153-3490.1964.tb00181.x, 1964.

Sodemann, H., Aemisegger, F., Pfahl, S., Bitter, M., Corsmeier, U., Feuerle, T., Graf, P., Hankers, R., Hsiao, G., Schulz, H., Wieser, A., and Wernli, H.: The stable isotopic composition of water vapour above Corsica during the HyMeX SOP1 campaign: insight into vertical mixing processes from lower-tropospheric survey flights, Atmos. Chem. Phys., 17, 6125– 6151, doi:10.5194/acp-17-6125-2017, 2017.

Thurnherr, I., Kozachek, A., Graf, P., Weng, Y., Bolshiyanov, D., Landwehr, S., Pfahl, S., Schmale, J., Sodemann, H., Steen-Larsen, H. C., Toffoli, A., Wernli, H., and Aemisegger, F.: Meridional and vertical variations of the water vapour isotopic composition in the marine boundary layer over the Atlantic and Southern Ocean, Atmos. Chem. Phys., 20, 5811–5835, doi:10.5194/acp-20-5811-2020, 2020.

Tremoy, G., Vimeux, F., Cattani, O., Mayaki, S., Souley, I. and Favreau, G., Measurements of water vapor isotope ratios with wavelength-scanned cavity ring-down spectroscopy technology: new insights and important caveats for deuterium excess measurements in tropical areas in comparison with isotope-ratio mass spectrometry. Rapid Commun. Mass Spectrom., 25: 3469-3480. doi:10.1002/rcm.5252, 2011.

Wei, Z., Lee, X., Aemisegger, F., Benetti, M., Berkelhammer, M., Bonne, J.-L., Casado, M., Caylor, K., Christner, E., Dyroff, C., García, O. E., González, Y., Griffis, T., Kurita, N., Liang, J., Liang, M.-C., Lin, G., Noone, D., Gribanov, K., Munksgaard, N.-C., Schneider, M., Ritter, F., Steen-Larsen, H. C., Vallet-Coulomb, C., Wen, X., Wright, J. S., Xiao, W., Yoshimura, K.: A global database of water vapour isotopes measured with high temporal resolution infrared laser spectroscopy, Scientific Data, 6, 180302, doi:10.1038/sdata.2018.302, 2019.

---

## Author Response (AR1)

We would like to thank all reviewers for their helpful advice and edits. We have tried to answer and fulfill their suggestions to the best of our abilities. Overall, we have made additional phrasing and typographical edits to improve readability and clarity. We have also switched phrasing of δD to δ²H and use in figures of d-excess to *dxs* for consistency. Other changes are detailed below.

**Improved Wind Data**
During the revision process, we found and included an improved wind data source. Both the initial and improved wind datasets come from the same observation station (THU airport). Our initial data set gave the wind data (speed and direction) as hourly data, and we thought that this was the direct data from the observation site. However, we later discovered that the airport observations are not necessarily hourly, and additional sub-hourly observations are often included (possibly when flights are arriving or departing). The initial data we used in the original manuscript appears to have aggregated all these values into hourly observations to produce a consistent hourly database; however, the aggregation method used seems to have simply averaged all observations in each hourly block. For wind directions, this causes problems due to the degree nature of observations (e.g., simply averaging a 350° north wind and 10° north wind will result in a 180° SOUTH wind observation). Our improved wind data source includes all the original observations from the airport, and we could then aggregate the data to hourly values correctly (through conversion to u and v components).

The resulting improved dataset does not greatly differ from our initial set and no major changes are made to our paper's conclusions. The greatest change is that, previously, the correlations between katabatic deviation (i.e., wind direction) and isotopic variables dropped to near zero in July, and we struggled to explain this phenomenon. This now appears to have been an artifact of the poor initial wind database, as the correlations calculated in the improved data set show a smooth pattern throughout summer as expected.

**RC1 Response**
The paper introduces a new valuable data set of water vapor isotopic observations in the coastal high Arctic Greenland. Such a data set in a region where long term monitoring is still relatively scarce would be very useful to the community in particular to the understanding of the processes influencing water isotopic composition. The overall article is well written and clearly presented. The results are presented in detail, together with a precise analysis of the local meteorology and climate. The article presents a clear statistical analysis of the influences of multiple parameters and their interactions on the water vapor isotopic composition at multiple timescales and is therefore potentially beneficial to the fields of research of paleoclimates using water isotopic proxies or of present day atmospheric moisture cycle in the Arctic region. However, some very important aspects of the calibration procedure of the water vapor isotopic observations on which all the analyses rely are not totally clear and need to be clarified in order to validate this data set. The calibration method description is sometimes a little enigmatic and some aspects should really be clarified as several of the recommendations for long term measurements of vapor isotopes do no seem to be totally respected (Bailey et al. 2015). Technical issues did not allow applying a normal calibration procedure and therefore choices have been made not to apply all the usually recommended corrections. These choices seem to have been made in favor of a larger temporal data coverage, but the data quality is probably affected. The deviations which could be introduced

by a low quality calibration are not necessarily sufficient to invalidate the data
set analyses, as the ambient air variability is very strong. But at the end, a clear overall
estimate of the precision and accuracy of the data set is needed to justify subsequent
analyses. If the precision is too low, some of the analyzed variations might not be significant.
I believe that these questions must be answered before publishing these data
set and analysis.

L.29: "and past reductions should be similarly preserved in local glacial ice": this is
based on the hypothesis that the local precipitation and vapor isotopic composition are
similarly affected by the local sea ice cover and that the isotopic signal of precipitation
is preserved in ice cores. This hypothesis is better detailed in the "implications" section,
which is sufficient. I believe this affirmation is rather strong for an abstract, as this has
not been proven, and should be removed here or expression with more caution.
**L29: Rephrased to be more clear that this is only a proposed concept by this paper. This
concept has been independently supported by other regional ice core research such as
Osterberg 2015, and we have not fully eliminated this from the abstract as we believe it
highlights an important connection between our research and the broader paleo
community.**

L.38: Casado et al. 2018 is about the Antarctic environment, not the Arctic and should
therefore be referred differently.
**L 38: Removed Casado citation.**

L. 140-142: What is the elevation of the inlet compared to the ground level? How long
is the inlet tubing? Is the inlet tubing heated and if yes, at which temperature? This
last information is important as there might be condensation occurring in a non-heated tube,
in particular in very cold environments. Does the setup include any protection at
the inlet to prevent snow flakes or rain from entering the system?
 **L 140-142: Clarified to answer questions and add additional information.**

L. 144 to 148: Part of this paragraph could be moved to the previous section describing
the local climate.
**L 144-148: Paragraph reworked as requested**

L. 155 to 157: The two liquid water isotopic standard have isotopic values of -2.24 and
-29.80 ‰in _18O, whereas your ambient air measurements have values ranging from
around -20 to -50 ‰. If the calibration scale does not encompass the values which are
measured, how can you be sure of the validity of this calibrationĂˇa?
**L 155-157: This is indeed a known issue. When the observation system was originally set
up, the plan was to observe values only in summer. However, the extension of the
system's observations through winter months does require standards with lower values
for a properly robust calibration. Such standards are planned to be added at the next
visit to the station. As for the potential impact on our results, it is true that the absolute
values of observations below our standard water range risk an accuracy offset or bias
that is not detected in our current calibrations. A paragraph has now been added to
acknowledge this. However, we believe that the overall impact of a possible low-value
bias on our study is fairly small because our conclusions and analysis largely focus on
relative value changes and not formulas/equations that require absolute accuracy. Thus,
even if the lowest observations are skewed in value somewhat too high or too low, this
bias would very likely be graduated in degree with increasingly lower isotopic values.**

**Thus, a single observation's relative position (higher or lower) to other observed values (what we base our analyses on) would be preserved even if a bias exists.**

L. 157 to 162: To justify the standard injection duration, which is lower than in many other studies, the stability of the isotopic values over the averaging period of the injection should be verified. A statistical analysis of the stability of _18O and _D values or a figure showing their evolution over the 10 minutes should be added to the supplementary materials at least.

**L 157-162: A table is now added to the supplement (Table S3) that provides the slopes of isotopic values vs. time for each of the calibration runs, and summarized in the main text. This shows that the isotopic change after 10 minutes is acceptably flat. While the mean trends are slightly positive for three of the calibration isotopic values (USGS 45 $\delta^{18}O$ and USGS 46 $\delta^{18}O$ and $\delta D$), the mean value is within one standard deviation of zero. We judged that any possible improvements to calibration accuracy from extending the calibration time would be very small and not worth the loss of dry air from the extended run durations.**

L. 168 to 169: If I understand yell, you finally estimated the stability of the system over two months and did not apply any correction over the two years. However, you can hardly justify that there is no drift of the instrument sensitivity at time scales longer than a few months with these observations. The instrument sensitivity can also be different each time it is restarted or the monitoring program is launched and a new calibration scale should be calculated. Has the instrument never been restarted over two years, or can you justify its stability? I believe this is a very important point regarding the quality and accuracy of your dataset.

**L 168-169: Some clarifying lines are added to the discussion here. Comparison of isotope-isotope and isotope-climate relationships across the period of record fail to show any clear evidence of long term drift or changes in sensitivity. The machine has not been moved or shut down for extended times, with the only restarts occurring after brief power outages. It is still possible that some degree of sensor drift has occurred over our period of record, but we do not see evidence of anything large enough to alter the study's conclusions that are largely based on isotopic changes of >5-20‰.**

L.177 to 186: If the humidity sensitivity has been estimated based on a experiment performed in July 2019, the dry air system had not yet been installed. If the humidity response function has been estimated based on measurements performed with the DRIERITE as a dry air source, it is highly possible that the remaining moisture in the dry air source is strongly influencing the measurements at low humidity. This can lead to strong deviations at humidity values below a few thousand ppms and could explain a very large part of the deviations depicted on Figure S1. It would be helpful to have an idea of the background humidity levels when injecting dry air produced via the DRIERITE without any liquid standard injection, in order to evaluate the validity of the humidity sensitivity corrections. If you apply a correction of the humidity sensitivity based on a experiment biased by a low quality air source, you will introduce this bias into your calibrated data set. Furthermore, more details should be given on the procedure to apply the humidity response correction has been applied.

**L 177-186: This comment is drawn from a misunderstanding of the humidity response calibration. The reviewer believes it to predate the installation of the dry air system based on our previously confusing wording in the manuscript. This wording has been altered to make it clear that the humidity response calibration was performed with the**

**dry air system. A line was added to make the application of the correction more explicitly clear.**

L. 196 to 197: How did you add the CRDS mixing ratio values to the values recorded by the weather station? Did you use these measurements to fill gaps in the weather station records, or did you use any kind of average between the two sensors? Did you apply any calibration of the mixing ratio measured by the CRDS analyzer, or did you compare it with measurements performed by the weather station?

**L 196-197: Line added to clarify. Mixing ratios from the Picarro were added as an independent weather variable, but mixing ratios were also calculated from the SMT weather station data for comparison. The two mixing ratio datasets have a Spearman correlation of +0.99.**

L. 580 – 604: The unique diel cycle observed in March in water isotopes is very similar to the observations of Bonne et al. 2020 in Siberia, which is cited in the previous section. The environment is a little different as there can not be any influence of katabatic winds in the Siberian sector, but this spring diel cycle is also attributed to the sublimation of the snow deposited earlier in winter, which has an isotopic composition different than the spring water vapour isotopic composition. Similarly, they don't see any significant diel cycle in other seasons. I think this would be worth including a short comment relating both situations.

**L 580-604: Commenting lines added, thanks for the suggestion.**

**Technical comments:** L. 122: Change "kmh⁻1" into "km h⁻1".
L. 139: "The L2130-i + SDM uses cavity ring-down spectroscopy". It does not seem necessary to add "+SDM", as the SDM is the calibration samples injection module, not the spectrometer itself.
L.140: Wouldn't "suited" be more suited than "amenable"?
L. 252: "over 30‰(-49.5‰to -17.5‰) 230‰(-377‰to -142‰), and 55‰(-7.6‰to 47.5‰)": correct positions of the commas "over 30‰(-49.5‰to -17.5‰), 230‰(-377‰to -142‰) and 55‰(-7.6‰to 47.5‰)"
L. 271: "the effects of the these factors"
L. 307: "predominate" > "predominant"
L. 318: our analysis reveals that a large amount
L. 342: "data's" > "data's"
L. 360-361: winds source have isotopically light vapor / winds source isotopically heavy vapor
L. 367: "on the L2130-i" is not necessary
L. 385: "composition, (Casado et al., 2018; Kopec et al., 2019)" : comma is not needed
L. 469: very warm observations at Thule come during this period
L. 590: "snow grains that have equilibrated"
L. 692: "adding additional": formulation could be improved
**Corrected as suggested**

**RC2 Response**
The article presents a 2-year record of water vapor isotopic measurements at a coastal station in Northwestern Greenland. This is part of a larger effort to build a network of water isotopic observations at different Arctic sites. The continuous record allows to analyze the isotopic variability at the diel, synoptic and seasonal time scales. Isotopic variations are interpreted in

terms of meteorological factors. Implications for paleoclimate studies are discussed. The paper reads very well. The rationales are sound. I only have minor comments.

section 4 "general overview of isotopic results": I found it difficult to read because it's only text and there is no reference to any figure or table. I suggest to add references to figure 2 and/or table 1 and to relate the text to them. Also, this section is very short and does not add much to the paper. Maybe it could be moved into an introductory sub-section of the next section?

**Section 4: This section is now largely integrated into the next section, with one part moved to new Supplement Section 3. Table 2 now has the correlations in one place.**

• l 255-265: this paragraph does not add much to the paper. This information is reused in the next sections. This paragraph could be removed and the information could be added when needed in the next sections. The correlation numbers could also be gathered in a table, which would make them easier to read and to refer to in the text.

**L 255-265: Paragraph removed and necessary information integrated into following section and table 2, as suggested.**

l 365: "personal field observations on multiple occasions": can we see them in Figure 2? It would be more convincing to directly show it in the Figure (or in a zoom of this figure) than to refer to personal observations.

**L 365: This sentence was deleted in edits, as the citing of anecdotal evidence was not needed as the previously stated statistics provide the necessary evidence. Visual supporting evidence is not easily clear at the temporal resolution of Figure 2, but is seen and discussed in more detail in the diel section and Figure 8.**

l 391: Is this effect supported by observations? Or is it too small to be seen? If so, clarify that this effect is expected to be weak?

**L 391: We are not sure which effect this comment is referring to. We assume it references the effect of plant transpiration, and we mentioned that we expect it to be very weak because plant cover is so sparse at Thule. We have rephrased the sentence to be more explicitly clear that this is an assumption of ours, but it is based on the cited work that quantified how little plant productivity exists in this landscape.**

• section 6 "Annual isotopic cycle"; the title does not represent the content of this section. This section looks at correlations within each month, so it is rather synoptic variability. This section could be renamed "Synoptic-scale isotopic variability".

**Section 6 (now 5): We have changed the title and some organization of this section to highlight its focus on seasonal changes in isotopic relationships and variability. However, we don't agree that the overall section is best described as focusing on synoptic patterns. Within this section, we discuss why the annual cycle observed in the isotopes emerges as a result of changes in the environment in different seasons. Some of these environmental differences include changes in the general synoptic patterns and thus we examine the isotopes and their relationships at monthly scales. Still, the overall focus of the section is not analyzing synoptic changes independently of seasonal identity (such as examining how particular synoptic atmospheric patterns consistently alter isotopic values whether in winter or summer, as we highlight in the following section), but rather specifically in the context of the annual isotopic cycle.**

l 429: can we see it in a Figure? If so, refer to it.

**L 429: Reference to figure 2 added**

• Same l 436.
**L 436: Reference to figure 2 added**

• l 442: "isotopic variables": only dxs, isn't?
**L 442: The referenced figures show that the correlations are very strong for both δ$^{18}$O and dxs, though one is strongly negative and the other strongly positive.**

l 444-449: is this paragraph about diel variability? If so, maybe this can be moved to a later section on diel variability?
**L 444-449: While this does mention changes occurring on a diel cycle, the focus here is that the sea ice breakup in spring directly leads to the sea breeze beginning which produces the strong katabatic deviation correlations of spring. The actual mechanics of the diel sea breeze and closer examination of the diel cycle in isotopes related to it are covered in the later diel section.**

• l 469: "com"?
**L 469: Edited**

section 7: this section is actually a deeper analysis of the previous paragraph. Therefore, I suggest to move this section into a last sub-section of section 6. It could be renamed "6.5. Moisture pulse events". The content can be merged with the last paragraph of section 6.4 to avoid redundancy
**Section 7 (now 5.6): With the reframing of section 6 (now 5), this moisture pulse section was added as a final subheading of that section.**

l 611: Why can't it just be the effect of boundary layer mixing? At midday, boundary layer mixing is more intense, bringing more depleted and high-dxs water vapor from the free troposphere down to the surface, e.g. the mechanism that you describe in lines 490-493.
**L 611: Line added to include this possibility. However, local evaporation is still a likely contributor as well, so both options are presented here.**

l 661: Start a new paragraph?
**L 661: Paragraph structure altered**

Figure 3: what are the white and magenta lines? Explain in caption
**Figure 3: Clarification added to caption (the lines show the 95% and 5% sea ice concentration)**

Figure 6: what is the time scale for the correlations? 10 minute, 1 hour or daily? Explain in caption.
**Figure 6: Resolutions used in calculations clarified in caption.**

**RC3 Response**
This paper presents a two-year time series of high time resolution water vapour isotope measurements from Thule in the northern Baffin Bay with the aim to investigate the synoptic drivers of the isotope variability measured in the region of the Greenland high Arctic. Five interacting factors are presented, that are thought to determine the isotope signals' variability at the daily to annual timescales. These factors include mainly local environmental conditions

(temperature, marine moisture availability, surface winds, NAO and the contribution of land evaporative sources). The relative contribution of the different factors is thought to change with the seasonal cycle and in the coming years with the interannual variability in the extent of the sea ice. Overall, I found this well-written paper inspiring to read, it presents good quality measurement data, shows carefully compiled figures and several interesting analyses. I found the discussion related to the role of sea ice particularly interesting. I have three major comment on the science as well as a few minor comments listed below.

The analysis on the five factors determining the stable water vapour isotope variability measured at Thule at different timescales is very interesting. However, I had difficulties to evaluate the independence of these five factors and also found them to be chosen in a subjective way. The authors give no motivational framework of the basic physical mechanisms that would justify choosing these 5 factors as basic variables that determine isotopic variations. Could the authors provide a more thorough introduction into why they think these five factors are the relevant ones to be studied? Others would be just as relevant such as e.g. the relative humidity with respect to sea surface temperature, cloud condensation temperature or sea surface temperature, which are the traditional variables that are studied as environmental controls of stable water isotope variability. For these traditional variables, physical frameworks exist that explain why they are relevant: e.g. the formation of clouds during moist adiabatic ascent of air parcels (Rayleigh distillation framework, Dansgaard 1964) for cloud condensation temperatures and the Craig and Gordon 1965 ocean evaporation model for SST and the relative humidity with respect to SST.

**These five factors were the variables that emerged as primary influences on and/or strongly related with the water vapor isotopic variability in our analyses. Our phrasing is altered slightly throughout the manuscript to be more clear of this fact. As our study was based on the comparative analysis of high frequency field observations (isotopes, weather), our results will of course highlight factors drawn from those observations. We do not exclude the influence of additional factors that we did not personally have observations for (e.g., cloud condensation temp, moisture source conditions), and these factors are mentioned in our discussion of the five factors. For example, we discuss Rayleigh distillation in the temperature section and the local marine moisture and synoptic flow/NAO sections both focus heavily on how different environmental conditions at the moisture source produce changes in the water vapor isotopes.**

**To develop a database of SST and RH at the moisture source and cloud condensation dynamics at a high temporal resolution would require an extensive amount of modeling and computational effort. We do not dispute that such an analysis would be very intriguing and informative, but it falls outside the realm of our study and its focus on observational data. We have highlighted this concept in the implications and conclusions section as valuable potential follow up research. We have also added more context for the moisture sourcing discussion on SST and RH in section 4.2.**

To me highlighting the importance of the atmospheric circulation and at the same time underlining the relevance of local environmental conditions is somewhat contradictory. Many previous studies have used trajectory analysis to show the relevance of environmental conditions at the moisture source for the variability of stable water isotope measurements in water vapour (e.g. Pfahl and Wernli 2008; Aemisegger et al. 2014, Aemisegger 2018; Thurnherr et al. 2020). Here the authors say the circulation and the local conditions are key. I would find it useful, if there was a comment on this apparent contradiction in the paper. Or if

it is not a contradiction, then to resolve the misunderstanding and explain why the results of this paper are in agreement with these previous studies.

**We do not see a contradiction in our argument that both the synoptic circulation and local conditions play a role in determining the final isotopic signature observed at Thule. Moisture source conditions are indeed very important to determining the vapor isotopes at a location, particularly when determining the mean values over longer periods of weeks or seasons. We fully acknowledge this in our discussion and particularly in sections 4.2, 4.3, 5.1, and 5.6.**

**However, on shorter scales it is certainly well-documented that local variables, such as sea breezes (Kopec 2014, Bréant 2019), dew condensation (Bastrikov 2014), and vapor exchange (Casado 2016) play a minor to major role in further modifying the isotopic composition of a moisture parcel upon arrival, particularly for $\delta^{18}$O and $\delta^2$H even if *dxs* better preserves the original moisture source signal. Additionally, our data highlight that many of these local variables are important only for parts of the year (e.g., sea breezes in summer) and thus may not be well-captured as an isotopic control in studies that focus mainly on factors that consistently can explain isotopic variability throughout the year or at coarser time resolutions. While atmospheric circulation and initial moisture source conditions may be able to explain a significant portion of longer-term isotopic variability, we believe it is a worthwhile endeavor to consider whether local environmental variability can explain the 'noise' left unexplained by only considering atmospheric circulation as a control. At their core, most of these local factors are, in fact, moisture source and routing effects, but examined at a more localized scale than traditional moisture source back trajectory-based studies. For example, the sea breeze-katabatic isotopic dichotomy is based on two different moisture sources, but back trajectory analysis may fail to capture this and other important mesoscale climatology influences due to coarse global circulation models and reanalysis data.**

**Additionally, we feel that highlighting the influence of local factors helps illustrate how the isotopic signature from a moisture source/specific circulation gets expressed/enhanced/preserved at the local scale. As we display in Figure 5, the local factors are tightly integrated with atmospheric circulation, and while one could simplify the system by subsuming all the local factors into the initiating atmospheric circulation variable, we feel this does a disservice to understanding the inner mechanisms of a complex system.**

My third major comment is a more technical one: the presentation of the calibration and postprocessing framework of the exceptionally long and very valuable Arctic water vapor isotope time series lacks some details in particular on the total uncertainty of the measurements (see also my minor comments 5-11, below).

**We have attempted to address issues with the calibration and improve the transparency of our uncertainties and unknowns throughout. See specific comments below and in other reviewers for details.**

**Minor Comments**
P. 1, L. 1: The sea ice extent seems to come out as the most important factor controlling if moisture is mainly sourced from the local environment or if it is transported from further away. This could be mentioned more clearly in the abstract before the five controlling factors. In my opinion it comes a bit late in the current version.
**Abstract reworded to emphasize sea ice role earlier.**

P. 2, L. 50: There were many studies investigating the quality of laser spectrometric measurements in the early 2010s, add "e.g." and maybe Sturm and Knohl 2012 and Aemisegger et al. 2012 could be cited as well, since the latter study particularly focused on the capability of laser systems to resolve the synoptic timescale variability of water vapour isotopes.
**References added.**

P. 2, L.50: The Yale database could be cited here, since it groups most of the already published water vapour isotope data: Wei et al. 2019.
**Reference added.**

P. 3, L. 75 "critically giving a second set of observations to derive annual patterns and anomalies" not sure if I understand this correctly. What do the authors mean here?
**Changed to: "permitting comparative multi-year analysis of isotopic patterns and anomalies". We are highlighting that we can better determine what is typical and atypical isotopic variability since we have more than just a single year's observations.**

P. 5, L. 140: Did the authors test the response times of their system using "Tygon tubing". Several early studies (e.g. Sturm and Knohl 2010; Tremoy et al. 2011; Aemisegger et al. 2012) showed that certain tubing materials induce very large residence times and unwanted strong interactions between the tubing wall and the sample gas.
**We observed no clear evidence of tubing effects on our observed data. This is now clarified in the methods.**

6) P. 5, L. 140: What was the residence time of the sampled air in the tubing, how long was the tubing, was it heated, was the inlet shielded? These are all very important points for performing high quality stable water vapour isotope measurements especially in extreme environments such as in northern Greenland.
**Information added to methods. Also, more detailed information on the tubing set up was found and corrected in the manuscript.**

7) P. 5, L. 151: Introduce the delta notation and the normalisation to the international VSMOW-VSLAP scale.
**Included.**

8) P.6, L. 155: The standards' isotopic composition does not bracket the measured isotope signals. The authors should explicitly mention this and comment on the expected impact of this extrapolation on the total uncertainty of their measurements.
**This is now explicitly mentioned and commented on.**

9) P. 6, L. 175: Even though the drift of the Picarro laser spectrometers is limited regular calibrations should be carried out to 1) survey the good functioning of the system and 2) to provide a long term assessment of the total uncertainty of the measurements (see, Aemisegger et al. 2012; Thurnherr et al. 2020). In particular, Thurnherr et al. 2020 shows that different post processing procedures lead to substantial changes in the isotope data, in particular, with respect to the treatment of the water vapour mixing ratio dependent isotope bias correction.

**We acknowledge the clear benefits that regular calibrations bring and did not intend to come across as dismissing their importance. The initial operational plan did have twice daily calibrations, but unfortunately the Drierite system failed to adequately work and assembling the funding and logistics to install a dry air system at Thule took a long time. We have rephrased the section to hopefully be more clear that this is a source of unquantified possible uncertainty.**

10) P. 6, L. 181: The precision (Allan variance, or standard deviation of a constant water vapour isotope signal) strongly depends on the water vapour mixing ratio (see Aemisegger et al. 2012 and Sodemann et al. 2017). Please indicate the total uncertainty of the measurements as a function of water vapour mixing ratio. This is very important, given the very low levels of humidity observed at Thule in winter. **Estimates of precision from humidity response are now given in the manuscript and with more detail in S2 and Table S5.**

11) P. 7, L. 195: Was the water vapour mixing ratio of the L2130 calibrated using a dew point generator or another humidity sensor installed in parallel? Without calibration the reading of the laser spectrometric volume mixing ratio may be biased. **The mixing ratio of the L2130 compares very well with mixing ratio calculated from our weather station data. We have added additional information to show this and clarified the consistent factor difference between the L2130 mixing ratio and the weather station ratio (x1.23).**

12) P. 8, L. 225: Please indicated the horizontal and vertical grid resolution of the MERRA-2 reanalysis data. Note that HYSPLIT is not a model but a post-processing tool, thus it cannot be "forced". I would suggest to write: "… with air parcel backtrajectories calculated based on three-dimensional MERRA-2 wind fields…". I am not convinced that choosing only 10 days per months produces a robust two-year climatology. But given the limited use that is made of the trajectory climatology in this paper, the approach is ok. **Phrasing suggestions and grid resolution added. "Robust representation" changed to "acceptable representation" to signify that we were not attempting to perform an intense analysis with the trajectory results, but rather add exploratory context.**

13) P. 9, L. 254: "The magnitude of irregular hourly to weekly variations" do you mean synoptic timescale variations? **This section has been changed somewhat, but this particular sentence now makes it clear that we attribute these irregular hourly-weekly variations to synoptic events. "The water vapor isotopes also have substantial sub-seasonal variability, and the magnitude of irregular hourly to weekly variations caused by synoptic scale weather events can approach 30–50% of the entire annual isotopic ranges."**

14) P. 10, L. 291: What is meant by "temperature-driven equilibrium fractionation"? I think the cited literature is a bit misleading. Dütsch et al. 2017 shows that the condensation temperature indeed has a certain impact on the deuterium excess. Pfahl and Sodemann et al. 2014 discuss the effect of the SST. **Edited to have cited literature and statements more clear and in agreement.**

15) P. 10, L. 293: Interesting that a negative correlation between the deuterium excess and the temperature is found! Did the authors also look at the correlation with

nearby SSTs? In climate reconstructions based on ice cores a positive relation between the deuterium excess and moisture source SST is assumed (e.g. Johnsen et al. 1989; Vimeux et al. 1999; Stenni et al. 2001). However, in a detailed analysis of the correlation behaviour between the deuterium excess and SST a recent study (Aemisegger and Sjolte 2018) found different regions (in particular at high latitudes) that are expected to exhibit a negative correlation based on the Craig Gordon model and the closure assumption. It is thought that this negative correlation arises from a positive feedback mechanism between the SST and the relative humidity with respect to SST. Such a negative correlation regime is expected to be dominant particularly in regions where the variability in air-sea interactions is mainly driven by variability in atmospheric circulation and not primarily by variations in ocean circulation. If the deuterium excess also shows a negative correlation to the nearby SST at Thule this would be evidence for such a behaviour. Of course, the time series at Thule is too short to look at this in detail. But still, I find the negative correlation between air temperature and deuterium excess that is found here very interesting and since it is of opposite sign with respect to the traditional interpretation of the deuterium excess in ice core studies, I would find it worthwhile to shortly discuss this.

**This is an interesting and intriguing angle of study, but we did not compare our dxs with SST, because determining the exact moisture source and extracting a SST from that moisture source in a consistent, high temporal resolution manner across our entire record was beyond the bounds and abilities of our study. However, we would highlight for you our finding from Figure 6a, where the relationship between dxs and air temperature is opposite between summer and non-summer months. In summer, we do observe a strong positive correlation between air temperature and dxs while outside of summer, it has a strong negative correlation.**

**The interpretation of dxs in ice cores you mentioned largely pertains to reconstructing past SSTs, and not reconstructing past air temperature. At Thule, air temperature is likely a poor proxy for moisture source SST because the strong seasonal insolation variations and local topographic influences such as downsloping and katabatic winds. As a result, it is difficult to robustly link dxs-air temp correlations and patterns to dxs-SST interpretations with our data. We have included a reference to your proposal in our conclusion as a direction of possible future research.**

16) P. 11, L. 320: The relative humidity is very low above forming ice? Can the authors show some evidence or cite some literature?
**This comes from the cited Kurita 2011, but the sentence has been rephrased to be more clear about the origin of the dry air in these regions.**

17) P. 11, L. 323: Phase changes in the NAO might be confusing in the context of isotopes and water phase changes. Is there another way to formulate the change in the NAO sign?
**Phase is the most common and recognized term for this, and we struggled to come up with an alternative that was as clear. However, we acknowledge the potential for confusion with water phases, and some of the phrasing is altered to make it clear that it is the NAO phase being discussed.**

18) P. 12, L341-L356: Very interesting discussion on the role of the NAO and the sea ice extent!

19) P. 14, L. 394: I think the relations to the "traditional driving variables" SST and relative humidity with respect to the SST should at least be shortly mentioned.
**This is now mentioned earlier in the discussion.**

20) P. 14, L. 396 and 399: "southerly" flow instead of "southern" flow.
**Changed throughout manuscript.**

21) P. 15, Section 6.3 How large is the influence of more distant land sources of e.g. northern Canada? From Fig. 3 one might think that they may play a significant role.
**The moisture uptake from the land regions is generally light compared to that of open ocean and Baffin Bay. Additionally, moisture arriving at Thule from these land areas would have crossed substantial open water and potentially isotopically exchanged to a high degree. A focused follow up research project could attempt to key in on specific times where the dominant moisture transport to Thule was from Canadian land and look for anomalous isotopic responses (and in fact, some work related to this is close to submission by one of the co-authors). However, since our quasi-climatology was simply constructed to give broad context on moisture sourcing, we do not have such day-specific analytical data in our database.**

22) P. 18, Section 7: Additionally, I suspect that Arctic anticyclones would play a key role in the synoptic timescale variability at Thule (in Greenland blocking situations). A case study of an Arctic Blocking event (over northern Russia) is shown in Schneider et al. 2019, which highlights the very large horizontal gradients resulting from the subsidence induced drying at the core of the anticyclone and the progressive atmospheric moisture uptake along the anticyclone edges.
**Anticyclones now more prominently mentioned in synoptic circulation discussion.**

23) P. 18, L. 542: I doubt that in the case of sea ice much water vapour at Thule originates from the deep tropics or subtropics. The analysis shown in Fig. 3 does not support such a statement. Here I think one can safely write "advection from the midlatitudes".
**Edited.**

24) P. 21, L. 609: I don't understand why an input of low $d_{18}O$/high dxs from evaposublimation results in a $d_{18}O$ maximum and dxs minimum at midday.
**The latter max and min were erroneously switched in the text. It is in fact a $\delta^{18}O$ min and dxs max that occurs at midday, and this has been corrected in the text.**

25) P. 21, L. 627: It would be great to clearly mention that evaporation of meltwater and sublimation of snow might not carry the same isotopic composition (see e.g. Christner et al. 2017).
**This is now mentioned in the initial paragraph discussing evaposublimation.**

26) P. 22, L. 664: Maybe not single extreme events but years with high frequency of occurrence of warm advection events, e.g. due to a northward shift of the storm track?
**Edited to include this possibility.**

27) P. 23: In the conclusion it would be very nice to mention the great use of this dataset

for the validation of high-resolution isotope-enabled simulations in the Arctic to study the importance of air-ice and air-sea interaction processes in more detail. Here the great value of the data is their long temporal coverage, thanks to which modelbased sensitivity experiments could be performed to test the importance of different driving factors (similar to the climatological sensitivity study over Europe by Christner et al. 2018).
**Mentioned in the conclusion now.**

[revised manuscript text omitted]

**Figure 2.** Observed data from Thule, Greenland, of (a) water vapor δ¹⁸O, (b) water vapor *dxs*, (c) mean air temperature, (d) water vapor mixing ratio, (e) relative humidity, (f) wind speed, (g) wind azimuth, (h) station pressure, (i) NAO index (NOAA, 2019), and (j) Baffin Bay sea ice extent (Fetterer et al., 2010). The time series of δ²H is very similar to δ¹⁸O and not shown. Data shown are daily mean values with hourly values as lighter backdrop line for higher resolution variables (a–h). All observations were taken at the

925 SMT site except for wind speed and azimuth which were observed at the THU airport. Yellow dashed vertical lines indicate moisture pulse events and orange dotted vertical lines indicate the timing of sea ice breakup near Thule.

[revised manuscript text omitted]

Table 2. Spearman correlation values between isotopic and meteorological variables at Thule. Two sets of correlations are reported here: analyses from the original data and analyses from the seasonally-adjusted data where the seasonal cycle was removed for isotopic, air temperature, mixing ratio, and sea ice data. Seasonally-adjusting the data weakens nearly all correlations by removing the common response to seasonal change, but strengthens isotopic correlations with relative humidity. Correlations are given for all available levels of temporal aggregation, but some variables do not have data at finer resolutions. Temporal aggregation slightly strengthens correlations, but does not generally change the order of variables when ranking by correlation strength. Due to the large sample sizes, all correlations are significant at p < 0.05 except for a few with very weak correlations (italicized).

2005

| | $\rho$ (10 min) | | | $\rho$ (hourly) | | | $\rho$ (daily) | | |
|---|---|---|---|---|---|---|---|---|---|
| **Original** | $\delta^{18}O$ | $\delta^2H$ | $dxs$ | $\delta^{18}O$ | $\delta^2H$ | $dxs$ | $\delta^{18}O$ | $\delta^2H$ | $dxs$ |
| Air temperature | +0.76 | +0.73 | -0.74 | +0.76 | +0.73 | -0.75 | +0.81 | +0.77 | -0.79 |
| $H_2O$ mixing ratio | +0.83 | +0.80 | -0.79 | +0.83 | +0.80 | -0.79 | +0.86 | +0.83 | -0.82 |
| Relative humidity | +0.06 | +0.06 | -0.07 | +0.06 | +0.06 | -0.07 | *0.00* | *-0.01* | *-0.01* |
| Station pressure | +0.09 | +0.08 | -0.12 | +0.09 | +0.08 | -0.12 | +0.10 | +0.09 | -0.13 |
| Wind speed | - | - | - | -0.08 | -0.07 | +0.11 | -0.14 | -0.13 | +0.16 |
| Katabatic deviation | - | - | - | +0.50 | +0.48 | -0.50 | +0.48 | +0.47 | -0.41 |
| NAO | - | - | - | - | - | - | -0.25 | -0.23 | +0.29 |
| AO | - | - | - | - | - | - | *-0.06* | *-0.04* | +0.17 |
| Baffin Bay ice extent | - | - | - | - | - | - | -0.69 | -0.71 | +0.47 |
| | | | | | | | | | |
| **Seasonally-adjusted** | $\delta^{18}O$ | $\delta^2H$ | $dxs$ | $\delta^{18}O$ | $\delta^2H$ | $dxs$ | $\delta^{18}O$ | $\delta^2H$ | $dxs$ |
| Air temperature | +0.24 | +0.20 | -0.31 | +0.25 | +0.20 | -0.32 | +0.32 | +0.27 | -0.38 |
| $H_2O$ mixing ratio | +0.55 | +0.49 | -0.58 | +0.55 | +0.49 | -0.58 | +0.57 | +0.51 | -0.60 |

**Formatted** … [8]
**Formatted Table** … [9]
**Formatted** … [28]
**Formatted Table** … [29]
**Formatted** … [30]
**Formatted** … [31]
**Formatted** … [32]
**Formatted** … [33]
**Formatted** … [34]
**Formatted** … [35]
**Formatted** … [36]
**Formatted** … [37]
**Formatted** … [38]
**Formatted** … [39]
**Formatted** … [40]
**Formatted** … [41]
**Formatted** … [42]
**Formatted** … [43]
**Formatted** … [44]
**Formatted** … [45]
**Formatted** … [46]
**Formatted** … [47]
**Formatted** … [48]
**Formatted** … [49]
**Formatted** … [50]
**Formatted** … [51]
**Formatted** … [52]
**Formatted** … [53]
**Formatted** … [54]

| Relative humidity | +0.46 | +0.43 | -0.44 | +0.38 | +0.36 | -0.33 | +0.32 | +0.30 | -0.29 |
|---|---|---|---|---|---|---|---|---|---|
| Station pressure | +0.04 | +0.03 | -0.07 | +0.04 | +0.03 | -0.07 | *+0.03* | *+0.02* | -0.07 |
| Wind speed | - | - | - | *-0.03* | *-0.02* | +0.06 | *-0.03* | *-0.03* | *+0.06* |
| Katabatic deviation | - | - | - | +0.19 | +0.17 | -0.18 | +0.10 | +0.09 | *-0.03* |
| NAO | - | - | - | - | - | - | -0.14 | -0.12 | +0.20 |
| AO | - | - | - | - | - | - | *-0.04* | *-0.02* | +0.17 |
| Baffin Bay ice extent | - | - | - | - | - | - | -0.02 | -0.03 | +0.05 |

| Page 6: [1] Deleted | Microsoft Office User | 7/8/20 3:46:00 PM |
|---|---|---|

| Page 6: [2] Deleted | Microsoft Office User | 8/17/20 2:31:00 PM |
|---|---|---|

| Page 8: [3] Deleted | Microsoft Office User | 8/6/20 5:14:00 PM |
|---|---|---|

| Page 10: [4] Deleted | Microsoft Office User | 8/7/20 3:44:00 PM |
|---|---|---|

| Page 13: [5] Deleted | Microsoft Office User | 8/23/20 4:41:00 PM |
|---|---|---|

| Page 16: [6] Deleted | Microsoft Office User | 8/7/20 3:45:00 PM |
|---|---|---|

| Page 47: [7] Deleted | Microsoft Office User | 8/23/20 2:08:00 PM |
|---|---|---|

| Page 47: [7] Deleted | Microsoft Office User | 8/23/20 2:08:00 PM |
|---|---|---|

| Page 47: [8] Formatted | Microsoft Office User | 8/16/20 7:18:00 PM |
|---|---|---|

Line spacing:  Multiple 1.15 li

| Page 47: [9] Formatted Table | Microsoft Office User | 8/16/20 9:45:00 PM |
|---|---|---|

Formatted Table

| Page 47: [10] Deleted | Microsoft Office User | 8/16/20 7:17:00 PM |
|---|---|---|

| Page 47: [10] Deleted | Microsoft Office User | 8/16/20 7:17:00 PM |
|---|---|---|

| Page 47: [11] Deleted | Microsoft Office User | 8/23/20 2:28:00 PM |
|---|---|---|

| Page 47: [11] Deleted | Microsoft Office User | 8/23/20 2:28:00 PM |
|---|---|---|

| Page 47: [12] Deleted | Microsoft Office User | 8/23/20 2:28:00 PM |
|---|---|---|

| Page 47: [12] Deleted | Microsoft Office User | 8/23/20 2:28:00 PM |
|---|---|---|

| Page 47: [13] Deleted | Microsoft Office User | 8/23/20 2:28:00 PM |
|---|---|---|

| Page 47: [14] Deleted | Microsoft Office User | 8/23/20 2:28:00 PM |
|---|---|---|

| Page 47: [14] Deleted | Microsoft Office User | 8/23/20 2:28:00 PM |
|---|---|---|

| Page 47: [15] Deleted | Microsoft Office User | 8/23/20 2:29:00 PM |
|---|---|---|

| Page 47: [15] Deleted | Microsoft Office User | 8/23/20 2:29:00 PM |
|---|---|---|

| Page 47: [16] Deleted | Microsoft Office User | 8/23/20 2:29:00 PM |
|---|---|---|

| Page 47: [16] Deleted | Microsoft Office User | 8/23/20 2:29:00 PM |
|---|---|---|

| Page 47: [17] Deleted | Microsoft Office User | 8/23/20 2:38:00 PM |
|---|---|---|

| Page 47: [17] Deleted | Microsoft Office User | 8/23/20 2:38:00 PM |
|---|---|---|

| Page 47: [18] Deleted | Microsoft Office User | 8/23/20 2:40:00 PM |
|---|---|---|

| Page 47: [18] Deleted | Microsoft Office User | 8/23/20 2:40:00 PM |
|---|---|---|

| Page 47: [19] Deleted | Microsoft Office User | 8/23/20 2:40:00 PM |
|---|---|---|

| Page 47: [19] Deleted | Microsoft Office User | 8/23/20 2:40:00 PM |
|---|---|---|

| Page 47: [20] Deleted | Microsoft Office User | 8/23/20 2:40:00 PM |
|---|---|---|

| Page 47: [20] Deleted | Microsoft Office User | 8/23/20 2:40:00 PM |
|---|---|---|

| Page 47: [21] Deleted | Microsoft Office User | 8/23/20 2:41:00 PM |
|---|---|---|

| Page 47: [22] Deleted | Microsoft Office User | 8/23/20 2:41:00 PM |
|---|---|---|

| Page 47: [22] Deleted | Microsoft Office User | 8/23/20 2:41:00 PM |
|---|---|---|

| Page 47: [23] Deleted | Microsoft Office User | 8/23/20 2:42:00 PM |
|---|---|---|

| Page 47: [23] Deleted | Microsoft Office User | 8/23/20 2:42:00 PM |
|---|---|---|

| Page 47: [24] Deleted | Microsoft Office User | 8/23/20 2:42:00 PM |
|---|---|---|

| Page 47: [24] Deleted | Microsoft Office User | 8/23/20 2:42:00 PM |
|---|---|---|

| Page 47: [25] Deleted | Microsoft Office User | 8/23/20 2:44:00 PM |
|---|---|---|

| Page 47: [25] Deleted | Microsoft Office User | 8/23/20 2:44:00 PM |
|---|---|---|

| Page 47: [26] Deleted | Microsoft Office User | 8/23/20 2:44:00 PM |
|---|---|---|

| Page 47: [26] Deleted | Microsoft Office User | 8/23/20 2:44:00 PM |
|---|---|---|

| Page 47: [27] Deleted | Microsoft Office User | 8/23/20 2:45:00 PM |
|---|---|---|

| Page 47: [27] Deleted | Microsoft Office User | 8/23/20 2:45:00 PM |
|---|---|---|

| Page 47: [28] Formatted | Microsoft Office User | 8/19/20 3:44:00 PM |
|---|---|---|

Justified, Add space between paragraphs of the same style

| Page 47: [29] Formatted Table | Microsoft Office User | 8/19/20 3:42:00 PM |
|---|---|---|

Formatted Table

| Page 47: [30] Formatted | Microsoft Office User | 8/19/20 3:44:00 PM |
|---|---|---|

| Page 47: [31] Formatted | Microsoft Office User | 8/19/20 3:44:00 PM |
|---|---|---|

Justified, Add space between paragraphs of the same style

| Page 47: [32] Formatted | Microsoft Office User | 8/19/20 3:46:00 PM |
|---|---|---|

Font: Not Italic

| Page 47: [33] Formatted | Microsoft Office User | 8/19/20 3:44:00 PM |
|---|---|---|

Add space between paragraphs of the same style

| Page 47: [34] Formatted | Microsoft Office User | 8/19/20 3:46:00 PM |
|---|---|---|

Font: Not Italic

| Page 47: [35] Formatted | Microsoft Office User | 8/19/20 3:46:00 PM |
|---|---|---|

Font: Not Italic

| Page 47: [35] Formatted | Microsoft Office User | 8/19/20 3:46:00 PM |
|---|---|---|

Font: Not Italic

| Page 47: [36] Formatted | Microsoft Office User | 8/19/20 3:46:00 PM |
|---|---|---|

Font: Italic

| Page 47: [37] Formatted | Microsoft Office User | 8/19/20 3:46:00 PM |
|---|---|---|

Font: Not Italic

| Page 47: [37] Formatted | Microsoft Office User | 8/19/20 3:46:00 PM |
|---|---|---|

Font: Not Italic

| Page 47: [38] Formatted | Microsoft Office User | 8/19/20 3:46:00 PM |
|---|---|---|

Font: Not Italic

| Page 47: [38] Formatted | Microsoft Office User | 8/19/20 3:46:00 PM |
|---|---|---|

Font: Not Italic

| Page 47: [39] Formatted | Microsoft Office User | 8/19/20 3:46:00 PM |
|---|---|---|

Font: Italic

| Page 47: [40] Formatted | Microsoft Office User | 8/19/20 3:46:00 PM |
|---|---|---|

Font: Not Italic

| Page 47: [40] Formatted | Microsoft Office User | 8/19/20 3:46:00 PM |
|---|---|---|

| Page 47: [41] Formatted | Microsoft Office User | 8/19/20 3:46:00 PM |
|---|---|---|

Font: Not Italic

| Page 47: [41] Formatted | Microsoft Office User | 8/19/20 3:46:00 PM |
|---|---|---|

Font: Not Italic

| Page 47: [42] Formatted | Microsoft Office User | 8/19/20 3:46:00 PM |
|---|---|---|

Font: Italic

| Page 47: [43] Formatted | Microsoft Office User | 8/19/20 3:44:00 PM |
|---|---|---|

Justified, Add space between paragraphs of the same style

| Page 47: [44] Formatted | Microsoft Office User | 8/19/20 3:44:00 PM |
|---|---|---|

Add space between paragraphs of the same style

| Page 47: [45] Formatted | Microsoft Office User | 8/19/20 3:45:00 PM |
|---|---|---|

Font: Not Italic

| Page 47: [46] Formatted | Microsoft Office User | 8/19/20 3:44:00 PM |
|---|---|---|

Justified, Add space between paragraphs of the same style

| Page 47: [47] Formatted | Microsoft Office User | 8/19/20 3:44:00 PM |
|---|---|---|

Add space between paragraphs of the same style

| Page 47: [48] Formatted | Microsoft Office User | 8/19/20 3:45:00 PM |
|---|---|---|

Font: Not Italic

| Page 47: [49] Formatted | Microsoft Office User | 8/19/20 3:44:00 PM |
|---|---|---|

Justified, Add space between paragraphs of the same style

| Page 47: [50] Formatted | Microsoft Office User | 8/19/20 3:44:00 PM |
|---|---|---|

Add space between paragraphs of the same style

| Page 47: [51] Formatted | Microsoft Office User | 8/19/20 3:45:00 PM |
|---|---|---|

Font: Not Italic

| Page 47: [52] Formatted | Microsoft Office User | 8/19/20 3:44:00 PM |
|---|---|---|

Justified, Add space between paragraphs of the same style

| Page 47: [53] Formatted | Microsoft Office User | 8/19/20 3:44:00 PM |
|---|---|---|

| Page 47: [54] Formatted | Microsoft Office User | 8/19/20 3:45:00 PM |
|---|---|---|

Font: Not Italic

| Page 47: [55] Formatted | Microsoft Office User | 8/19/20 3:44:00 PM |
|---|---|---|

Justified, Add space between paragraphs of the same style

| Page 47: [56] Formatted | Microsoft Office User | 8/19/20 3:44:00 PM |
|---|---|---|

Add space between paragraphs of the same style

| Page 47: [57] Formatted | Microsoft Office User | 8/19/20 3:45:00 PM |
|---|---|---|

Font: Not Italic

| Page 47: [58] Formatted | Microsoft Office User | 8/19/20 3:44:00 PM |
|---|---|---|

Justified, Add space between paragraphs of the same style

| Page 47: [59] Formatted | Microsoft Office User | 8/19/20 3:44:00 PM |
|---|---|---|

Add space between paragraphs of the same style

| Page 47: [60] Formatted | Microsoft Office User | 8/19/20 3:45:00 PM |
|---|---|---|

Font: Not Italic

| Page 47: [61] Formatted | Microsoft Office User | 8/19/20 3:44:00 PM |
|---|---|---|

Justified, Add space between paragraphs of the same style

| Page 47: [62] Formatted | Microsoft Office User | 8/19/20 3:44:00 PM |
|---|---|---|

Add space between paragraphs of the same style

| Page 47: [63] Formatted | Microsoft Office User | 8/19/20 3:45:00 PM |
|---|---|---|

Font: Not Italic

| Page 47: [64] Formatted | Microsoft Office User | 8/19/20 3:44:00 PM |
|---|---|---|

Justified, Add space between paragraphs of the same style

| Page 47: [65] Formatted | Microsoft Office User | 8/19/20 3:44:00 PM |
|---|---|---|

Add space between paragraphs of the same style

| Page 47: [66] Formatted | Microsoft Office User | 8/19/20 3:45:00 PM |
|---|---|---|

Font: Not Italic

| Page 47: [67] Formatted | Microsoft Office User | 8/19/20 3:44:00 PM |
|---|---|---|

| Page 47: [68] Formatted | Microsoft Office User | 8/19/20 3:44:00 PM |
|---|---|---|

Add space between paragraphs of the same style

| Page 47: [69] Formatted | Microsoft Office User | 8/19/20 3:44:00 PM |
|---|---|---|

Justified, Add space between paragraphs of the same style

| Page 47: [70] Formatted | Microsoft Office User | 8/19/20 3:44:00 PM |
|---|---|---|

Add space between paragraphs of the same style

| Page 47: [71] Formatted | Microsoft Office User | 8/19/20 3:44:00 PM |
|---|---|---|

Justified, Add space between paragraphs of the same style

| Page 47: [72] Formatted | Microsoft Office User | 8/19/20 3:46:00 PM |
|---|---|---|

Font: Not Italic

| Page 47: [73] Formatted | Microsoft Office User | 8/19/20 3:44:00 PM |
|---|---|---|

Add space between paragraphs of the same style

| Page 47: [74] Formatted | Microsoft Office User | 8/19/20 3:46:00 PM |
|---|---|---|

Font: Not Italic

| Page 47: [75] Formatted | Microsoft Office User | 8/19/20 3:46:00 PM |
|---|---|---|

Font: Not Italic

| Page 47: [75] Formatted | Microsoft Office User | 8/19/20 3:46:00 PM |
|---|---|---|

Font: Not Italic

| Page 47: [76] Formatted | Microsoft Office User | 8/19/20 3:46:00 PM |
|---|---|---|

Font: Not Italic

| Page 47: [76] Formatted | Microsoft Office User | 8/19/20 3:46:00 PM |
|---|---|---|

Font: Not Italic

| Page 47: [77] Formatted | Microsoft Office User | 8/19/20 3:46:00 PM |
|---|---|---|

Font: Not Italic

| Page 47: [77] Formatted | Microsoft Office User | 8/19/20 3:46:00 PM |
|---|---|---|

Font: Not Italic

| Page 47: [78] Formatted | Microsoft Office User | 8/19/20 3:46:00 PM |
|---|---|---|

| Page 47: [78] Formatted | Microsoft Office User | 8/19/20 3:46:00 PM |
|---|---|---|

Font: Not Italic

| Page 47: [79] Formatted | Microsoft Office User | 8/19/20 3:46:00 PM |
|---|---|---|

Font: Not Italic

| Page 47: [79] Formatted | Microsoft Office User | 8/19/20 3:46:00 PM |
|---|---|---|

Font: Not Italic

| Page 47: [80] Formatted | Microsoft Office User | 8/19/20 3:45:00 PM |
|---|---|---|

Font: Not Italic

| Page 47: [81] Formatted | Microsoft Office User | 8/19/20 3:44:00 PM |
|---|---|---|

Justified, Add space between paragraphs of the same style

| Page 47: [82] Formatted | Microsoft Office User | 8/19/20 3:44:00 PM |
|---|---|---|

Add space between paragraphs of the same style

| Page 47: [83] Formatted | Microsoft Office User | 8/19/20 3:45:00 PM |
|---|---|---|

Font: Not Italic

| Page 47: [84] Formatted | Microsoft Office User | 8/19/20 3:44:00 PM |
|---|---|---|

Justified, Add space between paragraphs of the same style

| Page 47: [85] Formatted | Microsoft Office User | 8/19/20 3:44:00 PM |
|---|---|---|

Add space between paragraphs of the same style

**S1 Quality checking and instrument stability issues**

The raw data from the L2130-i was passed through a series of quality checks prior to humidity response correction to remove readings that were well outside typical values observed at Thule. Most of these erroneous readings appear due to liquid water contamination of the intake tubing, likely from precipitation during intense cyclones. These quality checks identified and removed data where:

- L2130-i diagnostic data (e.g., chamber temperature, pressure, status, etc.) were unstable or out of typical ranges
- Water vapor mixing ratio >15000 ppmv
- $\delta^{18}O$ values >-15‰ AND *dxs* values <-20‰
- Standard deviation for five minute aggregates of $\delta^{18}O$ >8‰

An additional quality check was performed after humidity response calibration. Visual inspection was used to identify clearly abnormal isotopic or mixing ratio values (e.g., very large and/or abrupt changes not supported by meteorological data), and these observations were removed.

The machine was initially used at a tundra-based field site in Thule for a summer project in 2015, and it was installed its present location in October 2016. However, there were issues with cavity pressure stability and irregular isotopic readings which culminated in a full systems crash in May 2017. The system was restored on 04 Aug 2017 with stable cavity pressure that has continued through present. Data from before the system restoration has poor correlation in water vapor mixing ratio between the L2130-i and the SMT weather station. Winter isotopic values and mixing ratios are also much higher in the pre-restoration data than the next two winters despite generally similar winter weather. Out of caution, we have restricted our analyses and discussion to only post-restoration data.

**S2 Humidity response calibrations**

To correct for $\delta^{18}O$ and $\delta^{2}H$ accuracy and precision bias at low water vapor mixing ratios, we injected standard waters for ten minutes at ten different flow rates. The last 200 observations of each injection were saved, and a nonlinear regression was performed on the δX vs. mixing ratio relationship, where δX is either $\delta^{18}O$ or $\delta^{2}H$ to determine accuracy corrections. The nonlinear regression was of the form:

$$\delta X_{correction} = a + \frac{b}{q} \tag{S1}$$

where $\delta X_{correction}$ is the difference between the observed isotopic value and the actual standard isotopic value, *q* is the water vapor mixing ratio, and *a* and *b* are constants. Calculated values for regression parameters are given in Table S4. Confidence

40  intervals for predicted humidity response corrections were estimated using the predictNLS function from the *propagate* package in R.

Changes in analytical precision at low water vapor mixing ratios were calculated with a nonlinear regression of the form:

$$\delta X_{precision} = a + \frac{b}{q} \qquad\qquad (S2)$$

where $\delta X_{precision}$ is the standard error of the mean isotopic value for a given flow rate, $q$ is the water vapor mixing ratio, and $a$

45  and $b$ are constants. Calculated values for regression parameters are given in Table S5.

**S3 Isotope–isotope relationships**

Over the full dataset, $\delta^{18}O$ and $\delta^{2}H$ have a strong linear relationship with low parameter standard error: $\delta^{2}H = 6.959 \pm 0.003$ * $\delta^{18}O$ - $18.07 \pm 0.09‰$ ($r^2 = 0.98$, n=111138, 10 min data). Overall, this value is comparable to other slopes observed at other high latitude sites, such as 6.8 at Ivittuut, Greenland, (Bonne et al., 2014), 6.5 at NEEM, Greenland, (Steen-Larsen et al., 2013),

50  6.0–6.5 at Dome C, Antarctica (Casado et al., 2016), and 6.95 from the vapor mixing line at Kangerlussuaq, Greenland (Kopec et al., 2014). Changes in $\delta^{18}O$ are thus closely mirrored in $\delta^{2}H$, and most differences are only detectable on very short timescales (i.e., less than hourly) when some minor lead-lag between relative maxima and minima may occur. The *dxs* at Thule is negatively correlated with both $\delta^{18}O$ and $\delta^{2}H$ ($\rho_{10min}$ = -0.78 and -0.70, respectively).

[Figure]

**Figure S1. Results of the humidity response calibrations for $\delta^{18}O$ (a, c) and $\delta^2H$ (b, d) for two standard waters (USGS 45: top row and USGS 46: bottom row). Points show individual 1 $s^{-1}$ observations, while solid lines show the nonlinear regression of the data (blue: $\delta^{18}O$, pink: $\delta^2H$). Dashed horizontal lines show the actual isotopic value of the standard waters.**

[Figure]

**Fig S2: Comparison between time series of (a) δ¹⁸O and (b) *dxs* with NAO index, after all data have been differenced by one day.**

[Figure]

**Figure S3.** Water vapor $\delta^{18}O$ (top row; a–c) and *dxs* (bottom row; d–f) compared to meteorological variables for the period April through September. Data are split by whether the wind regime was katabatic (azimuth > 40° & < 180°) or sea breeze (azimuth > 240° & < 360°). A generalized additive smoothing model with 95% confidence intervals is overlaid to show trends of the mean values for each wind regime.

**28 April 2019       06 May 2019**

**Figure S4: Satellite imagery (top row; a, b) and sea ice concentration (bottom row; c, d) illustrating sea ice conditions before (left column; a, c) and after (right column; b, d) the late spring shift in isotopic and meteorological values at Thule in 2019. The images illustrate the opening of local oceans and snowpack melt as a result of local sea ice breakup. Satellite imagery provided by MODIS (Hall and Riggs, 2015) and sea ice concentration by the NSIDC (Fetterer et al., 2017).**

[Figure]

90 **Figure S5. Relationship between air temperature and water vapor mixing ratio (here, log-transformed), illustrating the decoupling that occurs above 5°C (vertical line). Data are split by whether the wind regime was katabatic (azimuth > 40° & < 180°) or sea breeze (azimuth > 240° & < 360°). A generalized additive smoothing model with 95% confidence intervals is overlaid to show trends of the mean values for each wind regime. The relationship decoupling exists in the data from both wind regimes.**

[Figure]

**Figure S6.** Comparison of isotopic and meteorological data during the period 18–26 Sep for 2017 (left) and 2018 (right). A diel cycle is observed in air temperature, relative humidity, and wind speed in both years, but in the isotopes and mixing ratio only for the year 2017. Y-axes are scaled the same for both years with the exception of δ¹⁸O. Dashed gray vertical lines indicate the time of daily thermal max from 20–25 Sep. The 2018 example, when no little to no isotopic cycling is observed, is representative of nearly all other autumn periods at Thule when diel cycles in temperature and relative humidity can be clearly observed in the general time series data.

Table S1. High latitude stationary sites with continuous observations of water vapor isotopes longer than two weeks reported in scientific literature.

| Site | Coordinates | Elevation | Setting | Period of record | Citation |
|---|---|---|---|---|---|
| Thule, NW Greenland | 76.51°N, 68.74°W | 229 m a.s.l. | Coastal | 08/2017 to 08/2019 | This study |
| Ivittuut, S Greenland | 61.21°N, 48.17°W | 30 m a.s.l. | Coastal | 09/2011 to 05/2013 | (Bonne et al., 2014) |
| Kangerlussuaq, W Greenland | 67.02°N, 50.69°W | 49 m a.s.l | Coastal | 07/2011 to 08/2011 | (Kopec et al., 2014) |
| NEEM, NW Greenland | 77.45°N, 51.05°W | 2484 m a.s.l. | Ice sheet | 05/2010 to 07/2010; 07/2011 to 08/2011; 05/2012 to 08/2012 | (Steen-Larsen et al., 2013; Steen-Larsen et al., 2014) |
| Summit, Greenland | 72.58°N, 38.46°W | 3216 m a.s.l. | Ice sheet | Summer 2011 to summer 2014 | (Bailey et al., 2015) |
| Selvogsviti, Iceland | 63.83°N, 21.47°W | 0 m a.s.l. | Coastal | 11/2011 to 04/2013 | (Steen-Larsen et al., 2015) |
| Kourovka, W Siberia | 57.04°N, 59.55°E | 300 m a.s.l. | Inland | 09/2012 to 08/2013 | (Bastrikov et al., 2014) |
| Samoylov Island, E Siberia | 72.37°N, 126.48°E | 0 m a.s.l. | Coastal | 07/2015 to 06/2017 | (Bonne et al., 2020) |
| Toolik Lake, N Alaska | 68.63°N, 149.60°W | 760 m a.s.l. | Inland | 05/2013 to 08/2013 | (Klein et al., 2015) |
| Dumont d'Urville, E Antarctica | 66.65°S, 140.00°E | 10 m a.s.l. | Coastal | 12/2016 to 02/2017 | (Bréant et al., 2019) |
| Syowa, E Antarctica | 69.00°S, 39.58°E | 0 m a.s.l. | Coastal | 12/2013 to 02/2014; 12/2014 to 02/2015 | (Kurita et al., 2016) |
| Kohnen, E Antarctica | 75.00°S, 0.07°E | 2892 m a.s.l | Ice sheet | 12/2013 to 01/2014 | (Ritter et al., 2016) |
| Dome C, E Antarctica | 75.10°S, 123.39°E | 3233 m a.s.l | Ice sheet | 12/2014 to 01/2015 | (Casado et al., 2016) |

Table S2. Standard waters calibration results after dry air system installation, performed roughly every 25 hours. Mean values and standard deviations of the last 200 observations for each calibration are given here. Some days do not have data due to failed calibration from a clogged injection needle or stuck injection piston. Some of the calibrations included here extend beyond the limit of ambient data discussed in the study, but station operation has continued without interruption.

| | USGS 45 | | | | | | USGS 46 | | | | | |
| | Mixing ratio (ppmv) | | δ$^{18}$O (‰) | | δ$^2$H (‰) | | Mixing ratio (ppmv) | | δ$^{18}$O (‰) | | δ$^2$H (‰) | |
| Date | Mean | SD | Mean | SD | Mean | SD | Mean | SD | Mean | SD | Mean | SD |
|---|---|---|---|---|---|---|---|---|---|---|---|---|
| 2019-08-01 | 5367 | 20 | -0.8 | 0.6 | -11 | 4 | 5325 | 14 | -28.8 | 0.6 | -241 | 4 |
| 2019-08-02 | 5378 | 16 | -0.9 | 0.6 | -9 | 4 | 5345 | 10 | -28.9 | 0.6 | -239 | 4 |
| 2019-08-04 | 5412 | 19 | -0.7 | 0.7 | -10 | 4 | 5396 | 14 | -28.7 | 0.6 | -240 | 4 |
| 2019-08-05 | 5421 | 18 | -0.8 | 0.6 | -10 | 4 | 5406 | 14 | -28.8 | 0.6 | -241 | 4 |
| 2019-08-06 | 5450 | 15 | -0.7 | 0.6 | -10 | 4 | 5446 | 17 | -28.7 | 0.6 | -240 | 4 |
| 2019-08-07 | 5459 | 15 | -0.6 | 0.6 | -10 | 4 | 5430 | 18 | -28.6 | 0.6 | -240 | 4 |
| 2019-08-09 | 5480 | 16 | -0.7 | 0.6 | -10 | 4 | 5431 | 17 | -28.8 | 0.6 | -240 | 4 |
| 2019-08-10 | 5438 | 12 | -0.9 | 0.6 | -10 | 4 | 5448 | 19 | -28.7 | 0.6 | -239 | 4 |
| 2019-08-11 | 5467 | 15 | -0.8 | 0.6 | -10 | 4 | 5476 | 59 | -28.5 | 0.6 | -239 | 4 |
| 2019-08-12 | 5392 | 57 | -0.8 | 0.7 | -10 | 4 | 5531 | 91 | -28.5 | 0.7 | -240 | 4 |

Moved down [1]: Table S2. Parameters for nonlinear
Formatted ... [3]
Formatted ... [4]
Formatted Table ... [2]
Formatted ... [5]
Formatted ... [17]
Formatted ... [18]
Formatted ... [6]
Formatted ... [7]
Formatted ... [8]
Formatted ... [10]
Formatted ... [11]
Formatted ... [12]
Formatted ... [14]
Formatted ... [9]
Formatted ... [13]
Formatted ... [15]
Formatted ... [16]
Formatted ... [19]
Formatted ... [20]
Formatted ... [21]
Formatted ... [22]
Formatted ... [23]
Formatted ... [24]
Formatted ... [25]
Formatted ... [26]
Formatted ... [27]
Formatted ... [28]

| | | | | | | | | | | | |
|---|---|---|---|---|---|---|---|---|---|---|---|
| 2019-08-14 | 5520 | 20 | -0.7 | 0.6 | -9 | 4 | 5524 | 18 | -28.6 | 0.7 | -239 |
| 2019-08-15 | 5523 | 13 | -0.8 | 0.6 | -10 | 4 | 5528 | 21 | -28.8 | 0.6 | -240 |
| 2019-08-16 | 5574 | 19 | -0.6 | 0.6 | -10 | 4 | 5568 | 20 | -28.5 | 0.6 | -240 |
| 2019-08-17 | 5599 | 31 | -0.6 | 0.6 | -9 | 4 | 5535 | 38 | -28.4 | 0.6 | -239 |
| 2019-08-18 | 5492 | 38 | -0.5 | 0.7 | -9 | 4 | 5545 | 24 | -28.8 | 0.7 | -240 |
| 2019-08-19 | 5478 | 61 | -0.9 | 0.6 | -10 | 4 | 5573 | 20 | -28.6 | 0.6 | -239 |
| 2019-08-20 | 5610 | 18 | -0.5 | 0.6 | -9 | 4 | 5544 | 79 | -28.8 | 0.6 | -240 |
| 2019-08-22 | 5488 | 43 | -1.1 | 0.6 | -11 | 4 | 5564 | 15 | -28.6 | 0.6 | -240 |
| 2019-08-23 | 5605 | 13 | -0.6 | 0.6 | -9 | 4 | 5622 | 18 | -28.6 | 0.6 | -240 |
| 2019-08-24 | 5566 | 17 | -0.9 | 0.6 | -10 | 4 | 5591 | 17 | -28.6 | 0.6 | -240 |
| 2019-08-25 | 5642 | 9 | -0.5 | 0.5 | -10 | 4 | 5602 | 38 | -28.3 | 0.7 | -239 |
| 2019-08-26 | 5519 | 32 | -1.0 | 0.6 | -11 | 4 | 5641 | 19 | -28.6 | 0.6 | -240 |
| 2019-08-27 | 5674 | 18 | -0.5 | 0.6 | -9 | 4 | 5620 | 24 | -28.6 | 0.6 | -240 |
| 2019-08-28 | 5616 | 14 | -0.5 | 0.6 | -10 | 4 | 5602 | 64 | -28.8 | 0.6 | -241 |
| 2019-08-29 | 5642 | 13 | -0.5 | 0.6 | -9 | 4 | 5449 | 104 | -29.1 | 0.7 | -241 |
| 2019-08-31 | 5632 | 11 | 0.0 | 0.7 | -9 | 4 | 5655 | 18 | -28.7 | 0.7 | -241 |
| 2019-09-01 | 5719 | 14 | -0.6 | 0.7 | -10 | 4 | 5668 | 17 | -28.8 | 0.6 | -240 |
| 2019-09-02 | 5727 | 29 | -0.7 | 0.6 | -10 | 4 | 5739 | 15 | -28.6 | 0.6 | -240 |
| 2019-09-03 | 5628 | 25 | -0.5 | 0.7 | -9 | 4 | 5671 | 20 | -28.5 | 0.6 | -241 |
| 2019-09-05 | 5687 | 27 | -0.7 | 0.6 | -10 | 4 | 5744 | 18 | -28.5 | 0.6 | -240 |
| 2019-09-06 | 5787 | 17 | -0.5 | 0.6 | -10 | 4 | 5734 | 21 | -28.6 | 0.6 | -241 |
| 2019-09-08 | 5769 | 15 | -0.2 | 0.7 | -9 | 4 | 5798 | 21 | -28.6 | 0.6 | -240 |
| 2019-09-09 | 5739 | 23 | -0.3 | 0.7 | -9 | 4 | 5762 | 20 | -28.4 | 0.6 | -241 |
| 2019-09-10 | 5729 | 13 | -0.5 | 0.6 | -9 | 4 | 5714 | 43 | -28.6 | 0.6 | -240 |
| 2019-09-11 | 5810 | 9 | -0.5 | 0.6 | -9 | 4 | 5749 | 19 | -28.5 | 0.6 | -240 |
| 2019-09-12 | 5737 | 20 | -0.4 | 0.7 | -9 | 4 | 5787 | 57 | -28.4 | 0.6 | -240 |
| 2019-09-13 | 5793 | 29 | -0.9 | 0.6 | -10 | 4 | 5788 | 28 | -28.1 | 0.6 | -239 |
| 2019-09-14 | 5871 | 17 | -0.1 | 0.7 | -8 | 4 | 5837 | 27 | -28.4 | 0.6 | -240 |
| 2019-09-16 | 5689 | 12 | -0.6 | 0.6 | -10 | 4 | 5852 | 24 | -28.5 | 0.6 | -240 |
| 2019-09-17 | 5723 | 37 | -0.8 | 0.7 | -10 | 4 | 5817 | 21 | -28.3 | 0.7 | -239 |
| 2019-09-18 | 5855 | 20 | -0.3 | 0.6 | -9 | 3 | 5901 | 35 | -28.4 | 0.6 | -240 |
| 2019-09-19 | 5864 | 11 | -0.5 | 0.6 | -9 | 4 | 5811 | 30 | -28.7 | 0.7 | -241 |
| 2019-09-20 | 5948 | 10 | -0.5 | 0.6 | -10 | 4 | 5966 | 14 | -28.7 | 0.6 | -241 |
| 2019-09-22 | 5942 | 12 | -0.5 | 0.6 | -9 | 4 | 5938 | 11 | -28.6 | 0.6 | -241 |
| 2019-09-24 | 5999 | 35 | -0.4 | 0.6 | -9 | 4 | 6009 | 17 | -28.5 | 0.6 | -240 |

**Formatted** ... [137]

**Formatted** ... [144]

**Formatted** ... [136]

**Formatted** ... [138]

**Formatted** ... [139]

**Formatted** ... [140]

**Formatted** ... [141]

**Formatted** ... [142]

**Formatted** ... [146]

**Formatted** ... [147]

**Formatted** ... [143]

**Formatted** ... [145]

**Formatted** ... [148]

**Formatted** ... [152]

**Formatted** ... [153]

**Formatted** ... [155]

**Formatted** ... [160]

**Formatted** ... [161]

**Formatted** ... [149]

**Formatted** ... [150]

**Formatted** ... [151]

**Formatted** ... [156]

**Formatted** ... [157]

**Formatted** ... [154]

**Formatted** ... [158]

**Formatted** ... [159]

**Formatted** ... [162]

| 2019-09-25 | 5933 | 7 | -0.4 | 0.6 | -9 | 4 | | 5929 | 15 | -28.6 | 0.6 | -240 |
|---|---|---|---|---|---|---|---|---|---|---|---|---|
| 2019-09-26 | 5887 | 15 | -0.6 | 0.7 | -9 | 4 | | 5967 | 18 | -28.4 | 0.6 | -239 |
| 2019-09-27 | 5792 | 17 | -0.9 | 0.6 | -9 | 4 | | 5855 | 21 | -29.2 | 0.7 | -242 |
| 2019-09-28 | 5887 | 13 | -0.4 | 0.6 | -9 | 4 | | 5880 | 13 | -28.5 | 0.6 | -240 |
| 2019-09-29 | 5832 | 21 | -0.3 | 0.5 | -10 | 4 | | 5866 | 16 | -28.7 | 0.6 | -241 |
| 2019-09-30 | 5815 | 32 | 0.1 | 0.6 | -8 | 4 | | 5837 | 19 | -28.5 | 0.6 | -241 |

245

**Table S3. Isotopic value trends for standard water calibrations performed after dry air system installation (Table S3). Trends are calculated as the slope of the last 200 observations taken during a calibration versus time, with one observation per second. The mean and standard deviation of all calibration runs is given at the bottom of the table.**

| | USGS 45 | | USGS 46 | |
|---|---|---|---|---|
| Date | $\delta^{18}O$ (‰ s$^{-1}$) | $\delta^{2}H$ (‰ s$^{-1}$) | $\delta^{18}O$ (‰ s$^{-1}$) | $\delta^{2}H$ (‰ s$^{-1}$) |
| 2019-08-01 | 0.0000 | 0.0047 | 0.0006 | -0.0043 |
| 2019-08-02 | 0.0004 | 0.0098 | 0.0001 | 0.0056 |
| 2019-08-04 | 0.0019 | 0.0036 | 0.0001 | -0.0064 |
| 2019-08-05 | 0.0021 | 0.0008 | 0.0021 | 0.0037 |
| 2019-08-06 | 0.0015 | -0.0063 | 0.0011 | 0.0089 |
| 2019-08-07 | 0.0008 | -0.0014 | 0.0009 | 0.0004 |
| 2019-08-09 | 0.0014 | 0.0089 | -0.0006 | -0.0015 |
| 2019-08-10 | 0.0014 | 0.0016 | 0.0001 | 0.0018 |
| 2019-08-11 | -0.0002 | 0.0033 | 0.0004 | 0.0061 |
| 2019-08-12 | -0.0032 | 0.0059 | -0.0003 | -0.0030 |
| 2019-08-14 | 0.0013 | 0.0055 | -0.0001 | -0.0027 |
| 2019-08-15 | -0.0001 | -0.0106 | 0.0030 | 0.0096 |
| 2019-08-16 | 0.0006 | 0.0052 | 0.0050 | 0.0064 |
| 2019-08-17 | 0.0007 | 0.0038 | 0.0015 | -0.0016 |
| 2019-08-18 | 0.0028 | 0.0040 | 0.0039 | 0.0110 |
| 2019-08-19 | 0.0018 | 0.0122 | 0.0012 | 0.0085 |
| 2019-08-20 | 0.0001 | 0.0112 | 0.0008 | 0.0045 |
| 2019-08-22 | 0.0011 | 0.0115 | 0.0019 | -0.0067 |
| 2019-08-23 | -0.0014 | 0.0010 | 0.0014 | 0.0042 |
| 2019-08-24 | -0.0005 | 0.0079 | 0.0002 | -0.0020 |
| 2019-08-25 | -0.0002 | 0.0122 | 0.0049 | 0.0058 |

| Date | | | | |
|------|--------|--------|---------|---------|
| 2019-08-26 | -0.0023 | 0.0001 | -0.0002 | 0.0041 |
| 2019-08-27 | 0.0015 | 0.0033 | 0.0010 | 0.0060 |
| 2019-08-28 | -0.0009 | 0.0078 | -0.0023 | -0.0082 |
| 2019-08-29 | -0.0003 | 0.0017 | 0.0009 | -0.0018 |
| 2019-08-31 | -0.0026 | -0.0083 | 0.0035 | 0.0009 |
| 2019-09-01 | -0.0031 | 0.0004 | 0.0017 | 0.0037 |
| 2019-09-02 | 0.0009 | 0.0113 | 0.0022 | -0.0043 |
| 2019-09-03 | 0.0011 | -0.0011 | 0.0006 | 0.0021 |
| 2019-09-05 | -0.0011 | 0.0021 | 0.0026 | 0.0085 |
| 2019-09-06 | -0.0002 | -0.0034 | 0.0019 | 0.0014 |
| 2019-09-08 | -0.0014 | 0.0030 | 0.0025 | -0.0038 |
| 2019-09-09 | -0.0014 | -0.0119 | 0.0018 | 0.0101 |
| 2019-09-10 | 0.0002 | -0.0063 | 0.0018 | -0.0027 |
| 2019-09-11 | -0.0017 | -0.0009 | 0.0034 | 0.0017 |
| 2019-09-12 | -0.0013 | 0.0051 | 0.0005 | 0.0064 |
| 2019-09-13 | -0.0021 | -0.0136 | 0.0022 | 0.0015 |
| 2019-09-14 | -0.0018 | 0.0093 | 0.0017 | 0.0103 |
| 2019-09-16 | -0.0013 | -0.0036 | 0.0012 | 0.0079 |
| 2019-09-17 | -0.0036 | 0.0061 | 0.0046 | 0.0107 |
| 2019-09-18 | -0.0014 | 0.0007 | 0.0017 | 0.0033 |
| 2019-09-19 | -0.0018 | -0.0035 | 0.0066 | 0.0243 |
| 2019-09-20 | 0.0037 | 0.0074 | -0.0007 | -0.0059 |
| 2019-09-22 | -0.0025 | -0.0006 | -0.0004 | -0.0009 |
| 2019-09-24 | 0.0000 | 0.0086 | 0.0015 | 0.0040 |
| 2019-09-25 | 0.0012 | 0.0037 | 0.0013 | 0.0059 |
| 2019-09-26 | -0.0023 | -0.0024 | 0.0030 | 0.0114 |
| 2019-09-27 | 0.0013 | 0.0107 | -0.0032 | -0.0111 |
| 2019-09-28 | 0.0000 | 0.0014 | 0.0016 | 0.0032 |
| 2019-09-29 | -0.0016 | 0.0032 | 0.0018 | -0.0081 |
| 2019-09-30 | -0.0017 | 0.0058 | 0.0004 | 0.0048 |
| | | | | |
| **Mean** | -0.0003 | 0.0026 | 0.0014 | 0.0026 |

| | | | | |
|---|---|---|---|---|
| **Standard deviation** | 0.0016 | 0.0062 | 0.0018 | 0.0064 |

**Table S4. Parameters for nonlinear regression of the humidity response accuracy corrections (Equation S1).**

| Isotopic species | Parameter | Estimate | Standard error | t value | Pr(>|t|) |
|---|---|---|---|---|---|
| $\delta^{18}O$ | *a* | 1.92 | 0.061 | 31.6 | <0.001 |
| | *b* | -3190 | 80 | -42.1 | <0.001 |
| $\delta^2H$ | *a* | 1.13 | 0.41 | 2.8 | 0.005 |
| | *b* | -10500 | 510 | -21.0 | <0.001 |

**Table S5. Parameters for nonlinear regression of the humidity response precision (Equation S2).**

| Isotopic species | Parameter | Estimate | Standard error | t value | Pr(>|t|) |
|---|---|---|---|---|---|
| $\delta^{18}O$ | *a* | -0.003 | 0.002 | -1.4 | 0.177 |
| | *b* | 260 | 3 | 102.2 | <0.001 |
| $\delta^2H$ | *a* | -0.019 | 0.016 | -1.2 | 0.238 |
| | *b* | 1690 | 20 | 85.7 | <0.001 |

**Table S6. Percentage of water vapor uptake attributed to Baffin Bay and the Labrador Sea for moisture arriving at Thule, based on back trajectory analysis and split by meteorological season.**

| Domain | DJF | MAM | JJA | SON |
|---|---|---|---|---|
| Entire Baffin Bay (Davis to Nares) | 35.6 | 44.4 | 51.2 | 47.2 |
| Labrador Sea | 13.7 | 16.3 | 12.2 | 11.7 |
| Entire Baffin Bay + Labrador Sea | 49.3 | 60.7 | 63.4 | 58.9 |

**References**

[revised manuscript text omitted]

Formatted Table

| Page 9: [3] Formatted | Microsoft Office User | 8/17/20 2:49:00 PM |
|---|---|---|

Superscript

| Page 9: [4] Formatted | Microsoft Office User | 8/17/20 2:49:00 PM |
|---|---|---|

Superscript

| Page 9: [5] Formatted | Microsoft Office User | 8/7/20 3:23:00 PM |
|---|---|---|

Centered

| Page 9: [6] Formatted | Microsoft Office User | 8/6/20 3:50:00 PM |
|---|---|---|

Font: (Default) Times New Roman

| Page 9: [7] Formatted | Microsoft Office User | 8/6/20 3:50:00 PM |
|---|---|---|

Font: (Default) Times New Roman

| Page 9: [8] Formatted | Microsoft Office User | 8/6/20 3:50:00 PM |
|---|---|---|

Font: (Default) Times New Roman

| Page 9: [9] Formatted | Microsoft Office User | 8/6/20 3:50:00 PM |
|---|---|---|

Font: (Default) Times New Roman

| Page 9: [10] Formatted | Microsoft Office User | 8/6/20 3:50:00 PM |
|---|---|---|

Font: (Default) Times New Roman

| Page 9: [11] Formatted | Microsoft Office User | 8/6/20 3:50:00 PM |
|---|---|---|

Font: (Default) Times New Roman

| Page 9: [12] Formatted | Microsoft Office User | 8/6/20 3:50:00 PM |
|---|---|---|

Font: (Default) Times New Roman

| Page 9: [13] Formatted | Microsoft Office User | 8/6/20 3:50:00 PM |
|---|---|---|

Font: (Default) Times New Roman

| Page 9: [14] Formatted | Microsoft Office User | 8/6/20 3:50:00 PM |
|---|---|---|

Font: (Default) Times New Roman

| Page 9: [16] Formatted | Microsoft Office User | 8/6/20 3:50:00 PM |
|---|---|---|

Font: (Default) Times New Roman

| Page 9: [17] Formatted | Microsoft Office User | 8/6/20 3:50:00 PM |
|---|---|---|

Font: (Default) Times New Roman

| Page 9: [18] Formatted | Microsoft Office User | 8/6/20 3:50:00 PM |
|---|---|---|

Font: (Default) Times New Roman

| Page 9: [19] Formatted | Microsoft Office User | 8/6/20 3:50:00 PM |
|---|---|---|

Font: (Default) Times New Roman

| Page 9: [20] Formatted | Microsoft Office User | 8/6/20 3:50:00 PM |
|---|---|---|

Font: (Default) Times New Roman

| Page 9: [21] Formatted | Microsoft Office User | 8/6/20 3:50:00 PM |
|---|---|---|

Font: (Default) Times New Roman

| Page 9: [22] Formatted | Microsoft Office User | 8/6/20 3:50:00 PM |
|---|---|---|

Font: (Default) Times New Roman

| Page 9: [23] Formatted | Microsoft Office User | 8/6/20 3:50:00 PM |
|---|---|---|

Font: (Default) Times New Roman

| Page 9: [24] Formatted | Microsoft Office User | 8/6/20 3:50:00 PM |
|---|---|---|

Font: (Default) Times New Roman

| Page 9: [25] Formatted | Microsoft Office User | 8/6/20 3:50:00 PM |
|---|---|---|

Font: (Default) Times New Roman

| Page 9: [26] Formatted | Microsoft Office User | 8/6/20 3:50:00 PM |
|---|---|---|

Font: (Default) Times New Roman

| Page 9: [27] Formatted | Microsoft Office User | 8/6/20 3:50:00 PM |
|---|---|---|

Font: (Default) Times New Roman

| Page 9: [28] Formatted | Microsoft Office User | 8/6/20 3:50:00 PM |
|---|---|---|

Font: (Default) Times New Roman

| Page 9: [30] Formatted | Microsoft Office User | 8/6/20 3:50:00 PM |

Font: (Default) Times New Roman

| Page 9: [31] Formatted | Microsoft Office User | 8/6/20 3:50:00 PM |

Font: (Default) Times New Roman

| Page 9: [32] Formatted | Microsoft Office User | 8/6/20 3:50:00 PM |

Font: (Default) Times New Roman

| Page 9: [33] Formatted | Microsoft Office User | 8/6/20 3:50:00 PM |

Font: (Default) Times New Roman

| Page 9: [34] Formatted | Microsoft Office User | 8/6/20 3:50:00 PM |

Font: (Default) Times New Roman

| Page 9: [35] Formatted | Microsoft Office User | 8/6/20 3:50:00 PM |

Font: (Default) Times New Roman

| Page 9: [36] Formatted | Microsoft Office User | 8/6/20 3:50:00 PM |

Font: (Default) Times New Roman

| Page 9: [37] Formatted | Microsoft Office User | 8/6/20 3:50:00 PM |

Font: (Default) Times New Roman

| Page 9: [38] Formatted | Microsoft Office User | 8/6/20 3:50:00 PM |

Font: (Default) Times New Roman

| Page 9: [39] Formatted | Microsoft Office User | 8/6/20 3:50:00 PM |

Font: (Default) Times New Roman

| Page 9: [40] Formatted | Microsoft Office User | 8/6/20 3:50:00 PM |

Font: (Default) Times New Roman

| Page 9: [41] Formatted | Microsoft Office User | 8/6/20 3:50:00 PM |

Font: (Default) Times New Roman

| Page 9: [42] Formatted | Microsoft Office User | 8/6/20 3:50:00 PM |

Font: (Default) Times New Roman

| Page 9: [43] Formatted | Microsoft Office User | 8/6/20 3:50:00 PM |

| Page 9: [44] Formatted | Microsoft Office User | 8/6/20 3:50:00 PM |
|---|---|---|

Font: (Default) Times New Roman

| Page 9: [45] Formatted | Microsoft Office User | 8/6/20 3:50:00 PM |
|---|---|---|

Font: (Default) Times New Roman

| Page 9: [46] Formatted | Microsoft Office User | 8/6/20 3:50:00 PM |
|---|---|---|

Font: (Default) Times New Roman

| Page 9: [47] Formatted | Microsoft Office User | 8/6/20 3:50:00 PM |
|---|---|---|

Font: (Default) Times New Roman

| Page 9: [48] Formatted | Microsoft Office User | 8/6/20 3:50:00 PM |
|---|---|---|

Font: (Default) Times New Roman

| Page 9: [49] Formatted | Microsoft Office User | 8/6/20 3:50:00 PM |
|---|---|---|

Font: (Default) Times New Roman

| Page 9: [50] Formatted | Microsoft Office User | 8/6/20 3:50:00 PM |
|---|---|---|

Font: (Default) Times New Roman

| Page 9: [51] Formatted | Microsoft Office User | 8/6/20 3:50:00 PM |
|---|---|---|

Font: (Default) Times New Roman

| Page 9: [52] Formatted | Microsoft Office User | 8/6/20 3:50:00 PM |
|---|---|---|

Font: (Default) Times New Roman

| Page 9: [53] Formatted | Microsoft Office User | 8/6/20 3:50:00 PM |
|---|---|---|

Font: (Default) Times New Roman

| Page 9: [54] Formatted | Microsoft Office User | 8/6/20 3:50:00 PM |
|---|---|---|

Font: (Default) Times New Roman

| Page 9: [55] Formatted | Microsoft Office User | 8/6/20 3:50:00 PM |
|---|---|---|

Font: (Default) Times New Roman

| Page 9: [56] Formatted | Microsoft Office User | 8/6/20 3:50:00 PM |
|---|---|---|

Font: (Default) Times New Roman

| Page 9: [57] Formatted | Microsoft Office User | 8/6/20 3:50:00 PM |
|---|---|---|

| Page 9: [58] Formatted | Microsoft Office User | 8/6/20 3:50:00 PM |
|---|---|---|

Font: (Default) Times New Roman

| Page 9: [59] Formatted | Microsoft Office User | 8/6/20 3:50:00 PM |
|---|---|---|

Font: (Default) Times New Roman

| Page 9: [60] Formatted | Microsoft Office User | 8/6/20 3:50:00 PM |
|---|---|---|

Font: (Default) Times New Roman

| Page 9: [61] Formatted | Microsoft Office User | 8/6/20 3:50:00 PM |
|---|---|---|

Font: (Default) Times New Roman

| Page 9: [62] Formatted | Microsoft Office User | 8/6/20 3:50:00 PM |
|---|---|---|

Font: (Default) Times New Roman

| Page 9: [63] Formatted | Microsoft Office User | 8/6/20 3:50:00 PM |
|---|---|---|

Font: (Default) Times New Roman

| Page 9: [64] Formatted | Microsoft Office User | 8/6/20 3:50:00 PM |
|---|---|---|

Font: (Default) Times New Roman

| Page 9: [65] Formatted | Microsoft Office User | 8/6/20 3:50:00 PM |
|---|---|---|

Font: (Default) Times New Roman

| Page 9: [66] Formatted | Microsoft Office User | 8/6/20 3:50:00 PM |
|---|---|---|

Font: (Default) Times New Roman

| Page 9: [67] Formatted | Microsoft Office User | 8/6/20 3:50:00 PM |
|---|---|---|

Font: (Default) Times New Roman

| Page 9: [68] Formatted | Microsoft Office User | 8/6/20 3:50:00 PM |
|---|---|---|

Font: (Default) Times New Roman

| Page 9: [69] Formatted | Microsoft Office User | 8/6/20 3:50:00 PM |
|---|---|---|

Font: (Default) Times New Roman

| Page 9: [70] Formatted | Microsoft Office User | 8/6/20 3:50:00 PM |
|---|---|---|

Font: (Default) Times New Roman

| Page 9: [71] Formatted | Microsoft Office User | 8/6/20 3:50:00 PM |
|---|---|---|

| Page 9: [72] Formatted | Microsoft Office User | 8/6/20 3:50:00 PM |
|---|---|---|

Font: (Default) Times New Roman

| Page 9: [73] Formatted | Microsoft Office User | 8/6/20 3:50:00 PM |
|---|---|---|

Font: (Default) Times New Roman

| Page 9: [74] Formatted | Microsoft Office User | 8/6/20 3:50:00 PM |
|---|---|---|

Font: (Default) Times New Roman

| Page 9: [75] Formatted | Microsoft Office User | 8/6/20 3:50:00 PM |
|---|---|---|

Font: (Default) Times New Roman

| Page 9: [76] Formatted | Microsoft Office User | 8/6/20 3:50:00 PM |
|---|---|---|

Font: (Default) Times New Roman

| Page 9: [77] Formatted | Microsoft Office User | 8/6/20 3:50:00 PM |
|---|---|---|

Font: (Default) Times New Roman

| Page 9: [78] Formatted | Microsoft Office User | 8/6/20 3:50:00 PM |
|---|---|---|

Font: (Default) Times New Roman

| Page 9: [79] Formatted | Microsoft Office User | 8/6/20 3:50:00 PM |
|---|---|---|

Font: (Default) Times New Roman

| Page 9: [80] Formatted | Microsoft Office User | 8/6/20 3:50:00 PM |
|---|---|---|

Font: (Default) Times New Roman

| Page 9: [81] Formatted | Microsoft Office User | 8/6/20 3:50:00 PM |
|---|---|---|

Font: (Default) Times New Roman

| Page 9: [82] Formatted | Microsoft Office User | 8/6/20 3:50:00 PM |
|---|---|---|

Font: (Default) Times New Roman

| Page 9: [83] Formatted | Microsoft Office User | 8/6/20 3:50:00 PM |
|---|---|---|

Font: (Default) Times New Roman

| Page 9: [84] Formatted | Microsoft Office User | 8/6/20 3:50:00 PM |
|---|---|---|

Font: (Default) Times New Roman

| Page 9: [85] Formatted | Microsoft Office User | 8/6/20 3:50:00 PM |
|---|---|---|

| Page 9: [86] Formatted | Microsoft Office User | 8/6/20 3:50:00 PM |
|---|---|---|

Font: (Default) Times New Roman

| Page 9: [87] Formatted | Microsoft Office User | 8/6/20 3:50:00 PM |
|---|---|---|

Font: (Default) Times New Roman

| Page 9: [88] Formatted | Microsoft Office User | 8/6/20 3:50:00 PM |
|---|---|---|

Font: (Default) Times New Roman

| Page 9: [89] Formatted | Microsoft Office User | 8/6/20 3:50:00 PM |
|---|---|---|

Font: (Default) Times New Roman

| Page 9: [90] Formatted | Microsoft Office User | 8/6/20 3:50:00 PM |
|---|---|---|

Font: (Default) Times New Roman

| Page 9: [91] Formatted | Microsoft Office User | 8/6/20 3:50:00 PM |
|---|---|---|

Font: (Default) Times New Roman

| Page 9: [92] Formatted | Microsoft Office User | 8/6/20 3:50:00 PM |
|---|---|---|

Font: (Default) Times New Roman

| Page 9: [93] Formatted | Microsoft Office User | 8/6/20 3:50:00 PM |
|---|---|---|

Font: (Default) Times New Roman

| Page 9: [94] Formatted | Microsoft Office User | 8/6/20 3:50:00 PM |
|---|---|---|

Font: (Default) Times New Roman

| Page 9: [95] Formatted | Microsoft Office User | 8/6/20 3:50:00 PM |
|---|---|---|

Font: (Default) Times New Roman

| Page 9: [96] Formatted | Microsoft Office User | 8/6/20 3:50:00 PM |
|---|---|---|

Font: (Default) Times New Roman

| Page 9: [97] Formatted | Microsoft Office User | 8/6/20 3:50:00 PM |
|---|---|---|

Font: (Default) Times New Roman

| Page 9: [98] Formatted | Microsoft Office User | 8/6/20 3:50:00 PM |
|---|---|---|

Font: (Default) Times New Roman

| Page 9: [99] Formatted | Microsoft Office User | 8/6/20 3:50:00 PM |
|---|---|---|

| Page 9: [100] Formatted | Microsoft Office User | 8/6/20 3:50:00 PM |
|---|---|---|

Font: (Default) Times New Roman

| Page 9: [101] Formatted | Microsoft Office User | 8/6/20 3:50:00 PM |
|---|---|---|

Font: (Default) Times New Roman

| Page 9: [102] Formatted | Microsoft Office User | 8/6/20 3:50:00 PM |
|---|---|---|

Font: (Default) Times New Roman

| Page 9: [103] Formatted | Microsoft Office User | 8/6/20 3:50:00 PM |
|---|---|---|

Font: (Default) Times New Roman

| Page 9: [104] Formatted | Microsoft Office User | 8/6/20 3:50:00 PM |
|---|---|---|

Font: (Default) Times New Roman

| Page 9: [105] Formatted | Microsoft Office User | 8/6/20 3:50:00 PM |
|---|---|---|

Font: (Default) Times New Roman

| Page 9: [106] Formatted | Microsoft Office User | 8/6/20 3:50:00 PM |
|---|---|---|

Font: (Default) Times New Roman

| Page 9: [107] Formatted | Microsoft Office User | 8/6/20 3:50:00 PM |
|---|---|---|

Font: (Default) Times New Roman

| Page 9: [108] Formatted | Microsoft Office User | 8/6/20 3:50:00 PM |
|---|---|---|

Font: (Default) Times New Roman

| Page 9: [109] Formatted | Microsoft Office User | 8/6/20 3:50:00 PM |
|---|---|---|

Font: (Default) Times New Roman

| Page 9: [110] Formatted | Microsoft Office User | 8/6/20 3:50:00 PM |
|---|---|---|

Font: (Default) Times New Roman

| Page 9: [111] Formatted | Microsoft Office User | 8/6/20 3:50:00 PM |
|---|---|---|

Font: (Default) Times New Roman

| Page 9: [112] Formatted | Microsoft Office User | 8/6/20 3:50:00 PM |
|---|---|---|

Font: (Default) Times New Roman

| Page 9: [113] Formatted | Microsoft Office User | 8/6/20 3:50:00 PM |
|---|---|---|

| Page 9: [114] Formatted | Microsoft Office User | 8/6/20 3:50:00 PM |
|---|---|---|

Font: (Default) Times New Roman

| Page 9: [115] Formatted | Microsoft Office User | 8/6/20 3:50:00 PM |
|---|---|---|

Font: (Default) Times New Roman

| Page 9: [116] Formatted | Microsoft Office User | 8/6/20 3:50:00 PM |
|---|---|---|

Font: (Default) Times New Roman

| Page 9: [117] Formatted | Microsoft Office User | 8/6/20 3:50:00 PM |
|---|---|---|

Font: (Default) Times New Roman

| Page 9: [118] Formatted | Microsoft Office User | 8/6/20 3:50:00 PM |
|---|---|---|

Font: (Default) Times New Roman

| Page 9: [119] Formatted | Microsoft Office User | 8/6/20 3:50:00 PM |
|---|---|---|

Font: (Default) Times New Roman

| Page 9: [120] Formatted | Microsoft Office User | 8/6/20 3:50:00 PM |
|---|---|---|

Font: (Default) Times New Roman

| Page 9: [121] Formatted | Microsoft Office User | 8/6/20 3:50:00 PM |
|---|---|---|

Font: (Default) Times New Roman

| Page 9: [122] Formatted | Microsoft Office User | 8/6/20 3:50:00 PM |
|---|---|---|

Font: (Default) Times New Roman

| Page 9: [123] Formatted | Microsoft Office User | 8/6/20 3:50:00 PM |
|---|---|---|

Font: (Default) Times New Roman

| Page 9: [124] Formatted | Microsoft Office User | 8/6/20 3:50:00 PM |
|---|---|---|

Font: (Default) Times New Roman

| Page 9: [125] Formatted | Microsoft Office User | 8/6/20 3:50:00 PM |
|---|---|---|

Font: (Default) Times New Roman

| Page 9: [126] Formatted | Microsoft Office User | 8/6/20 3:50:00 PM |
|---|---|---|

Font: (Default) Times New Roman

| Page 9: [127] Formatted | Microsoft Office User | 8/6/20 3:50:00 PM |
|---|---|---|

**Page 9: [128] Formatted**      **Microsoft Office User**      **8/6/20 3:50:00 PM**

Font: (Default) Times New Roman

**Page 9: [129] Formatted**      **Microsoft Office User**      **8/6/20 3:50:00 PM**

Font: (Default) Times New Roman

**Page 9: [130] Formatted**      **Microsoft Office User**      **8/6/20 3:50:00 PM**

Font: (Default) Times New Roman

**Page 9: [131] Formatted**      **Microsoft Office User**      **8/6/20 3:50:00 PM**

Font: (Default) Times New Roman

**Page 9: [132] Formatted**      **Microsoft Office User**      **8/6/20 3:50:00 PM**

Font: (Default) Times New Roman

**Page 9: [133] Formatted**      **Microsoft Office User**      **8/6/20 3:50:00 PM**

Font: (Default) Times New Roman

**Page 9: [134] Formatted**      **Microsoft Office User**      **8/6/20 3:50:00 PM**

Font: (Default) Times New Roman

**Page 9: [135] Formatted**      **Microsoft Office User**      **8/6/20 3:50:00 PM**

Font: (Default) Times New Roman

**Page 10: [136] Formatted**      **Microsoft Office User**      **8/6/20 3:50:00 PM**

Font: (Default) Times New Roman

**Page 10: [137] Formatted**      **Microsoft Office User**      **8/6/20 3:50:00 PM**

Font: (Default) Times New Roman

**Page 10: [138] Formatted**      **Microsoft Office User**      **8/6/20 3:50:00 PM**

Font: (Default) Times New Roman

**Page 10: [139] Formatted**      **Microsoft Office User**      **8/6/20 3:50:00 PM**

Font: (Default) Times New Roman

**Page 10: [140] Formatted**      **Microsoft Office User**      **8/6/20 3:50:00 PM**

Font: (Default) Times New Roman

**Page 10: [141] Formatted**      **Microsoft Office User**      **8/6/20 3:50:00 PM**

**Page 10: [142] Formatted**      **Microsoft Office User**      **8/6/20 3:50:00 PM**

Font: (Default) Times New Roman

**Page 10: [143] Formatted**      **Microsoft Office User**      **8/6/20 3:50:00 PM**

Font: (Default) Times New Roman

**Page 10: [144] Formatted**      **Microsoft Office User**      **8/6/20 3:50:00 PM**

Font: (Default) Times New Roman

**Page 10: [145] Formatted**      **Microsoft Office User**      **8/6/20 3:50:00 PM**

Font: (Default) Times New Roman

**Page 10: [146] Formatted**      **Microsoft Office User**      **8/6/20 3:50:00 PM**

Font: (Default) Times New Roman

**Page 10: [147] Formatted**      **Microsoft Office User**      **8/6/20 3:50:00 PM**

Font: (Default) Times New Roman

**Page 10: [148] Formatted**      **Microsoft Office User**      **8/6/20 3:50:00 PM**

Font: (Default) Times New Roman

**Page 10: [149] Formatted**      **Microsoft Office User**      **8/6/20 3:50:00 PM**

Font: (Default) Times New Roman

**Page 10: [150] Formatted**      **Microsoft Office User**      **8/6/20 3:50:00 PM**

Font: (Default) Times New Roman

**Page 10: [151] Formatted**      **Microsoft Office User**      **8/6/20 3:50:00 PM**

Font: (Default) Times New Roman

**Page 10: [152] Formatted**      **Microsoft Office User**      **8/6/20 3:50:00 PM**

Font: (Default) Times New Roman

**Page 10: [153] Formatted**      **Microsoft Office User**      **8/6/20 3:50:00 PM**

Font: (Default) Times New Roman

**Page 10: [154] Formatted**      **Microsoft Office User**      **8/6/20 3:50:00 PM**

Font: (Default) Times New Roman

**Page 10: [155] Formatted**      **Microsoft Office User**      **8/6/20 3:50:00 PM**

**Page 10: [156] Formatted** | **Microsoft Office User** | **8/6/20 3:50:00 PM**

Font: (Default) Times New Roman

**Page 10: [157] Formatted** | **Microsoft Office User** | **8/6/20 3:50:00 PM**

Font: (Default) Times New Roman

**Page 10: [158] Formatted** | **Microsoft Office User** | **8/6/20 3:50:00 PM**

Font: (Default) Times New Roman

**Page 10: [159] Formatted** | **Microsoft Office User** | **8/6/20 3:50:00 PM**

Font: (Default) Times New Roman

**Page 10: [160] Formatted** | **Microsoft Office User** | **8/6/20 3:50:00 PM**

Font: (Default) Times New Roman

**Page 10: [161] Formatted** | **Microsoft Office User** | **8/6/20 3:50:00 PM**

Font: (Default) Times New Roman

**Page 10: [162] Formatted** | **Microsoft Office User** | **8/6/20 3:50:00 PM**

Font: (Default) Times New Roman

**Page 10: [163] Formatted** | **Microsoft Office User** | **8/6/20 3:50:00 PM**

Font: (Default) Times New Roman

**Page 10: [164] Formatted** | **Microsoft Office User** | **8/6/20 3:50:00 PM**

Font: (Default) Times New Roman

**Page 10: [165] Formatted** | **Microsoft Office User** | **8/6/20 3:50:00 PM**

Font: (Default) Times New Roman

**Page 10: [166] Formatted** | **Microsoft Office User** | **8/6/20 3:50:00 PM**

Font: (Default) Times New Roman

**Page 10: [167] Formatted** | **Microsoft Office User** | **8/6/20 3:50:00 PM**

Font: (Default) Times New Roman

**Page 10: [168] Formatted** | **Microsoft Office User** | **8/6/20 3:50:00 PM**

Font: (Default) Times New Roman

**Page 10: [169] Formatted** | **Microsoft Office User** | **8/6/20 3:50:00 PM**

**Page 10: [170] Formatted** | **Microsoft Office User** | **8/6/20 3:50:00 PM**

Font: (Default) Times New Roman

**Page 10: [171] Formatted** | **Microsoft Office User** | **8/6/20 3:50:00 PM**

Font: (Default) Times New Roman

**Page 10: [172] Formatted** | **Microsoft Office User** | **8/6/20 3:50:00 PM**

Font: (Default) Times New Roman

**Page 10: [173] Formatted** | **Microsoft Office User** | **8/6/20 3:50:00 PM**

Font: (Default) Times New Roman

**Page 10: [174] Formatted** | **Microsoft Office User** | **8/6/20 3:50:00 PM**

Font: (Default) Times New Roman

**Page 10: [175] Formatted** | **Microsoft Office User** | **8/6/20 3:50:00 PM**

Font: (Default) Times New Roman

**Page 10: [176] Formatted** | **Microsoft Office User** | **8/6/20 3:50:00 PM**

Font: (Default) Times New Roman

**Page 10: [177] Formatted** | **Microsoft Office User** | **8/6/20 3:50:00 PM**

Font: (Default) Times New Roman

**Page 10: [178] Formatted** | **Microsoft Office User** | **8/6/20 3:50:00 PM**

Font: (Default) Times New Roman

**Page 10: [179] Formatted** | **Microsoft Office User** | **8/6/20 3:50:00 PM**

Font: (Default) Times New Roman

**Page 10: [180] Formatted** | **Microsoft Office User** | **8/6/20 3:50:00 PM**

Font: (Default) Times New Roman

**Page 10: [181] Formatted** | **Microsoft Office User** | **8/6/20 3:50:00 PM**

Font: (Default) Times New Roman

**Page 10: [182] Formatted** | **Microsoft Office User** | **8/6/20 3:50:00 PM**

Font: (Default) Times New Roman

**Page 10: [183] Formatted** | **Microsoft Office User** | **8/6/20 3:50:00 PM**

**Page 10: [184] Formatted**      **Microsoft Office User**      **8/6/20 3:50:00 PM**

Font: (Default) Times New Roman

**Page 10: [185] Formatted**      **Microsoft Office User**      **8/6/20 3:50:00 PM**

Font: (Default) Times New Roman

**Page 10: [186] Formatted**      **Microsoft Office User**      **8/6/20 3:50:00 PM**

Font: (Default) Times New Roman

**Page 10: [187] Formatted**      **Microsoft Office User**      **8/6/20 3:50:00 PM**

Font: (Default) Times New Roman

**Page 10: [188] Formatted**      **Microsoft Office User**      **8/6/20 3:50:00 PM**

Font: (Default) Times New Roman

**Page 10: [189] Formatted**      **Microsoft Office User**      **8/6/20 3:50:00 PM**

Font: (Default) Times New Roman

**Page 10: [190] Formatted**      **Microsoft Office User**      **8/6/20 3:50:00 PM**

Font: (Default) Times New Roman

**Page 10: [191] Formatted**      **Microsoft Office User**      **8/6/20 3:50:00 PM**

Font: (Default) Times New Roman

**Page 10: [192] Formatted**      **Microsoft Office User**      **8/6/20 3:50:00 PM**

Font: (Default) Times New Roman

**Page 10: [193] Formatted**      **Microsoft Office User**      **8/6/20 3:50:00 PM**

Font: (Default) Times New Roman

**Page 10: [194] Formatted**      **Microsoft Office User**      **8/6/20 3:50:00 PM**

Font: (Default) Times New Roman

**Page 10: [195] Formatted**      **Microsoft Office User**      **8/6/20 3:50:00 PM**

Font: (Default) Times New Roman

**Page 10: [196] Formatted**      **Microsoft Office User**      **8/6/20 3:50:00 PM**

Font: (Default) Times New Roman

**Page 10: [197] Formatted**      **Microsoft Office User**      **8/6/20 3:50:00 PM**

**Page 10: [198] Formatted**      **Microsoft Office User**      **8/6/20 3:50:00 PM**

Font: (Default) Times New Roman

**Page 10: [199] Formatted**      **Microsoft Office User**      **8/6/20 3:50:00 PM**

Font: (Default) Times New Roman

**Page 10: [200] Formatted**      **Microsoft Office User**      **8/6/20 3:50:00 PM**

Font: (Default) Times New Roman

**Page 10: [201] Formatted**      **Microsoft Office User**      **8/6/20 3:50:00 PM**

Font: (Default) Times New Roman

**Page 10: [202] Formatted**      **Microsoft Office User**      **8/6/20 3:50:00 PM**

Font: (Default) Times New Roman

**Page 10: [203] Formatted**      **Microsoft Office User**      **8/6/20 3:50:00 PM**

Font: (Default) Times New Roman

**Page 10: [204] Formatted**      **Microsoft Office User**      **8/6/20 3:50:00 PM**

Font: (Default) Times New Roman

**Page 10: [205] Formatted**      **Microsoft Office User**      **8/6/20 3:50:00 PM**

Font: (Default) Times New Roman

**Page 10: [206] Formatted**      **Microsoft Office User**      **8/6/20 3:50:00 PM**

Font: (Default) Times New Roman

**Page 10: [207] Formatted**      **Microsoft Office User**      **8/6/20 3:50:00 PM**

Font: (Default) Times New Roman

**Page 10: [208] Formatted**      **Microsoft Office User**      **8/6/20 3:50:00 PM**

Font: (Default) Times New Roman

**Page 10: [209] Formatted**      **Microsoft Office User**      **8/6/20 3:50:00 PM**

Font: (Default) Times New Roman

**Page 10: [210] Formatted**      **Microsoft Office User**      **8/6/20 3:50:00 PM**

Font: (Default) Times New Roman

**Page 10: [211] Formatted**      **Microsoft Office User**      **8/6/20 3:50:00 PM**

**Page 10: [212] Formatted**      **Microsoft Office User**      **8/6/20 3:50:00 PM**

Font: (Default) Times New Roman

**Page 10: [213] Formatted**      **Microsoft Office User**      **8/6/20 3:50:00 PM**

Font: (Default) Times New Roman

**Page 10: [214] Formatted**      **Microsoft Office User**      **8/6/20 3:50:00 PM**

Font: (Default) Times New Roman

**Page 10: [215] Formatted**      **Microsoft Office User**      **8/6/20 3:50:00 PM**

Font: (Default) Times New Roman

**Page 10: [216] Formatted**      **Microsoft Office User**      **8/6/20 3:50:00 PM**

Font: (Default) Times New Roman

**Page 10: [217] Formatted**      **Microsoft Office User**      **8/6/20 3:50:00 PM**

Font: (Default) Times New Roman

**Page 10: [218] Formatted**      **Microsoft Office User**      **8/6/20 3:50:00 PM**

Font: (Default) Times New Roman

**Page 10: [219] Formatted**      **Microsoft Office User**      **8/6/20 3:50:00 PM**

Font: (Default) Times New Roman

**Page 10: [220] Formatted**      **Microsoft Office User**      **8/6/20 3:50:00 PM**

Font: (Default) Times New Roman

**Page 10: [221] Formatted**      **Microsoft Office User**      **8/6/20 3:50:00 PM**

Font: (Default) Times New Roman

**Page 10: [222] Formatted**      **Microsoft Office User**      **8/6/20 3:50:00 PM**

Font: (Default) Times New Roman

**Page 10: [223] Formatted**      **Microsoft Office User**      **8/6/20 3:50:00 PM**

Font: (Default) Times New Roman

**Page 10: [224] Formatted**      **Microsoft Office User**      **8/6/20 3:50:00 PM**

Font: (Default) Times New Roman

**Page 10: [225] Formatted**      **Microsoft Office User**      **8/6/20 3:50:00 PM**

**Page 10: [226] Formatted**     **Microsoft Office User**     **8/6/20 3:50:00 PM**

Font: (Default) Times New Roman

**Page 10: [227] Formatted**     **Microsoft Office User**     **8/6/20 3:50:00 PM**

Font: (Default) Times New Roman

**Page 10: [228] Formatted**     **Microsoft Office User**     **8/6/20 3:50:00 PM**

Font: (Default) Times New Roman

**Page 10: [229] Formatted**     **Microsoft Office User**     **8/6/20 3:50:00 PM**

Font: (Default) Times New Roman

**Page 10: [230] Formatted**     **Microsoft Office User**     **8/6/20 3:50:00 PM**

Font: (Default) Times New Roman

**Page 10: [231] Formatted**     **Microsoft Office User**     **8/6/20 3:50:00 PM**

Font: (Default) Times New Roman

**Page 10: [232] Formatted**     **Microsoft Office User**     **8/6/20 3:50:00 PM**

Font: (Default) Times New Roman

**Page 10: [233] Formatted**     **Microsoft Office User**     **8/6/20 3:50:00 PM**

Font: (Default) Times New Roman

**Page 10: [234] Formatted**     **Microsoft Office User**     **8/6/20 3:50:00 PM**

Font: (Default) Times New Roman

**Page 10: [235] Formatted**     **Microsoft Office User**     **8/6/20 3:50:00 PM**

Font: (Default) Times New Roman

**Page 10: [236] Formatted**     **Microsoft Office User**     **8/6/20 3:50:00 PM**

Font: (Default) Times New Roman

**Page 10: [237] Formatted**     **Microsoft Office User**     **8/6/20 3:50:00 PM**

Font: (Default) Times New Roman

**Page 10: [238] Formatted**     **Microsoft Office User**     **8/6/20 3:50:00 PM**

Font: (Default) Times New Roman

**Page 10: [239] Formatted**     **Microsoft Office User**     **8/6/20 3:50:00 PM**

| Page 10: [240] Formatted | Microsoft Office User | 8/6/20 3:50:00 PM |
|---|---|---|

Font: (Default) Times New Roman

| Page 10: [241] Formatted | Microsoft Office User | 8/6/20 3:50:00 PM |
|---|---|---|

Font: (Default) Times New Roman

| Page 10: [242] Formatted | Microsoft Office User | 8/6/20 3:50:00 PM |
|---|---|---|

Font: (Default) Times New Roman

| Page 10: [243] Formatted | Microsoft Office User | 8/6/20 3:50:00 PM |
|---|---|---|

Font: (Default) Times New Roman

| Page 10: [244] Formatted | Microsoft Office User | 8/6/20 3:50:00 PM |
|---|---|---|

Font: (Default) Times New Roman

| Page 10: [245] Formatted | Microsoft Office User | 8/6/20 3:50:00 PM |
|---|---|---|

Font: (Default) Times New Roman

| Page 10: [246] Formatted | Microsoft Office User | 8/6/20 3:50:00 PM |
|---|---|---|

Font: (Default) Times New Roman

| Page 10: [247] Formatted | Microsoft Office User | 8/6/20 3:50:00 PM |
|---|---|---|

Font: (Default) Times New Roman

| Page 10: [248] Formatted | Microsoft Office User | 8/6/20 3:50:00 PM |
|---|---|---|

Font: (Default) Times New Roman

| Page 10: [249] Formatted | Microsoft Office User | 8/6/20 3:50:00 PM |
|---|---|---|

Font: (Default) Times New Roman

| Page 10: [250] Formatted | Microsoft Office User | 8/6/20 3:50:00 PM |
|---|---|---|

Font: (Default) Times New Roman

| Page 10: [251] Formatted | Microsoft Office User | 8/6/20 3:50:00 PM |
|---|---|---|

Font: (Default) Times New Roman

| Page 10: [252] Formatted | Microsoft Office User | 8/6/20 3:50:00 PM |
|---|---|---|

Font: (Default) Times New Roman

| Page 10: [253] Formatted | Microsoft Office User | 8/6/20 3:50:00 PM |
|---|---|---|

**Page 10: [254] Formatted** | **Microsoft Office User** | **8/6/20 3:50:00 PM**

Font: (Default) Times New Roman

**Page 10: [255] Formatted** | **Microsoft Office User** | **8/6/20 3:50:00 PM**

Font: (Default) Times New Roman

**Page 10: [256] Formatted** | **Microsoft Office User** | **8/6/20 3:50:00 PM**

Font: (Default) Times New Roman

**Page 10: [257] Formatted** | **Microsoft Office User** | **8/6/20 3:50:00 PM**

Font: (Default) Times New Roman

**Page 10: [258] Formatted** | **Microsoft Office User** | **8/6/20 3:50:00 PM**

Font: (Default) Times New Roman

**Page 10: [259] Formatted** | **Microsoft Office User** | **8/6/20 3:50:00 PM**

Font: (Default) Times New Roman

**Page 10: [260] Formatted** | **Microsoft Office User** | **8/6/20 3:50:00 PM**

Font: (Default) Times New Roman

**Page 10: [261] Formatted** | **Microsoft Office User** | **8/6/20 3:50:00 PM**

Font: (Default) Times New Roman

**Page 10: [262] Formatted** | **Microsoft Office User** | **8/6/20 3:50:00 PM**

Font: (Default) Times New Roman

**Page 10: [263] Formatted** | **Microsoft Office User** | **8/6/20 3:50:00 PM**

Font: (Default) Times New Roman

**Page 10: [264] Formatted** | **Microsoft Office User** | **8/6/20 3:50:00 PM**

Font: (Default) Times New Roman

**Page 10: [265] Formatted** | **Microsoft Office User** | **8/6/20 3:50:00 PM**

Font: (Default) Times New Roman

**Page 10: [266] Formatted** | **Microsoft Office User** | **8/6/20 3:50:00 PM**

Font: (Default) Times New Roman

**Page 10: [267] Formatted** | **Microsoft Office User** | **8/6/20 3:50:00 PM**

**Page 10: [268] Formatted**      **Microsoft Office User**      **8/6/20 3:50:00 PM**

Font: (Default) Times New Roman

**Page 10: [269] Formatted**      **Microsoft Office User**      **8/6/20 3:50:00 PM**

Font: (Default) Times New Roman

**Page 10: [270] Formatted**      **Microsoft Office User**      **8/6/20 3:50:00 PM**

Font: (Default) Times New Roman

**Page 10: [271] Formatted**      **Microsoft Office User**      **8/6/20 3:50:00 PM**

Font: (Default) Times New Roman

**Page 10: [272] Formatted**      **Microsoft Office User**      **8/6/20 3:50:00 PM**

Font: (Default) Times New Roman

**Page 10: [273] Formatted**      **Microsoft Office User**      **8/6/20 3:50:00 PM**

Font: (Default) Times New Roman

**Page 10: [274] Formatted**      **Microsoft Office User**      **8/6/20 3:50:00 PM**

Font: (Default) Times New Roman

**Page 10: [275] Formatted**      **Microsoft Office User**      **8/6/20 3:50:00 PM**

Font: (Default) Times New Roman

**Page 10: [276] Formatted**      **Microsoft Office User**      **8/6/20 3:50:00 PM**

Font: (Default) Times New Roman

**Page 10: [277] Formatted**      **Microsoft Office User**      **8/6/20 3:50:00 PM**

Font: (Default) Times New Roman

**Page 10: [278] Formatted**      **Microsoft Office User**      **8/6/20 3:50:00 PM**

Font: (Default) Times New Roman

**Page 10: [279] Formatted**      **Microsoft Office User**      **8/6/20 3:50:00 PM**

Font: (Default) Times New Roman

**Page 10: [280] Formatted**      **Microsoft Office User**      **8/6/20 3:50:00 PM**

Font: (Default) Times New Roman

**Page 10: [281] Formatted**      **Microsoft Office User**      **8/6/20 3:50:00 PM**

| Page 10: [282] Formatted | Microsoft Office User | 8/6/20 3:50:00 PM |
|---|---|---|

Font: (Default) Times New Roman

| Page 10: [283] Formatted | Microsoft Office User | 8/6/20 3:50:00 PM |
|---|---|---|

Font: (Default) Times New Roman

| Page 10: [284] Formatted | Microsoft Office User | 8/6/20 3:50:00 PM |
|---|---|---|

Font: (Default) Times New Roman

| Page 10: [285] Formatted | Microsoft Office User | 8/6/20 3:50:00 PM |
|---|---|---|

Font: (Default) Times New Roman

| Page 10: [286] Formatted | Microsoft Office User | 8/6/20 3:50:00 PM |
|---|---|---|

Font: (Default) Times New Roman

| Page 10: [287] Formatted | Microsoft Office User | 8/6/20 3:50:00 PM |
|---|---|---|

Font: (Default) Times New Roman

| Page 10: [288] Formatted | Microsoft Office User | 8/6/20 3:50:00 PM |
|---|---|---|

Font: (Default) Times New Roman

| Page 10: [289] Formatted | Microsoft Office User | 8/6/20 3:50:00 PM |
|---|---|---|

Font: (Default) Times New Roman

| Page 10: [290] Formatted | Microsoft Office User | 8/6/20 3:50:00 PM |
|---|---|---|

Font: (Default) Times New Roman

| Page 10: [291] Formatted | Microsoft Office User | 8/6/20 3:50:00 PM |
|---|---|---|

Font: (Default) Times New Roman

| Page 10: [292] Formatted | Microsoft Office User | 8/6/20 3:50:00 PM |
|---|---|---|

Font: (Default) Times New Roman

| Page 10: [293] Formatted | Microsoft Office User | 8/6/20 3:50:00 PM |
|---|---|---|

Font: (Default) Times New Roman

| Page 10: [294] Formatted | Microsoft Office User | 8/6/20 3:50:00 PM |
|---|---|---|

Font: (Default) Times New Roman

| Page 10: [295] Formatted | Microsoft Office User | 8/6/20 3:50:00 PM |
|---|---|---|

**Page 10: [296] Formatted**        **Microsoft Office User**        **8/6/20 3:50:00 PM**

Font: (Default) Times New Roman

**Page 10: [297] Formatted**        **Microsoft Office User**        **8/6/20 3:50:00 PM**

Font: (Default) Times New Roman

**Page 10: [298] Formatted**        **Microsoft Office User**        **8/6/20 3:50:00 PM**

Font: (Default) Times New Roman

**Page 10: [299] Formatted**        **Microsoft Office User**        **8/6/20 3:50:00 PM**

Font: (Default) Times New Roman

**Page 10: [300] Formatted**        **Microsoft Office User**        **8/6/20 3:50:00 PM**

Font: (Default) Times New Roman

**Page 10: [301] Formatted**        **Microsoft Office User**        **8/6/20 3:50:00 PM**

Font: (Default) Times New Roman

**Page 10: [302] Formatted**        **Microsoft Office User**        **8/6/20 3:50:00 PM**

Font: (Default) Times New Roman

**Page 10: [303] Formatted**        **Microsoft Office User**        **8/6/20 3:50:00 PM**

Font: (Default) Times New Roman

**Page 10: [304] Formatted**        **Microsoft Office User**        **8/6/20 3:50:00 PM**

Font: (Default) Times New Roman

**Page 10: [305] Formatted**        **Microsoft Office User**        **8/6/20 3:50:00 PM**

Font: (Default) Times New Roman

**Page 10: [306] Formatted**        **Microsoft Office User**        **8/6/20 3:50:00 PM**

Font: (Default) Times New Roman

**Page 10: [307] Formatted**        **Microsoft Office User**        **8/6/20 3:50:00 PM**

Font: (Default) Times New Roman

**Page 10: [308] Formatted**        **Microsoft Office User**        **8/6/20 3:50:00 PM**

Font: (Default) Times New Roman

**Page 10: [309] Formatted**        **Microsoft Office User**        **8/6/20 3:50:00 PM**

**Page 10: [310] Formatted**          **Microsoft Office User**          **8/6/20 3:50:00 PM**

Font: (Default) Times New Roman

**Page 10: [311] Formatted**          **Microsoft Office User**          **8/6/20 3:50:00 PM**

Font: (Default) Times New Roman

**Page 10: [312] Formatted**          **Microsoft Office User**          **8/6/20 3:50:00 PM**

Font: (Default) Times New Roman

**Page 10: [313] Formatted**          **Microsoft Office User**          **8/6/20 3:50:00 PM**

Font: (Default) Times New Roman

**Page 10: [314] Formatted**          **Microsoft Office User**          **8/6/20 3:50:00 PM**

Font: (Default) Times New Roman

**Page 10: [315] Formatted**          **Microsoft Office User**          **8/6/20 3:50:00 PM**

Font: (Default) Times New Roman

**Page 10: [316] Formatted**          **Microsoft Office User**          **8/6/20 3:50:00 PM**

Font: (Default) Times New Roman

**Page 10: [317] Formatted**          **Microsoft Office User**          **8/6/20 3:50:00 PM**

Font: (Default) Times New Roman

**Page 10: [318] Formatted**          **Microsoft Office User**          **8/6/20 3:50:00 PM**

Font: (Default) Times New Roman

**Page 10: [319] Formatted**          **Microsoft Office User**          **8/6/20 3:50:00 PM**

Font: (Default) Times New Roman

**Page 10: [320] Formatted**          **Microsoft Office User**          **8/6/20 3:50:00 PM**

Font: (Default) Times New Roman

**Page 10: [321] Formatted**          **Microsoft Office User**          **8/6/20 3:50:00 PM**

Font: (Default) Times New Roman

**Page 10: [322] Formatted**          **Microsoft Office User**          **8/6/20 3:50:00 PM**

Font: (Default) Times New Roman

**Page 10: [323] Formatted**          **Microsoft Office User**          **8/6/20 3:50:00 PM**

| Page 10: [324] Formatted | Microsoft Office User | 8/6/20 3:50:00 PM |
|---|---|---|

Font: (Default) Times New Roman

| Page 10: [325] Formatted | Microsoft Office User | 8/6/20 3:50:00 PM |
|---|---|---|

Font: (Default) Times New Roman

| Page 10: [326] Formatted | Microsoft Office User | 8/6/20 3:50:00 PM |
|---|---|---|

Font: (Default) Times New Roman

| Page 10: [327] Formatted | Microsoft Office User | 8/6/20 3:50:00 PM |
|---|---|---|

Font: (Default) Times New Roman

| Page 10: [328] Formatted | Microsoft Office User | 8/6/20 3:50:00 PM |
|---|---|---|

Font: (Default) Times New Roman

| Page 10: [329] Formatted | Microsoft Office User | 8/6/20 3:50:00 PM |
|---|---|---|

Font: (Default) Times New Roman

| Page 10: [330] Formatted | Microsoft Office User | 8/6/20 3:50:00 PM |
|---|---|---|

Font: (Default) Times New Roman

| Page 10: [331] Formatted | Microsoft Office User | 8/6/20 3:50:00 PM |
|---|---|---|

Font: (Default) Times New Roman

| Page 10: [332] Formatted | Microsoft Office User | 8/6/20 3:50:00 PM |
|---|---|---|

Font: (Default) Times New Roman

| Page 10: [333] Formatted | Microsoft Office User | 8/6/20 3:50:00 PM |
|---|---|---|

Font: (Default) Times New Roman

| Page 10: [334] Formatted | Microsoft Office User | 8/6/20 3:50:00 PM |
|---|---|---|

Font: (Default) Times New Roman

| Page 10: [335] Formatted | Microsoft Office User | 8/6/20 3:50:00 PM |
|---|---|---|

Font: (Default) Times New Roman

| Page 10: [336] Formatted | Microsoft Office User | 8/6/20 3:50:00 PM |
|---|---|---|

Font: (Default) Times New Roman

| Page 10: [337] Formatted | Microsoft Office User | 8/6/20 3:50:00 PM |
|---|---|---|

| Page 10: [338] Formatted | Microsoft Office User | 8/6/20 3:50:00 PM |
|---|---|---|

Font: (Default) Times New Roman

| Page 10: [339] Formatted | Microsoft Office User | 8/6/20 3:50:00 PM |
|---|---|---|

Font: (Default) Times New Roman

| Page 10: [340] Formatted | Microsoft Office User | 8/6/20 3:50:00 PM |
|---|---|---|

Font: (Default) Times New Roman

| Page 10: [341] Formatted | Microsoft Office User | 8/6/20 3:50:00 PM |
|---|---|---|

Font: (Default) Times New Roman

| Page 10: [342] Formatted | Microsoft Office User | 8/6/20 3:50:00 PM |
|---|---|---|

Font: (Default) Times New Roman

| Page 10: [343] Formatted | Microsoft Office User | 8/6/20 3:50:00 PM |
|---|---|---|

Font: (Default) Times New Roman

| Page 10: [344] Formatted | Microsoft Office User | 8/6/20 3:50:00 PM |
|---|---|---|

Font: (Default) Times New Roman

| Page 10: [345] Formatted | Microsoft Office User | 8/6/20 3:50:00 PM |
|---|---|---|

Font: (Default) Times New Roman

| Page 10: [346] Formatted | Microsoft Office User | 8/6/20 3:50:00 PM |
|---|---|---|

Font: (Default) Times New Roman

| Page 10: [347] Formatted | Microsoft Office User | 8/6/20 3:50:00 PM |
|---|---|---|

Font: (Default) Times New Roman

| Page 10: [348] Formatted | Microsoft Office User | 8/6/20 3:50:00 PM |
|---|---|---|

Font: (Default) Times New Roman

| Page 10: [349] Formatted | Microsoft Office User | 8/6/20 3:50:00 PM |
|---|---|---|

Font: (Default) Times New Roman

| Page 10: [350] Formatted | Microsoft Office User | 8/6/20 3:50:00 PM |
|---|---|---|

Font: (Default) Times New Roman

| Page 10: [351] Formatted | Microsoft Office User | 8/6/20 3:50:00 PM |
|---|---|---|

**Page 10: [352] Formatted** | **Microsoft Office User** | **8/6/20 3:50:00 PM**

Font: (Default) Times New Roman

**Page 10: [353] Formatted** | **Microsoft Office User** | **8/6/20 3:50:00 PM**

Font: (Default) Times New Roman

**Page 10: [354] Formatted** | **Microsoft Office User** | **8/6/20 3:50:00 PM**

Font: (Default) Times New Roman

**Page 10: [355] Formatted** | **Microsoft Office User** | **8/6/20 3:50:00 PM**

Font: (Default) Times New Roman

**Page 10: [356] Formatted** | **Microsoft Office User** | **8/6/20 3:50:00 PM**

Font: (Default) Times New Roman

**Page 10: [357] Formatted** | **Microsoft Office User** | **8/6/20 3:50:00 PM**

Font: (Default) Times New Roman

**Page 10: [358] Formatted** | **Microsoft Office User** | **8/6/20 3:50:00 PM**

Font: (Default) Times New Roman

**Page 10: [359] Formatted** | **Microsoft Office User** | **8/6/20 3:50:00 PM**

Font: (Default) Times New Roman

**Page 10: [360] Formatted** | **Microsoft Office User** | **8/6/20 3:50:00 PM**

Font: (Default) Times New Roman

**Page 10: [361] Formatted** | **Microsoft Office User** | **8/6/20 3:50:00 PM**

Font: (Default) Times New Roman

**Page 10: [362] Formatted** | **Microsoft Office User** | **8/6/20 3:50:00 PM**

Font: (Default) Times New Roman

**Page 10: [363] Formatted** | **Microsoft Office User** | **8/6/20 3:50:00 PM**

Font: (Default) Times New Roman

**Page 10: [364] Formatted** | **Microsoft Office User** | **8/6/20 3:50:00 PM**

Font: (Default) Times New Roman

**Page 10: [365] Formatted** | **Microsoft Office User** | **8/6/20 3:50:00 PM**

**Page 10: [366] Formatted**      **Microsoft Office User**      **8/6/20 3:50:00 PM**

Font: (Default) Times New Roman

**Page 10: [367] Formatted**      **Microsoft Office User**      **8/6/20 3:50:00 PM**

Font: (Default) Times New Roman

**Page 10: [368] Formatted**      **Microsoft Office User**      **8/6/20 3:50:00 PM**

Font: (Default) Times New Roman

**Page 10: [369] Formatted**      **Microsoft Office User**      **8/6/20 3:50:00 PM**

Font: (Default) Times New Roman

**Page 10: [370] Formatted**      **Microsoft Office User**      **8/6/20 3:50:00 PM**

Font: (Default) Times New Roman

**Page 10: [371] Formatted**      **Microsoft Office User**      **8/6/20 3:50:00 PM**

Font: (Default) Times New Roman

**Page 10: [372] Formatted**      **Microsoft Office User**      **8/6/20 3:50:00 PM**

Font: (Default) Times New Roman

**Page 10: [373] Formatted**      **Microsoft Office User**      **8/6/20 3:50:00 PM**

Font: (Default) Times New Roman

**Page 10: [374] Formatted**      **Microsoft Office User**      **8/6/20 3:50:00 PM**

Font: (Default) Times New Roman

**Page 10: [375] Formatted**      **Microsoft Office User**      **8/6/20 3:50:00 PM**

Font: (Default) Times New Roman

**Page 10: [376] Formatted**      **Microsoft Office User**      **8/6/20 3:50:00 PM**

Font: (Default) Times New Roman

**Page 10: [377] Formatted**      **Microsoft Office User**      **8/6/20 3:50:00 PM**

Font: (Default) Times New Roman

**Page 10: [378] Formatted**      **Microsoft Office User**      **8/6/20 3:50:00 PM**

Font: (Default) Times New Roman

**Page 10: [379] Formatted**      **Microsoft Office User**      **8/6/20 3:50:00 PM**

**Page 10: [380] Formatted**      **Microsoft Office User**      **8/6/20 3:50:00 PM**

Font: (Default) Times New Roman

**Page 10: [381] Formatted**      **Microsoft Office User**      **8/6/20 3:50:00 PM**

Font: (Default) Times New Roman

**Page 10: [382] Formatted**      **Microsoft Office User**      **8/6/20 3:50:00 PM**

Font: (Default) Times New Roman

**Page 10: [383] Formatted**      **Microsoft Office User**      **8/6/20 3:50:00 PM**

Font: (Default) Times New Roman

**Page 10: [384] Formatted**      **Microsoft Office User**      **8/6/20 3:50:00 PM**

Font: (Default) Times New Roman

**Page 10: [385] Formatted**      **Microsoft Office User**      **8/6/20 3:50:00 PM**

Font: (Default) Times New Roman

**Page 10: [386] Formatted**      **Microsoft Office User**      **8/6/20 3:50:00 PM**

Font: (Default) Times New Roman

**Page 10: [387] Formatted**      **Microsoft Office User**      **8/6/20 3:50:00 PM**

Font: (Default) Times New Roman

**Page 10: [388] Formatted**      **Microsoft Office User**      **8/6/20 3:50:00 PM**

Font: (Default) Times New Roman

**Page 10: [389] Formatted**      **Microsoft Office User**      **8/6/20 3:50:00 PM**

Font: (Default) Times New Roman

**Page 10: [390] Formatted**      **Microsoft Office User**      **8/6/20 3:50:00 PM**

Font: (Default) Times New Roman

**Page 10: [391] Formatted**      **Microsoft Office User**      **8/6/20 3:50:00 PM**

Font: (Default) Times New Roman

**Page 10: [392] Formatted**      **Microsoft Office User**      **8/6/20 3:50:00 PM**

Font: (Default) Times New Roman

**Page 10: [393] Formatted**      **Microsoft Office User**      **8/6/20 3:50:00 PM**

| Page 10: [394] Formatted | Microsoft Office User | 8/6/20 3:50:00 PM |
|---|---|---|

Font: (Default) Times New Roman

| Page 10: [395] Formatted | Microsoft Office User | 8/6/20 3:50:00 PM |
|---|---|---|

Font: (Default) Times New Roman

| Page 10: [396] Formatted | Microsoft Office User | 8/6/20 3:50:00 PM |
|---|---|---|

Font: (Default) Times New Roman

| Page 10: [397] Formatted | Microsoft Office User | 8/6/20 3:50:00 PM |
|---|---|---|

Font: (Default) Times New Roman

| Page 10: [398] Formatted | Microsoft Office User | 8/6/20 3:50:00 PM |
|---|---|---|

Font: (Default) Times New Roman

| Page 10: [399] Formatted | Microsoft Office User | 8/6/20 3:50:00 PM |
|---|---|---|

Font: (Default) Times New Roman

| Page 10: [400] Formatted | Microsoft Office User | 8/6/20 3:50:00 PM |
|---|---|---|

Font: (Default) Times New Roman

| Page 10: [401] Formatted | Microsoft Office User | 8/6/20 3:50:00 PM |
|---|---|---|

Font: (Default) Times New Roman

| Page 10: [402] Formatted | Microsoft Office User | 8/6/20 3:50:00 PM |
|---|---|---|

Font: (Default) Times New Roman

| Page 10: [403] Formatted | Microsoft Office User | 8/6/20 3:50:00 PM |
|---|---|---|

Font: (Default) Times New Roman

| Page 10: [404] Formatted | Microsoft Office User | 8/6/20 3:50:00 PM |
|---|---|---|

Font: (Default) Times New Roman

| Page 10: [405] Formatted | Microsoft Office User | 8/6/20 3:50:00 PM |
|---|---|---|

Font: (Default) Times New Roman

| Page 10: [406] Formatted | Microsoft Office User | 8/6/20 3:50:00 PM |
|---|---|---|

Font: (Default) Times New Roman

| Page 10: [407] Formatted | Microsoft Office User | 8/6/20 3:50:00 PM |
|---|---|---|

**Page 10: [408] Formatted** | **Microsoft Office User** | **8/6/20 3:50:00 PM**

Font: (Default) Times New Roman

**Page 10: [409] Formatted** | **Microsoft Office User** | **8/6/20 3:50:00 PM**

Font: (Default) Times New Roman

**Page 10: [410] Formatted** | **Microsoft Office User** | **8/6/20 3:50:00 PM**

Font: (Default) Times New Roman

**Page 10: [411] Formatted** | **Microsoft Office User** | **8/6/20 3:50:00 PM**

Font: (Default) Times New Roman

**Page 10: [412] Formatted** | **Microsoft Office User** | **8/6/20 3:50:00 PM**

Font: (Default) Times New Roman

**Page 10: [413] Formatted** | **Microsoft Office User** | **8/6/20 3:50:00 PM**

Font: (Default) Times New Roman

**Page 10: [414] Formatted** | **Microsoft Office User** | **8/6/20 3:50:00 PM**

Font: (Default) Times New Roman

**Page 10: [415] Formatted** | **Microsoft Office User** | **8/6/20 3:50:00 PM**

Font: (Default) Times New Roman

**Page 10: [416] Formatted** | **Microsoft Office User** | **8/6/20 3:50:00 PM**

Font: (Default) Times New Roman

**Page 10: [417] Formatted** | **Microsoft Office User** | **8/6/20 3:50:00 PM**

Font: (Default) Times New Roman

**Page 10: [418] Formatted** | **Microsoft Office User** | **8/6/20 3:50:00 PM**

Font: (Default) Times New Roman

**Page 10: [419] Formatted** | **Microsoft Office User** | **8/6/20 3:50:00 PM**

Font: (Default) Times New Roman

**Page 10: [420] Formatted** | **Microsoft Office User** | **8/6/20 3:50:00 PM**

Font: (Default) Times New Roman

**Page 10: [421] Formatted** | **Microsoft Office User** | **8/6/20 3:50:00 PM**

| Page 10: [422] Formatted | Microsoft Office User | 8/6/20 3:50:00 PM |
|---|---|---|

Font: (Default) Times New Roman

| Page 10: [423] Formatted | Microsoft Office User | 8/6/20 3:50:00 PM |
|---|---|---|

Font: (Default) Times New Roman

| Page 10: [424] Formatted | Microsoft Office User | 8/6/20 3:50:00 PM |
|---|---|---|

Font: (Default) Times New Roman

| Page 10: [425] Formatted | Microsoft Office User | 8/6/20 3:50:00 PM |
|---|---|---|

Font: (Default) Times New Roman

| Page 10: [426] Formatted | Microsoft Office User | 8/6/20 3:50:00 PM |
|---|---|---|

Font: (Default) Times New Roman

| Page 10: [427] Formatted | Microsoft Office User | 8/6/20 3:50:00 PM |
|---|---|---|

Font: (Default) Times New Roman

| Page 10: [428] Formatted | Microsoft Office User | 8/6/20 3:50:00 PM |
|---|---|---|

Font: (Default) Times New Roman

| Page 10: [429] Formatted | Microsoft Office User | 8/6/20 3:50:00 PM |
|---|---|---|

Font: (Default) Times New Roman

| Page 10: [430] Formatted | Microsoft Office User | 8/6/20 3:50:00 PM |
|---|---|---|

Font: (Default) Times New Roman

| Page 10: [431] Formatted | Microsoft Office User | 8/6/20 3:50:00 PM |
|---|---|---|

Font: (Default) Times New Roman

| Page 10: [432] Formatted | Microsoft Office User | 8/6/20 3:50:00 PM |
|---|---|---|

Font: (Default) Times New Roman

| Page 10: [433] Formatted | Microsoft Office User | 8/6/20 3:50:00 PM |
|---|---|---|

Font: (Default) Times New Roman

| Page 10: [434] Formatted | Microsoft Office User | 8/6/20 3:50:00 PM |
|---|---|---|

Font: (Default) Times New Roman

| Page 10: [435] Formatted | Microsoft Office User | 8/6/20 3:50:00 PM |
|---|---|---|

| Page 10: [436] Formatted | Microsoft Office User | 8/6/20 3:50:00 PM |
|---|---|---|

Font: (Default) Times New Roman

| Page 10: [437] Formatted | Microsoft Office User | 8/6/20 3:50:00 PM |
|---|---|---|

Font: (Default) Times New Roman

| Page 10: [438] Formatted | Microsoft Office User | 8/6/20 3:50:00 PM |
|---|---|---|

Font: (Default) Times New Roman

| Page 10: [439] Formatted | Microsoft Office User | 8/6/20 3:50:00 PM |
|---|---|---|

Font: (Default) Times New Roman

| Page 10: [440] Formatted | Microsoft Office User | 8/6/20 3:50:00 PM |
|---|---|---|

Font: (Default) Times New Roman

| Page 10: [441] Formatted | Microsoft Office User | 8/6/20 3:50:00 PM |
|---|---|---|

Font: (Default) Times New Roman

| Page 10: [442] Formatted | Microsoft Office User | 8/6/20 3:50:00 PM |
|---|---|---|

Font: (Default) Times New Roman

| Page 10: [443] Formatted | Microsoft Office User | 8/6/20 3:50:00 PM |
|---|---|---|

Font: (Default) Times New Roman

| Page 10: [444] Formatted | Microsoft Office User | 8/6/20 3:50:00 PM |
|---|---|---|

Font: (Default) Times New Roman

| Page 10: [445] Formatted | Microsoft Office User | 8/6/20 3:50:00 PM |
|---|---|---|

Font: (Default) Times New Roman

| Page 10: [446] Formatted | Microsoft Office User | 8/6/20 3:50:00 PM |
|---|---|---|

Font: (Default) Times New Roman

| Page 10: [447] Formatted | Microsoft Office User | 8/6/20 3:50:00 PM |
|---|---|---|

Font: (Default) Times New Roman

| Page 10: [448] Formatted | Microsoft Office User | 8/6/20 3:50:00 PM |
|---|---|---|

Font: (Default) Times New Roman

| Page 10: [449] Formatted | Microsoft Office User | 8/6/20 3:50:00 PM |
|---|---|---|

**Page 10: [450] Formatted**       **Microsoft Office User**       **8/6/20 3:50:00 PM**

Font: (Default) Times New Roman

**Page 10: [451] Formatted**       **Microsoft Office User**       **8/6/20 3:50:00 PM**

Font: (Default) Times New Roman

**Page 10: [452] Formatted**       **Microsoft Office User**       **8/6/20 3:50:00 PM**

Font: (Default) Times New Roman

**Page 10: [453] Formatted**       **Microsoft Office User**       **8/6/20 3:50:00 PM**

Font: (Default) Times New Roman

**Page 10: [454] Formatted**       **Microsoft Office User**       **8/6/20 3:50:00 PM**

Font: (Default) Times New Roman

**Page 10: [455] Formatted**       **Microsoft Office User**       **8/6/20 3:50:00 PM**

Font: (Default) Times New Roman

**Page 10: [456] Formatted**       **Microsoft Office User**       **8/6/20 3:50:00 PM**

Font: (Default) Times New Roman

**Page 10: [457] Formatted**       **Microsoft Office User**       **8/6/20 3:50:00 PM**

Font: (Default) Times New Roman

**Page 10: [458] Formatted**       **Microsoft Office User**       **8/6/20 3:50:00 PM**

Font: (Default) Times New Roman

**Page 10: [459] Formatted**       **Microsoft Office User**       **8/6/20 3:50:00 PM**

Font: (Default) Times New Roman

**Page 10: [460] Formatted**       **Microsoft Office User**       **8/6/20 3:50:00 PM**

Font: (Default) Times New Roman

**Page 10: [461] Formatted**       **Microsoft Office User**       **8/6/20 3:50:00 PM**

Font: (Default) Times New Roman

**Page 10: [462] Formatted**       **Microsoft Office User**       **8/6/20 3:50:00 PM**

Font: (Default) Times New Roman

**Page 10: [463] Formatted**       **Microsoft Office User**       **8/6/20 3:50:00 PM**

**Page 10: [464] Formatted** | **Microsoft Office User** | **8/6/20 3:50:00 PM**

Font: (Default) Times New Roman

**Page 10: [465] Formatted** | **Microsoft Office User** | **8/6/20 3:50:00 PM**

Font: (Default) Times New Roman

**Page 10: [466] Formatted** | **Microsoft Office User** | **8/6/20 3:50:00 PM**

Font: (Default) Times New Roman

**Page 10: [467] Formatted** | **Microsoft Office User** | **8/6/20 3:50:00 PM**

Font: (Default) Times New Roman

**Page 10: [468] Formatted** | **Microsoft Office User** | **8/6/20 3:50:00 PM**

Font: (Default) Times New Roman

**Page 10: [469] Formatted** | **Microsoft Office User** | **8/6/20 3:50:00 PM**

Font: (Default) Times New Roman

**Page 10: [470] Formatted** | **Microsoft Office User** | **8/6/20 3:50:00 PM**

Font: (Default) Times New Roman

**Page 10: [471] Formatted** | **Microsoft Office User** | **8/6/20 3:50:00 PM**

Font: (Default) Times New Roman

**Page 10: [472] Formatted** | **Microsoft Office User** | **8/6/20 3:50:00 PM**

Font: (Default) Times New Roman

**Page 10: [473] Formatted** | **Microsoft Office User** | **8/6/20 3:50:00 PM**

Font: (Default) Times New Roman

**Page 10: [474] Formatted** | **Microsoft Office User** | **8/6/20 3:50:00 PM**

Font: (Default) Times New Roman

**Page 10: [475] Formatted** | **Microsoft Office User** | **8/6/20 3:50:00 PM**

Font: (Default) Times New Roman

**Page 10: [476] Formatted** | **Microsoft Office User** | **8/6/20 3:50:00 PM**

Font: (Default) Times New Roman

**Page 10: [477] Formatted** | **Microsoft Office User** | **8/6/20 3:50:00 PM**

| Page 10: [478] Formatted | Microsoft Office User | 8/6/20 3:50:00 PM |
|---|---|---|

Font: (Default) Times New Roman

| Page 10: [479] Formatted | Microsoft Office User | 8/6/20 3:50:00 PM |
|---|---|---|

Font: (Default) Times New Roman

| Page 10: [480] Formatted | Microsoft Office User | 8/6/20 3:50:00 PM |
|---|---|---|

Font: (Default) Times New Roman

| Page 10: [481] Formatted | Microsoft Office User | 8/6/20 3:50:00 PM |
|---|---|---|

Font: (Default) Times New Roman

| Page 10: [482] Formatted | Microsoft Office User | 8/6/20 3:50:00 PM |
|---|---|---|

Font: (Default) Times New Roman

| Page 10: [483] Formatted | Microsoft Office User | 8/6/20 3:50:00 PM |
|---|---|---|

Font: (Default) Times New Roman

| Page 10: [484] Formatted | Microsoft Office User | 8/6/20 3:50:00 PM |
|---|---|---|

Font: (Default) Times New Roman

| Page 10: [485] Formatted | Microsoft Office User | 8/6/20 3:50:00 PM |
|---|---|---|

Font: (Default) Times New Roman

| Page 10: [486] Formatted | Microsoft Office User | 8/6/20 3:50:00 PM |
|---|---|---|

Font: (Default) Times New Roman

| Page 10: [487] Formatted | Microsoft Office User | 8/6/20 3:50:00 PM |
|---|---|---|

Font: (Default) Times New Roman

| Page 10: [488] Formatted | Microsoft Office User | 8/6/20 3:50:00 PM |
|---|---|---|

Font: (Default) Times New Roman

| Page 10: [489] Formatted | Microsoft Office User | 8/6/20 3:50:00 PM |
|---|---|---|

Font: (Default) Times New Roman

| Page 10: [490] Formatted | Microsoft Office User | 8/6/20 3:50:00 PM |
|---|---|---|

Font: (Default) Times New Roman

| Page 10: [491] Formatted | Microsoft Office User | 8/6/20 3:50:00 PM |
|---|---|---|

**Page 10: [492] Formatted** | **Microsoft Office User** | **8/6/20 3:50:00 PM**

Font: (Default) Times New Roman

**Page 10: [493] Formatted** | **Microsoft Office User** | **8/6/20 3:50:00 PM**

Font: (Default) Times New Roman

**Page 10: [494] Formatted** | **Microsoft Office User** | **8/6/20 3:50:00 PM**

Font: (Default) Times New Roman

**Page 10: [495] Formatted** | **Microsoft Office User** | **8/6/20 3:50:00 PM**

Font: (Default) Times New Roman

**Page 10: [496] Formatted** | **Microsoft Office User** | **8/6/20 3:50:00 PM**

Font: (Default) Times New Roman

**Page 10: [497] Formatted** | **Microsoft Office User** | **8/6/20 3:50:00 PM**

Font: (Default) Times New Roman

**Page 10: [498] Formatted** | **Microsoft Office User** | **8/6/20 3:50:00 PM**

Font: (Default) Times New Roman

**Page 10: [499] Formatted** | **Microsoft Office User** | **8/6/20 3:50:00 PM**

Font: (Default) Times New Roman

**Page 10: [500] Formatted** | **Microsoft Office User** | **8/6/20 3:50:00 PM**

Font: (Default) Times New Roman

**Page 10: [501] Formatted** | **Microsoft Office User** | **8/6/20 3:50:00 PM**

Font: (Default) Times New Roman

**Page 10: [502] Formatted** | **Microsoft Office User** | **8/6/20 3:50:00 PM**

Font: (Default) Times New Roman

**Page 10: [503] Formatted** | **Microsoft Office User** | **8/6/20 3:50:00 PM**

Font: (Default) Times New Roman

**Page 10: [504] Formatted** | **Microsoft Office User** | **8/6/20 3:50:00 PM**

Font: (Default) Times New Roman

**Page 10: [505] Formatted** | **Microsoft Office User** | **8/6/20 3:50:00 PM**

| Page 10: [506] Formatted | Microsoft Office User | 8/6/20 3:50:00 PM |
|---|---|---|

Font: (Default) Times New Roman

| Page 10: [507] Formatted | Microsoft Office User | 8/6/20 3:50:00 PM |
|---|---|---|

Font: (Default) Times New Roman

| Page 10: [508] Formatted | Microsoft Office User | 8/6/20 3:50:00 PM |
|---|---|---|

Font: (Default) Times New Roman

| Page 10: [509] Formatted | Microsoft Office User | 8/6/20 3:50:00 PM |
|---|---|---|

Font: (Default) Times New Roman

| Page 10: [510] Formatted | Microsoft Office User | 8/6/20 3:50:00 PM |
|---|---|---|

Font: (Default) Times New Roman

| Page 10: [511] Formatted | Microsoft Office User | 8/6/20 3:50:00 PM |
|---|---|---|

Font: (Default) Times New Roman

| Page 10: [512] Formatted | Microsoft Office User | 8/6/20 3:50:00 PM |
|---|---|---|

Font: (Default) Times New Roman

| Page 10: [513] Formatted | Microsoft Office User | 8/6/20 3:50:00 PM |
|---|---|---|

Font: (Default) Times New Roman

| Page 10: [514] Formatted | Microsoft Office User | 8/6/20 3:50:00 PM |
|---|---|---|

Font: (Default) Times New Roman

| Page 10: [515] Formatted | Microsoft Office User | 8/6/20 3:50:00 PM |
|---|---|---|

Font: (Default) Times New Roman

| Page 10: [516] Formatted | Microsoft Office User | 8/6/20 3:50:00 PM |
|---|---|---|

Font: (Default) Times New Roman

| Page 10: [517] Formatted | Microsoft Office User | 8/6/20 3:50:00 PM |
|---|---|---|

Font: (Default) Times New Roman

| Page 10: [518] Formatted | Microsoft Office User | 8/6/20 3:50:00 PM |
|---|---|---|

Font: (Default) Times New Roman

| Page 10: [519] Formatted | Microsoft Office User | 8/6/20 3:50:00 PM |
|---|---|---|

| Page 10: [520] Formatted | Microsoft Office User | 8/6/20 3:50:00 PM |
|---|---|---|

Font: (Default) Times New Roman

| Page 10: [521] Formatted | Microsoft Office User | 8/6/20 3:50:00 PM |
|---|---|---|

Font: (Default) Times New Roman

| Page 10: [522] Formatted | Microsoft Office User | 8/6/20 3:50:00 PM |
|---|---|---|

Font: (Default) Times New Roman

| Page 10: [523] Formatted | Microsoft Office User | 8/6/20 3:50:00 PM |
|---|---|---|

Font: (Default) Times New Roman

| Page 10: [524] Formatted | Microsoft Office User | 8/6/20 3:50:00 PM |
|---|---|---|

Font: (Default) Times New Roman

| Page 10: [525] Formatted | Microsoft Office User | 8/6/20 3:50:00 PM |
|---|---|---|

Font: (Default) Times New Roman

| Page 10: [526] Formatted | Microsoft Office User | 8/6/20 3:50:00 PM |
|---|---|---|

Font: (Default) Times New Roman

| Page 10: [527] Formatted | Microsoft Office User | 8/6/20 3:50:00 PM |
|---|---|---|

Font: (Default) Times New Roman

| Page 10: [528] Formatted | Microsoft Office User | 8/6/20 3:50:00 PM |
|---|---|---|

Font: (Default) Times New Roman

| Page 10: [529] Formatted | Microsoft Office User | 8/6/20 3:50:00 PM |
|---|---|---|

Font: (Default) Times New Roman

| Page 10: [530] Formatted | Microsoft Office User | 8/6/20 3:50:00 PM |
|---|---|---|

Font: (Default) Times New Roman

| Page 10: [531] Formatted | Microsoft Office User | 8/6/20 3:50:00 PM |
|---|---|---|

Font: (Default) Times New Roman

| Page 10: [532] Formatted | Microsoft Office User | 8/6/20 3:50:00 PM |
|---|---|---|

Font: (Default) Times New Roman

| Page 10: [533] Formatted | Microsoft Office User | 8/6/20 3:50:00 PM |
|---|---|---|

**Page 10: [534] Formatted**       **Microsoft Office User**       **8/6/20 3:50:00 PM**

Font: (Default) Times New Roman

**Page 10: [535] Formatted**       **Microsoft Office User**       **8/6/20 3:50:00 PM**

Font: (Default) Times New Roman

**Page 10: [536] Formatted**       **Microsoft Office User**       **8/6/20 3:50:00 PM**

Font: (Default) Times New Roman

**Page 10: [537] Formatted**       **Microsoft Office User**       **8/6/20 3:50:00 PM**

Font: (Default) Times New Roman

**Page 10: [538] Formatted**       **Microsoft Office User**       **8/6/20 3:50:00 PM**

Font: (Default) Times New Roman

**Page 10: [539] Formatted**       **Microsoft Office User**       **8/6/20 3:50:00 PM**

Font: (Default) Times New Roman

**Page 10: [540] Formatted**       **Microsoft Office User**       **8/6/20 3:50:00 PM**

Font: (Default) Times New Roman

**Page 10: [541] Formatted**       **Microsoft Office User**       **8/6/20 3:50:00 PM**

Font: (Default) Times New Roman

**Page 10: [542] Formatted**       **Microsoft Office User**       **8/6/20 3:50:00 PM**

Font: (Default) Times New Roman

**Page 10: [543] Formatted**       **Microsoft Office User**       **8/6/20 3:50:00 PM**

Font: (Default) Times New Roman

**Page 10: [544] Formatted**       **Microsoft Office User**       **8/6/20 3:50:00 PM**

Font: (Default) Times New Roman

**Page 10: [545] Formatted**       **Microsoft Office User**       **8/6/20 3:50:00 PM**

Font: (Default) Times New Roman

**Page 10: [546] Formatted**       **Microsoft Office User**       **8/6/20 3:50:00 PM**

Font: (Default) Times New Roman

**Page 10: [547] Formatted**       **Microsoft Office User**       **8/6/20 3:50:00 PM**

**Page 10: [548] Formatted**        **Microsoft Office User**        **8/6/20 3:50:00 PM**

Font: (Default) Times New Roman

**Page 10: [549] Formatted**        **Microsoft Office User**        **8/6/20 3:50:00 PM**

Font: (Default) Times New Roman

**Page 10: [550] Formatted**        **Microsoft Office User**        **8/6/20 3:50:00 PM**

Font: (Default) Times New Roman

**Page 10: [551] Formatted**        **Microsoft Office User**        **8/6/20 3:50:00 PM**

Font: (Default) Times New Roman

**Page 10: [552] Formatted**        **Microsoft Office User**        **8/6/20 3:50:00 PM**

Font: (Default) Times New Roman

**Page 10: [553] Formatted**        **Microsoft Office User**        **8/6/20 3:50:00 PM**

Font: (Default) Times New Roman

**Page 10: [554] Formatted**        **Microsoft Office User**        **8/6/20 3:50:00 PM**

Font: (Default) Times New Roman

**Page 10: [555] Formatted**        **Microsoft Office User**        **8/6/20 3:50:00 PM**

Font: (Default) Times New Roman

**Page 10: [556] Formatted**        **Microsoft Office User**        **8/6/20 3:50:00 PM**

Font: (Default) Times New Roman

**Page 10: [557] Formatted**        **Microsoft Office User**        **8/6/20 3:50:00 PM**

Font: (Default) Times New Roman

**Page 10: [558] Formatted**        **Microsoft Office User**        **8/6/20 3:50:00 PM**

Font: (Default) Times New Roman

**Page 10: [559] Formatted**        **Microsoft Office User**        **8/6/20 3:50:00 PM**

Font: (Default) Times New Roman

**Page 10: [560] Formatted**        **Microsoft Office User**        **8/6/20 3:50:00 PM**

Font: (Default) Times New Roman

**Page 10: [561] Formatted**        **Microsoft Office User**        **8/6/20 3:50:00 PM**

| Page 10: [562] Formatted | Microsoft Office User | 8/6/20 3:50:00 PM |
|---|---|---|

Font: (Default) Times New Roman

| Page 10: [563] Formatted | Microsoft Office User | 8/6/20 3:50:00 PM |
|---|---|---|

Font: (Default) Times New Roman

| Page 10: [564] Formatted | Microsoft Office User | 8/6/20 3:50:00 PM |
|---|---|---|

Font: (Default) Times New Roman

| Page 10: [565] Formatted | Microsoft Office User | 8/6/20 3:50:00 PM |
|---|---|---|

Font: (Default) Times New Roman

| Page 10: [566] Formatted | Microsoft Office User | 8/6/20 3:50:00 PM |
|---|---|---|

Font: (Default) Times New Roman

| Page 10: [567] Formatted | Microsoft Office User | 8/6/20 3:50:00 PM |
|---|---|---|

Font: (Default) Times New Roman

| Page 10: [568] Formatted | Microsoft Office User | 8/6/20 3:50:00 PM |
|---|---|---|

Font: (Default) Times New Roman

| Page 10: [569] Formatted | Microsoft Office User | 8/6/20 3:50:00 PM |
|---|---|---|

Font: (Default) Times New Roman

| Page 10: [570] Formatted | Microsoft Office User | 8/6/20 3:50:00 PM |
|---|---|---|

Font: (Default) Times New Roman

| Page 10: [571] Formatted | Microsoft Office User | 8/6/20 3:50:00 PM |
|---|---|---|

Font: (Default) Times New Roman

| Page 10: [572] Formatted | Microsoft Office User | 8/6/20 3:50:00 PM |
|---|---|---|

Font: (Default) Times New Roman

| Page 10: [573] Formatted | Microsoft Office User | 8/6/20 3:50:00 PM |
|---|---|---|

Font: (Default) Times New Roman

| Page 10: [574] Formatted | Microsoft Office User | 8/6/20 3:50:00 PM |
|---|---|---|

Font: (Default) Times New Roman

| Page 10: [575] Formatted | Microsoft Office User | 8/6/20 3:50:00 PM |
|---|---|---|

| Page 10: [576] Formatted | Microsoft Office User | 8/6/20 3:50:00 PM |

Font: (Default) Times New Roman

| Page 10: [577] Formatted | Microsoft Office User | 8/6/20 3:50:00 PM |

Font: (Default) Times New Roman

| Page 10: [578] Formatted | Microsoft Office User | 8/6/20 3:50:00 PM |

Font: (Default) Times New Roman

| Page 11: [579] Formatted | Microsoft Office User | 8/6/20 3:50:00 PM |

Font: (Default) Times New Roman

| Page 11: [580] Formatted | Microsoft Office User | 8/6/20 3:50:00 PM |

Font: (Default) Times New Roman

| Page 11: [581] Formatted | Microsoft Office User | 8/6/20 3:50:00 PM |

Font: (Default) Times New Roman

| Page 11: [582] Formatted | Microsoft Office User | 8/6/20 3:50:00 PM |

Font: (Default) Times New Roman

| Page 11: [583] Formatted | Microsoft Office User | 8/6/20 3:50:00 PM |

Font: (Default) Times New Roman

| Page 11: [584] Formatted | Microsoft Office User | 8/6/20 3:50:00 PM |

Font: (Default) Times New Roman

| Page 11: [585] Formatted | Microsoft Office User | 8/6/20 3:50:00 PM |

Font: (Default) Times New Roman

| Page 11: [586] Formatted | Microsoft Office User | 8/6/20 3:50:00 PM |

Font: (Default) Times New Roman

| Page 11: [587] Formatted | Microsoft Office User | 8/6/20 3:50:00 PM |

Font: (Default) Times New Roman

| Page 11: [588] Formatted | Microsoft Office User | 8/6/20 3:50:00 PM |

Font: (Default) Times New Roman

| Page 11: [589] Formatted | Microsoft Office User | 8/6/20 3:50:00 PM |

**Page 11: [590] Formatted**      **Microsoft Office User**      **8/6/20 3:50:00 PM**

Font: (Default) Times New Roman

**Page 11: [591] Formatted**      **Microsoft Office User**      **8/6/20 3:50:00 PM**

Font: (Default) Times New Roman

**Page 11: [592] Formatted**      **Microsoft Office User**      **8/6/20 3:50:00 PM**

Font: (Default) Times New Roman

**Page 11: [593] Formatted**      **Microsoft Office User**      **8/6/20 3:50:00 PM**

Font: (Default) Times New Roman

**Page 11: [594] Formatted**      **Microsoft Office User**      **8/6/20 3:50:00 PM**

Font: (Default) Times New Roman

**Page 11: [595] Formatted**      **Microsoft Office User**      **8/6/20 3:50:00 PM**

Font: (Default) Times New Roman

**Page 11: [596] Formatted**      **Microsoft Office User**      **8/6/20 3:50:00 PM**

Font: (Default) Times New Roman

**Page 11: [597] Formatted**      **Microsoft Office User**      **8/6/20 3:50:00 PM**

Font: (Default) Times New Roman

**Page 11: [598] Formatted**      **Microsoft Office User**      **8/6/20 3:50:00 PM**

Font: (Default) Times New Roman

**Page 11: [599] Formatted**      **Microsoft Office User**      **8/6/20 3:50:00 PM**

Font: (Default) Times New Roman

**Page 11: [600] Formatted**      **Microsoft Office User**      **8/6/20 3:50:00 PM**

Font: (Default) Times New Roman

**Page 11: [601] Formatted**      **Microsoft Office User**      **8/6/20 3:50:00 PM**

Font: (Default) Times New Roman

**Page 11: [602] Formatted**      **Microsoft Office User**      **8/6/20 3:50:00 PM**

Font: (Default) Times New Roman

**Page 11: [603] Formatted**      **Microsoft Office User**      **8/6/20 3:50:00 PM**

**Page 11: [604] Formatted**      **Microsoft Office User**      **8/6/20 3:50:00 PM**

Font: (Default) Times New Roman

**Page 11: [605] Formatted**      **Microsoft Office User**      **8/6/20 3:50:00 PM**

Font: (Default) Times New Roman

**Page 11: [606] Formatted**      **Microsoft Office User**      **8/6/20 3:50:00 PM**

Font: (Default) Times New Roman

**Page 11: [607] Formatted**      **Microsoft Office User**      **8/6/20 3:50:00 PM**

Font: (Default) Times New Roman

**Page 11: [608] Formatted**      **Microsoft Office User**      **8/6/20 3:50:00 PM**

Font: (Default) Times New Roman

**Page 11: [609] Formatted**      **Microsoft Office User**      **8/6/20 3:50:00 PM**

Font: (Default) Times New Roman

**Page 11: [610] Formatted**      **Microsoft Office User**      **8/6/20 3:50:00 PM**

Font: (Default) Times New Roman

**Page 11: [611] Formatted**      **Microsoft Office User**      **8/6/20 3:50:00 PM**

Font: (Default) Times New Roman

**Page 11: [612] Formatted**      **Microsoft Office User**      **8/6/20 3:50:00 PM**

Font: (Default) Times New Roman

**Page 11: [613] Formatted**      **Microsoft Office User**      **8/6/20 3:50:00 PM**

Font: (Default) Times New Roman

**Page 11: [614] Formatted**      **Microsoft Office User**      **8/6/20 3:50:00 PM**

Font: (Default) Times New Roman

**Page 11: [615] Formatted**      **Microsoft Office User**      **8/6/20 3:50:00 PM**

Font: (Default) Times New Roman

**Page 11: [616] Formatted**      **Microsoft Office User**      **8/6/20 3:50:00 PM**

Font: (Default) Times New Roman

**Page 11: [617] Formatted**      **Microsoft Office User**      **8/6/20 3:50:00 PM**

| Page 11: [618] Formatted | Microsoft Office User | 8/6/20 3:50:00 PM |
|---|---|---|

Font: (Default) Times New Roman

| Page 11: [619] Formatted | Microsoft Office User | 8/6/20 3:50:00 PM |
|---|---|---|

Font: (Default) Times New Roman

| Page 11: [620] Formatted | Microsoft Office User | 8/6/20 3:50:00 PM |
|---|---|---|

Font: (Default) Times New Roman

| Page 11: [621] Formatted | Microsoft Office User | 8/6/20 4:03:00 PM |
|---|---|---|

Caption

| Page 11: [622] Formatted | Microsoft Office User | 8/17/20 2:50:00 PM |
|---|---|---|

English (US)

| Page 11: [623] Formatted Table | Microsoft Office User | 8/20/20 6:10:00 PM |
|---|---|---|

Formatted Table

| Page 11: [624] Formatted | Microsoft Office User | 8/17/20 2:50:00 PM |
|---|---|---|

Font: Bold

| Page 11: [625] Formatted | Microsoft Office User | 8/6/20 3:56:00 PM |
|---|---|---|

Centered

| Page 11: [626] Formatted | Microsoft Office User | 8/17/20 2:50:00 PM |
|---|---|---|

Font: Bold

| Page 11: [627] Formatted | Microsoft Office User | 8/7/20 3:23:00 PM |
|---|---|---|

Centered, Don't add space between paragraphs of the same style, Line spacing:  single

| Page 11: [628] Formatted | Microsoft Office User | 8/17/20 2:50:00 PM |
|---|---|---|

Font color: Black

| Page 11: [629] Formatted | Microsoft Office User | 8/17/20 2:50:00 PM |
|---|---|---|

Font color: Black

| Page 11: [630] Formatted | Microsoft Office User | 8/17/20 2:50:00 PM |
|---|---|---|

Superscript

| Page 11: [630] Formatted | Microsoft Office User | 8/17/20 2:50:00 PM |
|---|---|---|

| Page 11: [631] Formatted | Microsoft Office User | 8/17/20 2:50:00 PM |
|---|---|---|

Font color: Black

| Page 11: [632] Formatted | Microsoft Office User | 8/17/20 2:50:00 PM |
|---|---|---|

Superscript

| Page 11: [632] Formatted | Microsoft Office User | 8/17/20 2:50:00 PM |
|---|---|---|

Superscript

| Page 11: [633] Formatted | Microsoft Office User | 8/6/20 3:49:00 PM |
|---|---|---|

Right, Don't add space between paragraphs of the same style, Line spacing:  single

| Page 11: [634] Formatted | Microsoft Office User | 8/17/20 2:50:00 PM |
|---|---|---|

Font color: Black

| Page 11: [635] Formatted | Microsoft Office User | 8/17/20 2:50:00 PM |
|---|---|---|

Font: (Default) Times New Roman

| Page 11: [635] Formatted | Microsoft Office User | 8/17/20 2:50:00 PM |
|---|---|---|

Font: (Default) Times New Roman

| Page 11: [636] Formatted | Microsoft Office User | 8/17/20 2:50:00 PM |
|---|---|---|

Font: (Default) Times New Roman

| Page 11: [636] Formatted | Microsoft Office User | 8/17/20 2:50:00 PM |
|---|---|---|

Font: (Default) Times New Roman

| Page 11: [637] Formatted | Microsoft Office User | 8/17/20 2:50:00 PM |
|---|---|---|

Font: (Default) Times New Roman

| Page 11: [637] Formatted | Microsoft Office User | 8/17/20 2:50:00 PM |
|---|---|---|

Font: (Default) Times New Roman

| Page 11: [638] Formatted | Microsoft Office User | 8/17/20 2:50:00 PM |
|---|---|---|

Font: (Default) Times New Roman

| Page 11: [638] Formatted | Microsoft Office User | 8/17/20 2:50:00 PM |
|---|---|---|

Font: (Default) Times New Roman

| Page 11: [639] Formatted | Microsoft Office User | 8/6/20 3:49:00 PM |
|---|---|---|

| Page 11: [640] Formatted | Microsoft Office User | 8/17/20 2:50:00 PM |
|---|---|---|

Font color: Black

| Page 11: [641] Formatted | Microsoft Office User | 8/17/20 2:50:00 PM |
|---|---|---|

Font: (Default) Times New Roman

| Page 11: [641] Formatted | Microsoft Office User | 8/17/20 2:50:00 PM |
|---|---|---|

Font: (Default) Times New Roman

| Page 11: [642] Formatted | Microsoft Office User | 8/17/20 2:50:00 PM |
|---|---|---|

Font: (Default) Times New Roman

| Page 11: [642] Formatted | Microsoft Office User | 8/17/20 2:50:00 PM |
|---|---|---|

Font: (Default) Times New Roman

| Page 11: [643] Formatted | Microsoft Office User | 8/17/20 2:50:00 PM |
|---|---|---|

Font: (Default) Times New Roman

| Page 11: [643] Formatted | Microsoft Office User | 8/17/20 2:50:00 PM |
|---|---|---|

Font: (Default) Times New Roman

| Page 11: [644] Formatted | Microsoft Office User | 8/17/20 2:50:00 PM |
|---|---|---|

Font: (Default) Times New Roman

| Page 11: [644] Formatted | Microsoft Office User | 8/17/20 2:50:00 PM |
|---|---|---|

Font: (Default) Times New Roman

| Page 11: [645] Formatted | Microsoft Office User | 8/6/20 3:49:00 PM |
|---|---|---|

Right, Don't add space between paragraphs of the same style, Line spacing:  single

| Page 11: [646] Formatted | Microsoft Office User | 8/17/20 2:50:00 PM |
|---|---|---|

Font color: Black, English (UK)

| Page 11: [647] Formatted | Microsoft Office User | 8/17/20 2:50:00 PM |
|---|---|---|

Font: (Default) Times New Roman

| Page 11: [647] Formatted | Microsoft Office User | 8/17/20 2:50:00 PM |
|---|---|---|

Font: (Default) Times New Roman

| Page 11: [648] Formatted | Microsoft Office User | 8/17/20 2:50:00 PM |
|---|---|---|

| Page 11: [648] Formatted | Microsoft Office User | 8/17/20 2:50:00 PM |

Font: (Default) Times New Roman

| Page 11: [649] Formatted | Microsoft Office User | 8/17/20 2:50:00 PM |

Font: (Default) Times New Roman

| Page 11: [649] Formatted | Microsoft Office User | 8/17/20 2:50:00 PM |

Font: (Default) Times New Roman

| Page 11: [650] Formatted | Microsoft Office User | 8/17/20 2:50:00 PM |

Font: (Default) Times New Roman

| Page 11: [650] Formatted | Microsoft Office User | 8/17/20 2:50:00 PM |

Font: (Default) Times New Roman

| Page 11: [651] Formatted | Microsoft Office User | 8/6/20 3:49:00 PM |

Right, Don't add space between paragraphs of the same style, Line spacing:  single

| Page 11: [652] Formatted | Microsoft Office User | 8/17/20 2:50:00 PM |

Font color: Black

| Page 11: [653] Formatted | Microsoft Office User | 8/17/20 2:50:00 PM |

Font: (Default) Times New Roman

| Page 11: [653] Formatted | Microsoft Office User | 8/17/20 2:50:00 PM |

Font: (Default) Times New Roman

| Page 11: [654] Formatted | Microsoft Office User | 8/17/20 2:50:00 PM |

Font: (Default) Times New Roman

| Page 11: [654] Formatted | Microsoft Office User | 8/17/20 2:50:00 PM |

Font: (Default) Times New Roman

| Page 11: [655] Formatted | Microsoft Office User | 8/17/20 2:50:00 PM |

Font: (Default) Times New Roman

| Page 11: [655] Formatted | Microsoft Office User | 8/17/20 2:50:00 PM |

Font: (Default) Times New Roman

| Page 11: [656] Formatted | Microsoft Office User | 8/17/20 2:50:00 PM |

| Page 11: [656] Formatted | Microsoft Office User | 8/17/20 2:50:00 PM |
|---|---|---|

Font: (Default) Times New Roman

| Page 11: [657] Formatted | Microsoft Office User | 8/6/20 3:49:00 PM |
|---|---|---|

Right, Don't add space between paragraphs of the same style, Line spacing:  single

| Page 11: [658] Formatted | Microsoft Office User | 8/17/20 2:50:00 PM |
|---|---|---|

Font color: Black

| Page 11: [659] Formatted | Microsoft Office User | 8/17/20 2:50:00 PM |
|---|---|---|

Font: (Default) Times New Roman

| Page 11: [659] Formatted | Microsoft Office User | 8/17/20 2:50:00 PM |
|---|---|---|

Font: (Default) Times New Roman

| Page 11: [660] Formatted | Microsoft Office User | 8/17/20 2:50:00 PM |
|---|---|---|

Font: (Default) Times New Roman

| Page 11: [660] Formatted | Microsoft Office User | 8/17/20 2:50:00 PM |
|---|---|---|

Font: (Default) Times New Roman

| Page 11: [661] Formatted | Microsoft Office User | 8/17/20 2:50:00 PM |
|---|---|---|

Font: (Default) Times New Roman

| Page 11: [661] Formatted | Microsoft Office User | 8/17/20 2:50:00 PM |
|---|---|---|

Font: (Default) Times New Roman

| Page 11: [662] Formatted | Microsoft Office User | 8/17/20 2:50:00 PM |
|---|---|---|

Font: (Default) Times New Roman

| Page 11: [662] Formatted | Microsoft Office User | 8/17/20 2:50:00 PM |
|---|---|---|

Font: (Default) Times New Roman

| Page 11: [663] Formatted | Microsoft Office User | 8/6/20 3:49:00 PM |
|---|---|---|

Right, Don't add space between paragraphs of the same style, Line spacing:  single

| Page 11: [664] Formatted | Microsoft Office User | 8/17/20 2:50:00 PM |
|---|---|---|

Font color: Black

| Page 11: [665] Formatted | Microsoft Office User | 8/17/20 2:50:00 PM |
|---|---|---|

| Page 11: [665] Formatted | Microsoft Office User | 8/17/20 2:50:00 PM |
|---|---|---|

Font: (Default) Times New Roman

| Page 11: [666] Formatted | Microsoft Office User | 8/17/20 2:50:00 PM |
|---|---|---|

Font: (Default) Times New Roman

| Page 11: [666] Formatted | Microsoft Office User | 8/17/20 2:50:00 PM |
|---|---|---|

Font: (Default) Times New Roman

| Page 11: [667] Formatted | Microsoft Office User | 8/17/20 2:50:00 PM |
|---|---|---|

Font: (Default) Times New Roman

| Page 11: [667] Formatted | Microsoft Office User | 8/17/20 2:50:00 PM |
|---|---|---|

Font: (Default) Times New Roman

| Page 11: [668] Formatted | Microsoft Office User | 8/17/20 2:50:00 PM |
|---|---|---|

Font: (Default) Times New Roman

| Page 11: [668] Formatted | Microsoft Office User | 8/17/20 2:50:00 PM |
|---|---|---|

Font: (Default) Times New Roman

| Page 11: [669] Formatted | Microsoft Office User | 8/6/20 3:49:00 PM |
|---|---|---|

Right, Don't add space between paragraphs of the same style, Line spacing:  single

| Page 11: [670] Formatted | Microsoft Office User | 8/17/20 2:50:00 PM |
|---|---|---|

Font color: Black

| Page 11: [671] Formatted | Microsoft Office User | 8/17/20 2:50:00 PM |
|---|---|---|

Font: (Default) Times New Roman

| Page 11: [671] Formatted | Microsoft Office User | 8/17/20 2:50:00 PM |
|---|---|---|

Font: (Default) Times New Roman

| Page 11: [672] Formatted | Microsoft Office User | 8/17/20 2:50:00 PM |
|---|---|---|

Font: (Default) Times New Roman

| Page 11: [672] Formatted | Microsoft Office User | 8/17/20 2:50:00 PM |
|---|---|---|

Font: (Default) Times New Roman

| Page 11: [673] Formatted | Microsoft Office User | 8/17/20 2:50:00 PM |
|---|---|---|

| Page 11: [673] Formatted | Microsoft Office User | 8/17/20 2:50:00 PM |

Font: (Default) Times New Roman

| Page 11: [674] Formatted | Microsoft Office User | 8/17/20 2:50:00 PM |

Font: (Default) Times New Roman

| Page 11: [674] Formatted | Microsoft Office User | 8/17/20 2:50:00 PM |

Font: (Default) Times New Roman

| Page 11: [675] Formatted | Microsoft Office User | 8/6/20 3:49:00 PM |

Right, Don't add space between paragraphs of the same style, Line spacing: single

| Page 11: [676] Formatted | Microsoft Office User | 8/17/20 2:50:00 PM |

Font color: Black

| Page 11: [677] Formatted | Microsoft Office User | 8/17/20 2:50:00 PM |

Font: (Default) Times New Roman

| Page 11: [677] Formatted | Microsoft Office User | 8/17/20 2:50:00 PM |

Font: (Default) Times New Roman

| Page 11: [678] Formatted | Microsoft Office User | 8/17/20 2:50:00 PM |

Font: (Default) Times New Roman

| Page 11: [678] Formatted | Microsoft Office User | 8/17/20 2:50:00 PM |

Font: (Default) Times New Roman

| Page 11: [679] Formatted | Microsoft Office User | 8/17/20 2:50:00 PM |

Font: (Default) Times New Roman

| Page 11: [679] Formatted | Microsoft Office User | 8/17/20 2:50:00 PM |

Font: (Default) Times New Roman

| Page 11: [680] Formatted | Microsoft Office User | 8/17/20 2:50:00 PM |

Font: (Default) Times New Roman

| Page 11: [680] Formatted | Microsoft Office User | 8/17/20 2:50:00 PM |

Font: (Default) Times New Roman

| Page 11: [681] Formatted | Microsoft Office User | 8/6/20 3:49:00 PM |

| Page 11: [682] Formatted | Microsoft Office User | 8/17/20 2:50:00 PM |
|---|---|---|

Font color: Black

| Page 11: [683] Formatted | Microsoft Office User | 8/17/20 2:50:00 PM |
|---|---|---|

Font: (Default) Times New Roman

| Page 11: [683] Formatted | Microsoft Office User | 8/17/20 2:50:00 PM |
|---|---|---|

Font: (Default) Times New Roman

| Page 11: [684] Formatted | Microsoft Office User | 8/17/20 2:50:00 PM |
|---|---|---|

Font: (Default) Times New Roman

| Page 11: [684] Formatted | Microsoft Office User | 8/17/20 2:50:00 PM |
|---|---|---|

Font: (Default) Times New Roman

| Page 11: [685] Formatted | Microsoft Office User | 8/17/20 2:50:00 PM |
|---|---|---|

Font: (Default) Times New Roman

| Page 11: [685] Formatted | Microsoft Office User | 8/17/20 2:50:00 PM |
|---|---|---|

Font: (Default) Times New Roman

| Page 11: [686] Formatted | Microsoft Office User | 8/17/20 2:50:00 PM |
|---|---|---|

Font: (Default) Times New Roman

| Page 11: [686] Formatted | Microsoft Office User | 8/17/20 2:50:00 PM |
|---|---|---|

Font: (Default) Times New Roman

| Page 11: [687] Formatted | Microsoft Office User | 8/6/20 3:49:00 PM |
|---|---|---|

Right, Don't add space between paragraphs of the same style, Line spacing:  single

| Page 11: [688] Formatted | Microsoft Office User | 8/17/20 2:50:00 PM |
|---|---|---|

Font color: Black

| Page 11: [689] Formatted | Microsoft Office User | 8/17/20 2:50:00 PM |
|---|---|---|

Font: (Default) Times New Roman

| Page 11: [689] Formatted | Microsoft Office User | 8/17/20 2:50:00 PM |
|---|---|---|

Font: (Default) Times New Roman

| Page 11: [690] Formatted | Microsoft Office User | 8/17/20 2:50:00 PM |
|---|---|---|

| Page 11: [690] Formatted | Microsoft Office User | 8/17/20 2:50:00 PM |

Font: (Default) Times New Roman

| Page 11: [691] Formatted | Microsoft Office User | 8/17/20 2:50:00 PM |

Font: (Default) Times New Roman

| Page 11: [691] Formatted | Microsoft Office User | 8/17/20 2:50:00 PM |

Font: (Default) Times New Roman

| Page 11: [692] Formatted | Microsoft Office User | 8/17/20 2:50:00 PM |

Font: (Default) Times New Roman

| Page 11: [692] Formatted | Microsoft Office User | 8/17/20 2:50:00 PM |

Font: (Default) Times New Roman

| Page 11: [693] Formatted | Microsoft Office User | 8/6/20 3:49:00 PM |

Right, Don't add space between paragraphs of the same style, Line spacing:  single

| Page 11: [694] Formatted | Microsoft Office User | 8/17/20 2:50:00 PM |

Font color: Black

| Page 11: [695] Formatted | Microsoft Office User | 8/17/20 2:50:00 PM |

Font: (Default) Times New Roman

| Page 11: [695] Formatted | Microsoft Office User | 8/17/20 2:50:00 PM |

Font: (Default) Times New Roman

| Page 11: [696] Formatted | Microsoft Office User | 8/17/20 2:50:00 PM |

Font: (Default) Times New Roman

| Page 11: [696] Formatted | Microsoft Office User | 8/17/20 2:50:00 PM |

Font: (Default) Times New Roman

| Page 11: [697] Formatted | Microsoft Office User | 8/17/20 2:50:00 PM |

Font: (Default) Times New Roman

| Page 11: [697] Formatted | Microsoft Office User | 8/17/20 2:50:00 PM |

Font: (Default) Times New Roman

| Page 11: [698] Formatted | Microsoft Office User | 8/17/20 2:50:00 PM |

| Page 11: [698] Formatted | Microsoft Office User | 8/17/20 2:50:00 PM |
| --- | --- | --- |

Font: (Default) Times New Roman

| Page 11: [699] Formatted | Microsoft Office User | 8/6/20 3:49:00 PM |
| --- | --- | --- |

Right, Don't add space between paragraphs of the same style, Line spacing:  single

| Page 11: [700] Formatted | Microsoft Office User | 8/17/20 2:50:00 PM |
| --- | --- | --- |

Font color: Black

| Page 11: [701] Formatted | Microsoft Office User | 8/17/20 2:50:00 PM |
| --- | --- | --- |

Font: (Default) Times New Roman

| Page 11: [701] Formatted | Microsoft Office User | 8/17/20 2:50:00 PM |
| --- | --- | --- |

Font: (Default) Times New Roman

| Page 11: [702] Formatted | Microsoft Office User | 8/17/20 2:50:00 PM |
| --- | --- | --- |

Font: (Default) Times New Roman

| Page 11: [702] Formatted | Microsoft Office User | 8/17/20 2:50:00 PM |
| --- | --- | --- |

Font: (Default) Times New Roman

| Page 11: [703] Formatted | Microsoft Office User | 8/17/20 2:50:00 PM |
| --- | --- | --- |

Font: (Default) Times New Roman

| Page 11: [703] Formatted | Microsoft Office User | 8/17/20 2:50:00 PM |
| --- | --- | --- |

Font: (Default) Times New Roman

| Page 11: [704] Formatted | Microsoft Office User | 8/17/20 2:50:00 PM |
| --- | --- | --- |

Font: (Default) Times New Roman

| Page 11: [704] Formatted | Microsoft Office User | 8/17/20 2:50:00 PM |
| --- | --- | --- |

Font: (Default) Times New Roman

| Page 11: [705] Formatted | Microsoft Office User | 8/6/20 3:49:00 PM |
| --- | --- | --- |

Right, Don't add space between paragraphs of the same style, Line spacing:  single

| Page 11: [706] Formatted | Microsoft Office User | 8/17/20 2:50:00 PM |
| --- | --- | --- |

Font color: Black

| Page 11: [707] Formatted | Microsoft Office User | 8/17/20 2:50:00 PM |
| --- | --- | --- |

| Page 11: [707] Formatted | Microsoft Office User | 8/17/20 2:50:00 PM |

Font: (Default) Times New Roman

| Page 11: [708] Formatted | Microsoft Office User | 8/17/20 2:50:00 PM |

Font: (Default) Times New Roman

| Page 11: [708] Formatted | Microsoft Office User | 8/17/20 2:50:00 PM |

Font: (Default) Times New Roman

| Page 11: [709] Formatted | Microsoft Office User | 8/17/20 2:50:00 PM |

Font: (Default) Times New Roman

| Page 11: [709] Formatted | Microsoft Office User | 8/17/20 2:50:00 PM |

Font: (Default) Times New Roman

| Page 11: [710] Formatted | Microsoft Office User | 8/17/20 2:50:00 PM |

Font: (Default) Times New Roman

| Page 11: [710] Formatted | Microsoft Office User | 8/17/20 2:50:00 PM |

Font: (Default) Times New Roman

| Page 11: [711] Formatted | Microsoft Office User | 8/6/20 3:49:00 PM |

Right, Don't add space between paragraphs of the same style, Line spacing:  single

| Page 11: [712] Formatted | Microsoft Office User | 8/17/20 2:50:00 PM |

Font color: Black

| Page 11: [713] Formatted | Microsoft Office User | 8/17/20 2:50:00 PM |

Font: (Default) Times New Roman

| Page 11: [713] Formatted | Microsoft Office User | 8/17/20 2:50:00 PM |

Font: (Default) Times New Roman

| Page 11: [714] Formatted | Microsoft Office User | 8/17/20 2:50:00 PM |

Font: (Default) Times New Roman

| Page 11: [714] Formatted | Microsoft Office User | 8/17/20 2:50:00 PM |

Font: (Default) Times New Roman

| Page 11: [715] Formatted | Microsoft Office User | 8/17/20 2:50:00 PM |

| Page 11: [715] Formatted | Microsoft Office User | 8/17/20 2:50:00 PM |
|---|---|---|

Font: (Default) Times New Roman

| Page 11: [716] Formatted | Microsoft Office User | 8/17/20 2:50:00 PM |
|---|---|---|

Font: (Default) Times New Roman

| Page 11: [716] Formatted | Microsoft Office User | 8/17/20 2:50:00 PM |
|---|---|---|

Font: (Default) Times New Roman

| Page 11: [717] Formatted | Microsoft Office User | 8/6/20 3:49:00 PM |
|---|---|---|

Right, Don't add space between paragraphs of the same style, Line spacing: single

| Page 11: [718] Formatted | Microsoft Office User | 8/17/20 2:50:00 PM |
|---|---|---|

Font color: Black

| Page 11: [719] Formatted | Microsoft Office User | 8/17/20 2:50:00 PM |
|---|---|---|

Font: (Default) Times New Roman

| Page 11: [719] Formatted | Microsoft Office User | 8/17/20 2:50:00 PM |
|---|---|---|

Font: (Default) Times New Roman

| Page 11: [720] Formatted | Microsoft Office User | 8/17/20 2:50:00 PM |
|---|---|---|

Font: (Default) Times New Roman

| Page 11: [720] Formatted | Microsoft Office User | 8/17/20 2:50:00 PM |
|---|---|---|

Font: (Default) Times New Roman

| Page 11: [721] Formatted | Microsoft Office User | 8/17/20 2:50:00 PM |
|---|---|---|

Font: (Default) Times New Roman

| Page 11: [721] Formatted | Microsoft Office User | 8/17/20 2:50:00 PM |
|---|---|---|

Font: (Default) Times New Roman

| Page 11: [722] Formatted | Microsoft Office User | 8/17/20 2:50:00 PM |
|---|---|---|

Font: (Default) Times New Roman

| Page 11: [722] Formatted | Microsoft Office User | 8/17/20 2:50:00 PM |
|---|---|---|

Font: (Default) Times New Roman

| Page 11: [723] Formatted | Microsoft Office User | 8/6/20 3:49:00 PM |
|---|---|---|

| Page 11: [724] Formatted | Microsoft Office User | 8/17/20 2:50:00 PM |
|---|---|---|

Font color: Black

| Page 11: [725] Formatted | Microsoft Office User | 8/17/20 2:50:00 PM |
|---|---|---|

Font: (Default) Times New Roman

| Page 11: [725] Formatted | Microsoft Office User | 8/17/20 2:50:00 PM |
|---|---|---|

Font: (Default) Times New Roman

| Page 11: [726] Formatted | Microsoft Office User | 8/17/20 2:50:00 PM |
|---|---|---|

Font: (Default) Times New Roman

| Page 11: [726] Formatted | Microsoft Office User | 8/17/20 2:50:00 PM |
|---|---|---|

Font: (Default) Times New Roman

| Page 11: [727] Formatted | Microsoft Office User | 8/17/20 2:50:00 PM |
|---|---|---|

Font: (Default) Times New Roman

| Page 11: [727] Formatted | Microsoft Office User | 8/17/20 2:50:00 PM |
|---|---|---|

Font: (Default) Times New Roman

| Page 11: [728] Formatted | Microsoft Office User | 8/17/20 2:50:00 PM |
|---|---|---|

Font: (Default) Times New Roman

| Page 11: [728] Formatted | Microsoft Office User | 8/17/20 2:50:00 PM |
|---|---|---|

Font: (Default) Times New Roman

| Page 11: [729] Formatted | Microsoft Office User | 8/6/20 3:49:00 PM |
|---|---|---|

Right, Don't add space between paragraphs of the same style, Line spacing:  single

| Page 11: [730] Formatted | Microsoft Office User | 8/17/20 2:50:00 PM |
|---|---|---|

Font color: Black

| Page 11: [731] Formatted | Microsoft Office User | 8/17/20 2:50:00 PM |
|---|---|---|

Font: (Default) Times New Roman

| Page 11: [731] Formatted | Microsoft Office User | 8/17/20 2:50:00 PM |
|---|---|---|

Font: (Default) Times New Roman

| Page 11: [732] Formatted | Microsoft Office User | 8/17/20 2:50:00 PM |
|---|---|---|

| Page 11: [732] Formatted | Microsoft Office User | 8/17/20 2:50:00 PM |
|---|---|---|

Font: (Default) Times New Roman

| Page 11: [733] Formatted | Microsoft Office User | 8/17/20 2:50:00 PM |
|---|---|---|

Font: (Default) Times New Roman

| Page 11: [733] Formatted | Microsoft Office User | 8/17/20 2:50:00 PM |
|---|---|---|

Font: (Default) Times New Roman

| Page 11: [734] Formatted | Microsoft Office User | 8/17/20 2:50:00 PM |
|---|---|---|

Font: (Default) Times New Roman

| Page 11: [734] Formatted | Microsoft Office User | 8/17/20 2:50:00 PM |
|---|---|---|

Font: (Default) Times New Roman

| Page 11: [735] Formatted | Microsoft Office User | 8/6/20 3:49:00 PM |
|---|---|---|

Right, Don't add space between paragraphs of the same style, Line spacing:  single

| Page 11: [736] Formatted | Microsoft Office User | 8/17/20 2:50:00 PM |
|---|---|---|

Font color: Black

| Page 11: [737] Formatted | Microsoft Office User | 8/17/20 2:50:00 PM |
|---|---|---|

Font: (Default) Times New Roman

| Page 11: [737] Formatted | Microsoft Office User | 8/17/20 2:50:00 PM |
|---|---|---|

Font: (Default) Times New Roman

| Page 11: [738] Formatted | Microsoft Office User | 8/17/20 2:50:00 PM |
|---|---|---|

Font: (Default) Times New Roman

| Page 11: [738] Formatted | Microsoft Office User | 8/17/20 2:50:00 PM |
|---|---|---|

Font: (Default) Times New Roman

| Page 11: [739] Formatted | Microsoft Office User | 8/17/20 2:50:00 PM |
|---|---|---|

Font: (Default) Times New Roman

| Page 11: [739] Formatted | Microsoft Office User | 8/17/20 2:50:00 PM |
|---|---|---|

Font: (Default) Times New Roman

| Page 11: [740] Formatted | Microsoft Office User | 8/17/20 2:50:00 PM |
|---|---|---|

| Page 11: [740] Formatted | Microsoft Office User | 8/17/20 2:50:00 PM |
|---|---|---|

Font: (Default) Times New Roman

| Page 11: [741] Formatted | Microsoft Office User | 8/6/20 3:49:00 PM |
|---|---|---|

Right, Don't add space between paragraphs of the same style, Line spacing:  single

| Page 11: [742] Formatted | Microsoft Office User | 8/17/20 2:50:00 PM |
|---|---|---|

Font color: Black

| Page 11: [743] Formatted | Microsoft Office User | 8/17/20 2:50:00 PM |
|---|---|---|

Font: (Default) Times New Roman

| Page 11: [743] Formatted | Microsoft Office User | 8/17/20 2:50:00 PM |
|---|---|---|

Font: (Default) Times New Roman

| Page 11: [744] Formatted | Microsoft Office User | 8/17/20 2:50:00 PM |
|---|---|---|

Font: (Default) Times New Roman

| Page 11: [744] Formatted | Microsoft Office User | 8/17/20 2:50:00 PM |
|---|---|---|

Font: (Default) Times New Roman

| Page 11: [745] Formatted | Microsoft Office User | 8/17/20 2:50:00 PM |
|---|---|---|

Font: (Default) Times New Roman

| Page 11: [745] Formatted | Microsoft Office User | 8/17/20 2:50:00 PM |
|---|---|---|

Font: (Default) Times New Roman

| Page 11: [746] Formatted | Microsoft Office User | 8/17/20 2:50:00 PM |
|---|---|---|

Font: (Default) Times New Roman

| Page 11: [746] Formatted | Microsoft Office User | 8/17/20 2:50:00 PM |
|---|---|---|

Font: (Default) Times New Roman

| Page 11: [747] Formatted | Microsoft Office User | 8/6/20 3:49:00 PM |
|---|---|---|

Right, Don't add space between paragraphs of the same style, Line spacing:  single

| Page 11: [748] Formatted | Microsoft Office User | 8/17/20 2:50:00 PM |
|---|---|---|

Font color: Black

| Page 11: [749] Formatted | Microsoft Office User | 8/17/20 2:50:00 PM |
|---|---|---|

| Page 11: [749] Formatted | Microsoft Office User | 8/17/20 2:50:00 PM |
|---|---|---|

Font: (Default) Times New Roman

| Page 11: [750] Formatted | Microsoft Office User | 8/17/20 2:50:00 PM |
|---|---|---|

Font: (Default) Times New Roman

| Page 11: [750] Formatted | Microsoft Office User | 8/17/20 2:50:00 PM |
|---|---|---|

Font: (Default) Times New Roman

| Page 11: [751] Formatted | Microsoft Office User | 8/17/20 2:50:00 PM |
|---|---|---|

Font: (Default) Times New Roman

| Page 11: [751] Formatted | Microsoft Office User | 8/17/20 2:50:00 PM |
|---|---|---|

Font: (Default) Times New Roman

| Page 11: [752] Formatted | Microsoft Office User | 8/17/20 2:50:00 PM |
|---|---|---|

Font: (Default) Times New Roman

| Page 11: [752] Formatted | Microsoft Office User | 8/17/20 2:50:00 PM |
|---|---|---|

Font: (Default) Times New Roman

| Page 11: [753] Formatted | Microsoft Office User | 8/6/20 3:49:00 PM |
|---|---|---|

Right, Don't add space between paragraphs of the same style, Line spacing:  single

| Page 11: [754] Formatted | Microsoft Office User | 8/17/20 2:50:00 PM |
|---|---|---|

Font color: Black

| Page 11: [755] Formatted | Microsoft Office User | 8/17/20 2:50:00 PM |
|---|---|---|

Font: (Default) Times New Roman

| Page 11: [755] Formatted | Microsoft Office User | 8/17/20 2:50:00 PM |
|---|---|---|

Font: (Default) Times New Roman

| Page 11: [756] Formatted | Microsoft Office User | 8/17/20 2:50:00 PM |
|---|---|---|

Font: (Default) Times New Roman

| Page 11: [756] Formatted | Microsoft Office User | 8/17/20 2:50:00 PM |
|---|---|---|

Font: (Default) Times New Roman

| Page 11: [757] Formatted | Microsoft Office User | 8/17/20 2:50:00 PM |
|---|---|---|

| Page 11: [757] Formatted | Microsoft Office User | 8/17/20 2:50:00 PM |
|---|---|---|

Font: (Default) Times New Roman

| Page 11: [758] Formatted | Microsoft Office User | 8/17/20 2:50:00 PM |
|---|---|---|

Font: (Default) Times New Roman

| Page 11: [758] Formatted | Microsoft Office User | 8/17/20 2:50:00 PM |
|---|---|---|

Font: (Default) Times New Roman

| Page 12: [759] Formatted | Microsoft Office User | 8/6/20 3:49:00 PM |
|---|---|---|

Right, Don't add space between paragraphs of the same style, Line spacing: single

| Page 12: [760] Formatted | Microsoft Office User | 8/17/20 2:50:00 PM |
|---|---|---|

Font color: Black

| Page 12: [761] Formatted | Microsoft Office User | 8/17/20 2:50:00 PM |
|---|---|---|

Font: (Default) Times New Roman

| Page 12: [761] Formatted | Microsoft Office User | 8/17/20 2:50:00 PM |
|---|---|---|

Font: (Default) Times New Roman

| Page 12: [762] Formatted | Microsoft Office User | 8/17/20 2:50:00 PM |
|---|---|---|

Font: (Default) Times New Roman

| Page 12: [762] Formatted | Microsoft Office User | 8/17/20 2:50:00 PM |
|---|---|---|

Font: (Default) Times New Roman

| Page 12: [763] Formatted | Microsoft Office User | 8/17/20 2:50:00 PM |
|---|---|---|

Font: (Default) Times New Roman

| Page 12: [763] Formatted | Microsoft Office User | 8/17/20 2:50:00 PM |
|---|---|---|

Font: (Default) Times New Roman

| Page 12: [764] Formatted | Microsoft Office User | 8/17/20 2:50:00 PM |
|---|---|---|

Font: (Default) Times New Roman

| Page 12: [764] Formatted | Microsoft Office User | 8/17/20 2:50:00 PM |
|---|---|---|

Font: (Default) Times New Roman

| Page 12: [765] Formatted | Microsoft Office User | 8/6/20 3:49:00 PM |
|---|---|---|

| Page 12: [766] Formatted | Microsoft Office User | 8/17/20 2:50:00 PM |

Font color: Black

| Page 12: [767] Formatted | Microsoft Office User | 8/17/20 2:50:00 PM |

Font: (Default) Times New Roman

| Page 12: [767] Formatted | Microsoft Office User | 8/17/20 2:50:00 PM |

Font: (Default) Times New Roman

| Page 12: [768] Formatted | Microsoft Office User | 8/17/20 2:50:00 PM |

Font: (Default) Times New Roman

| Page 12: [768] Formatted | Microsoft Office User | 8/17/20 2:50:00 PM |

Font: (Default) Times New Roman

| Page 12: [769] Formatted | Microsoft Office User | 8/17/20 2:50:00 PM |

Font: (Default) Times New Roman

| Page 12: [769] Formatted | Microsoft Office User | 8/17/20 2:50:00 PM |

Font: (Default) Times New Roman

| Page 12: [770] Formatted | Microsoft Office User | 8/17/20 2:50:00 PM |

Font: (Default) Times New Roman

| Page 12: [770] Formatted | Microsoft Office User | 8/17/20 2:50:00 PM |

Font: (Default) Times New Roman

| Page 12: [771] Formatted | Microsoft Office User | 8/6/20 3:49:00 PM |

Right, Don't add space between paragraphs of the same style, Line spacing:  single

| Page 12: [772] Formatted | Microsoft Office User | 8/17/20 2:50:00 PM |

Font color: Black

| Page 12: [773] Formatted | Microsoft Office User | 8/17/20 2:50:00 PM |

Font: (Default) Times New Roman

| Page 12: [773] Formatted | Microsoft Office User | 8/17/20 2:50:00 PM |

Font: (Default) Times New Roman

| Page 12: [774] Formatted | Microsoft Office User | 8/17/20 2:50:00 PM |

| Page 12: [774] Formatted | Microsoft Office User | 8/17/20 2:50:00 PM |

Font: (Default) Times New Roman

| Page 12: [775] Formatted | Microsoft Office User | 8/17/20 2:50:00 PM |

Font: (Default) Times New Roman

| Page 12: [775] Formatted | Microsoft Office User | 8/17/20 2:50:00 PM |

Font: (Default) Times New Roman

| Page 12: [776] Formatted | Microsoft Office User | 8/17/20 2:50:00 PM |

Font: (Default) Times New Roman

| Page 12: [776] Formatted | Microsoft Office User | 8/17/20 2:50:00 PM |

Font: (Default) Times New Roman

| Page 12: [777] Formatted | Microsoft Office User | 8/6/20 3:49:00 PM |

Right, Don't add space between paragraphs of the same style, Line spacing:  single

| Page 12: [778] Formatted | Microsoft Office User | 8/17/20 2:50:00 PM |

Font color: Black

| Page 12: [779] Formatted | Microsoft Office User | 8/17/20 2:50:00 PM |

Font: (Default) Times New Roman

| Page 12: [779] Formatted | Microsoft Office User | 8/17/20 2:50:00 PM |

Font: (Default) Times New Roman

| Page 12: [780] Formatted | Microsoft Office User | 8/17/20 2:50:00 PM |

Font: (Default) Times New Roman

| Page 12: [780] Formatted | Microsoft Office User | 8/17/20 2:50:00 PM |

Font: (Default) Times New Roman

| Page 12: [781] Formatted | Microsoft Office User | 8/17/20 2:50:00 PM |

Font: (Default) Times New Roman

| Page 12: [781] Formatted | Microsoft Office User | 8/17/20 2:50:00 PM |

Font: (Default) Times New Roman

| Page 12: [782] Formatted | Microsoft Office User | 8/17/20 2:50:00 PM |

| Page 12: [782] Formatted | Microsoft Office User | 8/17/20 2:50:00 PM |
|---|---|---|

Font: (Default) Times New Roman

| Page 12: [783] Formatted | Microsoft Office User | 8/6/20 3:49:00 PM |
|---|---|---|

Right, Don't add space between paragraphs of the same style, Line spacing:  single

| Page 12: [784] Formatted | Microsoft Office User | 8/17/20 2:50:00 PM |
|---|---|---|

Font color: Black

| Page 12: [785] Formatted | Microsoft Office User | 8/17/20 2:50:00 PM |
|---|---|---|

Font: (Default) Times New Roman

| Page 12: [785] Formatted | Microsoft Office User | 8/17/20 2:50:00 PM |
|---|---|---|

Font: (Default) Times New Roman

| Page 12: [786] Formatted | Microsoft Office User | 8/17/20 2:50:00 PM |
|---|---|---|

Font: (Default) Times New Roman

| Page 12: [786] Formatted | Microsoft Office User | 8/17/20 2:50:00 PM |
|---|---|---|

Font: (Default) Times New Roman

| Page 12: [787] Formatted | Microsoft Office User | 8/17/20 2:50:00 PM |
|---|---|---|

Font: (Default) Times New Roman

| Page 12: [787] Formatted | Microsoft Office User | 8/17/20 2:50:00 PM |
|---|---|---|

Font: (Default) Times New Roman

| Page 12: [788] Formatted | Microsoft Office User | 8/17/20 2:50:00 PM |
|---|---|---|

Font: (Default) Times New Roman

| Page 12: [788] Formatted | Microsoft Office User | 8/17/20 2:50:00 PM |
|---|---|---|

Font: (Default) Times New Roman

| Page 12: [789] Formatted | Microsoft Office User | 8/6/20 3:49:00 PM |
|---|---|---|

Right, Don't add space between paragraphs of the same style, Line spacing:  single

| Page 12: [790] Formatted | Microsoft Office User | 8/17/20 2:50:00 PM |
|---|---|---|

Font color: Black

| Page 12: [791] Formatted | Microsoft Office User | 8/17/20 2:50:00 PM |
|---|---|---|

| Page 12: [791] Formatted | Microsoft Office User | 8/17/20 2:50:00 PM |
|---|---|---|

Font: (Default) Times New Roman

| Page 12: [792] Formatted | Microsoft Office User | 8/17/20 2:50:00 PM |
|---|---|---|

Font: (Default) Times New Roman

| Page 12: [792] Formatted | Microsoft Office User | 8/17/20 2:50:00 PM |
|---|---|---|

Font: (Default) Times New Roman

| Page 12: [793] Formatted | Microsoft Office User | 8/17/20 2:50:00 PM |
|---|---|---|

Font: (Default) Times New Roman

| Page 12: [793] Formatted | Microsoft Office User | 8/17/20 2:50:00 PM |
|---|---|---|

Font: (Default) Times New Roman

| Page 12: [794] Formatted | Microsoft Office User | 8/17/20 2:50:00 PM |
|---|---|---|

Font: (Default) Times New Roman

| Page 12: [794] Formatted | Microsoft Office User | 8/17/20 2:50:00 PM |
|---|---|---|

Font: (Default) Times New Roman

| Page 12: [795] Formatted | Microsoft Office User | 8/6/20 3:49:00 PM |
|---|---|---|

Right, Don't add space between paragraphs of the same style, Line spacing:  single

| Page 12: [796] Formatted | Microsoft Office User | 8/17/20 2:50:00 PM |
|---|---|---|

Font color: Black

| Page 12: [797] Formatted | Microsoft Office User | 8/17/20 2:50:00 PM |
|---|---|---|

Font: (Default) Times New Roman

| Page 12: [797] Formatted | Microsoft Office User | 8/17/20 2:50:00 PM |
|---|---|---|

Font: (Default) Times New Roman

| Page 12: [798] Formatted | Microsoft Office User | 8/17/20 2:50:00 PM |
|---|---|---|

Font: (Default) Times New Roman

| Page 12: [798] Formatted | Microsoft Office User | 8/17/20 2:50:00 PM |
|---|---|---|

Font: (Default) Times New Roman

| Page 12: [799] Formatted | Microsoft Office User | 8/17/20 2:50:00 PM |
|---|---|---|

| Page 12: [799] Formatted | Microsoft Office User | 8/17/20 2:50:00 PM |
|---|---|---|

Font: (Default) Times New Roman

| Page 12: [800] Formatted | Microsoft Office User | 8/17/20 2:50:00 PM |
|---|---|---|

Font: (Default) Times New Roman

| Page 12: [800] Formatted | Microsoft Office User | 8/17/20 2:50:00 PM |
|---|---|---|

Font: (Default) Times New Roman

| Page 12: [801] Formatted | Microsoft Office User | 8/6/20 3:49:00 PM |
|---|---|---|

Right, Don't add space between paragraphs of the same style, Line spacing:  single

| Page 12: [802] Formatted | Microsoft Office User | 8/17/20 2:50:00 PM |
|---|---|---|

Font color: Black

| Page 12: [803] Formatted | Microsoft Office User | 8/17/20 2:50:00 PM |
|---|---|---|

Font: (Default) Times New Roman

| Page 12: [803] Formatted | Microsoft Office User | 8/17/20 2:50:00 PM |
|---|---|---|

Font: (Default) Times New Roman

| Page 12: [804] Formatted | Microsoft Office User | 8/17/20 2:50:00 PM |
|---|---|---|

Font: (Default) Times New Roman

| Page 12: [804] Formatted | Microsoft Office User | 8/17/20 2:50:00 PM |
|---|---|---|

Font: (Default) Times New Roman

| Page 12: [805] Formatted | Microsoft Office User | 8/17/20 2:50:00 PM |
|---|---|---|

Font: (Default) Times New Roman

| Page 12: [805] Formatted | Microsoft Office User | 8/17/20 2:50:00 PM |
|---|---|---|

Font: (Default) Times New Roman

| Page 12: [806] Formatted | Microsoft Office User | 8/17/20 2:50:00 PM |
|---|---|---|

Font: (Default) Times New Roman

| Page 12: [806] Formatted | Microsoft Office User | 8/17/20 2:50:00 PM |
|---|---|---|

Font: (Default) Times New Roman

| Page 12: [807] Formatted | Microsoft Office User | 8/6/20 3:49:00 PM |
|---|---|---|

| Page 12: [808] Formatted | Microsoft Office User | 8/17/20 2:50:00 PM |
|---|---|---|

Font color: Black

| Page 12: [809] Formatted | Microsoft Office User | 8/17/20 2:50:00 PM |
|---|---|---|

Font: (Default) Times New Roman

| Page 12: [809] Formatted | Microsoft Office User | 8/17/20 2:50:00 PM |
|---|---|---|

Font: (Default) Times New Roman

| Page 12: [810] Formatted | Microsoft Office User | 8/17/20 2:50:00 PM |
|---|---|---|

Font: (Default) Times New Roman

| Page 12: [810] Formatted | Microsoft Office User | 8/17/20 2:50:00 PM |
|---|---|---|

Font: (Default) Times New Roman

| Page 12: [811] Formatted | Microsoft Office User | 8/17/20 2:50:00 PM |
|---|---|---|

Font: (Default) Times New Roman

| Page 12: [811] Formatted | Microsoft Office User | 8/17/20 2:50:00 PM |
|---|---|---|

Font: (Default) Times New Roman

| Page 12: [812] Formatted | Microsoft Office User | 8/17/20 2:50:00 PM |
|---|---|---|

Font: (Default) Times New Roman

| Page 12: [812] Formatted | Microsoft Office User | 8/17/20 2:50:00 PM |
|---|---|---|

Font: (Default) Times New Roman

| Page 12: [813] Formatted | Microsoft Office User | 8/6/20 3:49:00 PM |
|---|---|---|

Right, Don't add space between paragraphs of the same style, Line spacing:  single

| Page 12: [814] Formatted | Microsoft Office User | 8/17/20 2:50:00 PM |
|---|---|---|

Font color: Black

| Page 12: [815] Formatted | Microsoft Office User | 8/17/20 2:50:00 PM |
|---|---|---|

Font: (Default) Times New Roman

| Page 12: [815] Formatted | Microsoft Office User | 8/17/20 2:50:00 PM |
|---|---|---|

Font: (Default) Times New Roman

| Page 12: [816] Formatted | Microsoft Office User | 8/17/20 2:50:00 PM |
|---|---|---|

| Page 12: [816] Formatted | Microsoft Office User | 8/17/20 2:50:00 PM |
|---|---|---|

Font: (Default) Times New Roman

| Page 12: [817] Formatted | Microsoft Office User | 8/17/20 2:50:00 PM |
|---|---|---|

Font: (Default) Times New Roman

| Page 12: [817] Formatted | Microsoft Office User | 8/17/20 2:50:00 PM |
|---|---|---|

Font: (Default) Times New Roman

| Page 12: [818] Formatted | Microsoft Office User | 8/17/20 2:50:00 PM |
|---|---|---|

Font: (Default) Times New Roman

| Page 12: [818] Formatted | Microsoft Office User | 8/17/20 2:50:00 PM |
|---|---|---|

Font: (Default) Times New Roman

| Page 12: [819] Formatted | Microsoft Office User | 8/6/20 3:49:00 PM |
|---|---|---|

Right, Don't add space between paragraphs of the same style, Line spacing:  single

| Page 12: [820] Formatted | Microsoft Office User | 8/17/20 2:50:00 PM |
|---|---|---|

Font color: Black

| Page 12: [821] Formatted | Microsoft Office User | 8/17/20 2:50:00 PM |
|---|---|---|

Font: (Default) Times New Roman

| Page 12: [821] Formatted | Microsoft Office User | 8/17/20 2:50:00 PM |
|---|---|---|

Font: (Default) Times New Roman

| Page 12: [822] Formatted | Microsoft Office User | 8/17/20 2:50:00 PM |
|---|---|---|

Font: (Default) Times New Roman

| Page 12: [822] Formatted | Microsoft Office User | 8/17/20 2:50:00 PM |
|---|---|---|

Font: (Default) Times New Roman

| Page 12: [823] Formatted | Microsoft Office User | 8/17/20 2:50:00 PM |
|---|---|---|

Font: (Default) Times New Roman

| Page 12: [823] Formatted | Microsoft Office User | 8/17/20 2:50:00 PM |
|---|---|---|

Font: (Default) Times New Roman

| Page 12: [824] Formatted | Microsoft Office User | 8/17/20 2:50:00 PM |
|---|---|---|

| Page 12: [824] Formatted | Microsoft Office User | 8/17/20 2:50:00 PM |
|---|---|---|

Font: (Default) Times New Roman

| Page 12: [825] Formatted | Microsoft Office User | 8/6/20 3:49:00 PM |
|---|---|---|

Right, Don't add space between paragraphs of the same style, Line spacing:  single

| Page 12: [826] Formatted | Microsoft Office User | 8/17/20 2:50:00 PM |
|---|---|---|

Font color: Black

| Page 12: [827] Formatted | Microsoft Office User | 8/17/20 2:50:00 PM |
|---|---|---|

Font: (Default) Times New Roman

| Page 12: [827] Formatted | Microsoft Office User | 8/17/20 2:50:00 PM |
|---|---|---|

Font: (Default) Times New Roman

| Page 12: [828] Formatted | Microsoft Office User | 8/17/20 2:50:00 PM |
|---|---|---|

Font: (Default) Times New Roman

| Page 12: [828] Formatted | Microsoft Office User | 8/17/20 2:50:00 PM |
|---|---|---|

Font: (Default) Times New Roman

| Page 12: [829] Formatted | Microsoft Office User | 8/17/20 2:50:00 PM |
|---|---|---|

Font: (Default) Times New Roman

| Page 12: [829] Formatted | Microsoft Office User | 8/17/20 2:50:00 PM |
|---|---|---|

Font: (Default) Times New Roman

| Page 12: [830] Formatted | Microsoft Office User | 8/17/20 2:50:00 PM |
|---|---|---|

Font: (Default) Times New Roman

| Page 12: [830] Formatted | Microsoft Office User | 8/17/20 2:50:00 PM |
|---|---|---|

Font: (Default) Times New Roman

| Page 12: [831] Formatted | Microsoft Office User | 8/6/20 3:49:00 PM |
|---|---|---|

Right, Don't add space between paragraphs of the same style, Line spacing:  single

| Page 12: [832] Formatted | Microsoft Office User | 8/17/20 2:50:00 PM |
|---|---|---|

Font color: Black

| Page 12: [833] Formatted | Microsoft Office User | 8/17/20 2:50:00 PM |
|---|---|---|

| Page 12: [833] Formatted | Microsoft Office User | 8/17/20 2:50:00 PM |
|---|---|---|

Font: (Default) Times New Roman

| Page 12: [834] Formatted | Microsoft Office User | 8/17/20 2:50:00 PM |
|---|---|---|

Font: (Default) Times New Roman

| Page 12: [834] Formatted | Microsoft Office User | 8/17/20 2:50:00 PM |
|---|---|---|

Font: (Default) Times New Roman

| Page 12: [835] Formatted | Microsoft Office User | 8/17/20 2:50:00 PM |
|---|---|---|

Font: (Default) Times New Roman

| Page 12: [835] Formatted | Microsoft Office User | 8/17/20 2:50:00 PM |
|---|---|---|

Font: (Default) Times New Roman

| Page 12: [836] Formatted | Microsoft Office User | 8/17/20 2:50:00 PM |
|---|---|---|

Font: (Default) Times New Roman

| Page 12: [836] Formatted | Microsoft Office User | 8/17/20 2:50:00 PM |
|---|---|---|

Font: (Default) Times New Roman

| Page 12: [837] Formatted | Microsoft Office User | 8/6/20 3:49:00 PM |
|---|---|---|

Right, Don't add space between paragraphs of the same style, Line spacing:  single

| Page 12: [838] Formatted | Microsoft Office User | 8/17/20 2:50:00 PM |
|---|---|---|

Font color: Black

| Page 12: [839] Formatted | Microsoft Office User | 8/17/20 2:50:00 PM |
|---|---|---|

Font: (Default) Times New Roman

| Page 12: [839] Formatted | Microsoft Office User | 8/17/20 2:50:00 PM |
|---|---|---|

Font: (Default) Times New Roman

| Page 12: [840] Formatted | Microsoft Office User | 8/17/20 2:50:00 PM |
|---|---|---|

Font: (Default) Times New Roman

| Page 12: [840] Formatted | Microsoft Office User | 8/17/20 2:50:00 PM |
|---|---|---|

Font: (Default) Times New Roman

| Page 12: [841] Formatted | Microsoft Office User | 8/17/20 2:50:00 PM |
|---|---|---|

| Page 12: [841] Formatted | Microsoft Office User | 8/17/20 2:50:00 PM |
|---|---|---|

Font: (Default) Times New Roman

| Page 12: [842] Formatted | Microsoft Office User | 8/17/20 2:50:00 PM |
|---|---|---|

Font: (Default) Times New Roman

| Page 12: [842] Formatted | Microsoft Office User | 8/17/20 2:50:00 PM |
|---|---|---|

Font: (Default) Times New Roman

| Page 12: [843] Formatted | Microsoft Office User | 8/6/20 3:49:00 PM |
|---|---|---|

Right, Don't add space between paragraphs of the same style, Line spacing:  single

| Page 12: [844] Formatted | Microsoft Office User | 8/17/20 2:50:00 PM |
|---|---|---|

Font color: Black

| Page 12: [845] Formatted | Microsoft Office User | 8/17/20 2:50:00 PM |
|---|---|---|

Font: (Default) Times New Roman

| Page 12: [845] Formatted | Microsoft Office User | 8/17/20 2:50:00 PM |
|---|---|---|

Font: (Default) Times New Roman

| Page 12: [846] Formatted | Microsoft Office User | 8/17/20 2:50:00 PM |
|---|---|---|

Font: (Default) Times New Roman

| Page 12: [846] Formatted | Microsoft Office User | 8/17/20 2:50:00 PM |
|---|---|---|

Font: (Default) Times New Roman

| Page 12: [847] Formatted | Microsoft Office User | 8/17/20 2:50:00 PM |
|---|---|---|

Font: (Default) Times New Roman

| Page 12: [847] Formatted | Microsoft Office User | 8/17/20 2:50:00 PM |
|---|---|---|

Font: (Default) Times New Roman

| Page 12: [848] Formatted | Microsoft Office User | 8/17/20 2:50:00 PM |
|---|---|---|

Font: (Default) Times New Roman

| Page 12: [848] Formatted | Microsoft Office User | 8/17/20 2:50:00 PM |
|---|---|---|

Font: (Default) Times New Roman

| Page 12: [849] Formatted | Microsoft Office User | 8/6/20 3:49:00 PM |
|---|---|---|

| Page 12: [850] Formatted | Microsoft Office User | 8/17/20 2:50:00 PM |
|---|---|---|

Font color: Black

| Page 12: [851] Formatted | Microsoft Office User | 8/17/20 2:50:00 PM |
|---|---|---|

Font: (Default) Times New Roman

| Page 12: [851] Formatted | Microsoft Office User | 8/17/20 2:50:00 PM |
|---|---|---|

Font: (Default) Times New Roman

| Page 12: [852] Formatted | Microsoft Office User | 8/17/20 2:50:00 PM |
|---|---|---|

Font: (Default) Times New Roman

| Page 12: [852] Formatted | Microsoft Office User | 8/17/20 2:50:00 PM |
|---|---|---|

Font: (Default) Times New Roman

| Page 12: [853] Formatted | Microsoft Office User | 8/17/20 2:50:00 PM |
|---|---|---|

Font: (Default) Times New Roman

| Page 12: [853] Formatted | Microsoft Office User | 8/17/20 2:50:00 PM |
|---|---|---|

Font: (Default) Times New Roman

| Page 12: [854] Formatted | Microsoft Office User | 8/17/20 2:50:00 PM |
|---|---|---|

Font: (Default) Times New Roman

| Page 12: [854] Formatted | Microsoft Office User | 8/17/20 2:50:00 PM |
|---|---|---|

Font: (Default) Times New Roman

| Page 12: [855] Formatted | Microsoft Office User | 8/6/20 3:49:00 PM |
|---|---|---|

Right, Don't add space between paragraphs of the same style, Line spacing:  single

| Page 12: [856] Formatted | Microsoft Office User | 8/17/20 2:50:00 PM |
|---|---|---|

Font color: Black

| Page 12: [857] Formatted | Microsoft Office User | 8/17/20 2:50:00 PM |
|---|---|---|

Font: (Default) Times New Roman

| Page 12: [857] Formatted | Microsoft Office User | 8/17/20 2:50:00 PM |
|---|---|---|

Font: (Default) Times New Roman

| Page 12: [858] Formatted | Microsoft Office User | 8/17/20 2:50:00 PM |
|---|---|---|

| Page 12: [858] Formatted | Microsoft Office User | 8/17/20 2:50:00 PM |
|---|---|---|

Font: (Default) Times New Roman

| Page 12: [859] Formatted | Microsoft Office User | 8/17/20 2:50:00 PM |
|---|---|---|

Font: (Default) Times New Roman

| Page 12: [859] Formatted | Microsoft Office User | 8/17/20 2:50:00 PM |
|---|---|---|

Font: (Default) Times New Roman

| Page 12: [860] Formatted | Microsoft Office User | 8/17/20 2:50:00 PM |
|---|---|---|

Font: (Default) Times New Roman

| Page 12: [860] Formatted | Microsoft Office User | 8/17/20 2:50:00 PM |
|---|---|---|

Font: (Default) Times New Roman

| Page 12: [861] Formatted | Microsoft Office User | 8/6/20 3:49:00 PM |
|---|---|---|

Right, Don't add space between paragraphs of the same style, Line spacing:  single

| Page 12: [862] Formatted | Microsoft Office User | 8/17/20 2:50:00 PM |
|---|---|---|

Font color: Black

| Page 12: [863] Formatted | Microsoft Office User | 8/17/20 2:50:00 PM |
|---|---|---|

Font: (Default) Times New Roman

| Page 12: [863] Formatted | Microsoft Office User | 8/17/20 2:50:00 PM |
|---|---|---|

Font: (Default) Times New Roman

| Page 12: [864] Formatted | Microsoft Office User | 8/17/20 2:50:00 PM |
|---|---|---|

Font: (Default) Times New Roman

| Page 12: [864] Formatted | Microsoft Office User | 8/17/20 2:50:00 PM |
|---|---|---|

Font: (Default) Times New Roman

| Page 12: [865] Formatted | Microsoft Office User | 8/17/20 2:50:00 PM |
|---|---|---|

Font: (Default) Times New Roman

| Page 12: [865] Formatted | Microsoft Office User | 8/17/20 2:50:00 PM |
|---|---|---|

Font: (Default) Times New Roman

| Page 12: [866] Formatted | Microsoft Office User | 8/17/20 2:50:00 PM |
|---|---|---|

| Page 12: [866] Formatted | Microsoft Office User | 8/17/20 2:50:00 PM |
|---|---|---|

Font: (Default) Times New Roman

| Page 12: [867] Formatted | Microsoft Office User | 8/6/20 3:49:00 PM |
|---|---|---|

Right, Don't add space between paragraphs of the same style, Line spacing: single

| Page 12: [868] Formatted | Microsoft Office User | 8/17/20 2:50:00 PM |
|---|---|---|

Font color: Black

| Page 12: [869] Formatted | Microsoft Office User | 8/17/20 2:50:00 PM |
|---|---|---|

Font: (Default) Times New Roman

| Page 12: [869] Formatted | Microsoft Office User | 8/17/20 2:50:00 PM |
|---|---|---|

Font: (Default) Times New Roman

| Page 12: [870] Formatted | Microsoft Office User | 8/17/20 2:50:00 PM |
|---|---|---|

Font: (Default) Times New Roman

| Page 12: [870] Formatted | Microsoft Office User | 8/17/20 2:50:00 PM |
|---|---|---|

Font: (Default) Times New Roman

| Page 12: [871] Formatted | Microsoft Office User | 8/17/20 2:50:00 PM |
|---|---|---|

Font: (Default) Times New Roman

| Page 12: [871] Formatted | Microsoft Office User | 8/17/20 2:50:00 PM |
|---|---|---|

Font: (Default) Times New Roman

| Page 12: [872] Formatted | Microsoft Office User | 8/17/20 2:50:00 PM |
|---|---|---|

Font: (Default) Times New Roman

| Page 12: [872] Formatted | Microsoft Office User | 8/17/20 2:50:00 PM |
|---|---|---|

Font: (Default) Times New Roman

| Page 12: [873] Formatted | Microsoft Office User | 8/6/20 3:49:00 PM |
|---|---|---|

Right, Don't add space between paragraphs of the same style, Line spacing: single

| Page 12: [874] Formatted | Microsoft Office User | 8/17/20 2:50:00 PM |
|---|---|---|

Font color: Black

| Page 12: [875] Formatted | Microsoft Office User | 8/17/20 2:50:00 PM |
|---|---|---|

| Page 12: [875] Formatted | Microsoft Office User | 8/17/20 2:50:00 PM |
|---|---|---|

Font: (Default) Times New Roman

[remaining 13,790 characters of this post omitted]